# Topological Blindspots: Understanding and Extending Topological Deep Learning Through the Lens of Expressivity

**Yam Eitan**[*,1]  **Yoav Gelberg**[*,2]  **Guy Bar-Shalom**[1]  **Fabrizio Frasca**[1]
**Michael Bronstein**[2,3]  **Haggai Maron**[1,4]
[1]Technion – Israel Institute of Technology  [2]University of Oxford  [3]AITHYRA
[4]NVIDIA Research

## Abstract

Topological deep learning (TDL) is a rapidly growing field that seeks to leverage topological structure in data and facilitate learning from data supported on topological objects, ranging from molecules to 3D shapes. Most TDL architectures can be unified under the framework of higher-order message-passing (HOMP), which generalizes graph message-passing to higher-order domains. In the first part of the paper, we explore HOMP's expressive power from a topological perspective, demonstrating the framework's inability to capture fundamental topological and metric invariants such as diameter, orientability, planarity, and homology. In addition, we demonstrate HOMP's limitations in fully leveraging lifting and pooling methods on graphs. To the best of our knowledge, this is the first work to study the expressivity of TDL from a *topological* perspective. In the second part of the paper, we develop two new classes of architectures – multi-cellular networks (MCN) and scalable MCN (SMCN) – which draw inspiration from expressive GNNs. MCN can reach full expressivity, but scaling it to large data objects can be computationally expansive. Designed as a more scalable alternative, SMCN still mitigates many of HOMP's expressivity limitations. Finally, we create new benchmarks for evaluating models based on their ability to learn topological properties of complexes. We then evaluate SMCN on these benchmarks and on real-world graph datasets, demonstrating improvements over both HOMP baselines and expressive graph methods, highlighting the value of expressively leveraging topological information. Code and data are available at
`https://github.com/yoavgelberg/SMCN`.

## 1 Introduction

Topological Deep Learning (TDL) is an emerging field focused on learning from data supported on topological objects. Higher-order message-passing (HOMP) (Hajij et al., 2022a;b) has emerged as a key framework in TDL, unifying architectures designed for various topological data types. Originally introduced for simplicial complexes (Bodnar et al., 2021b), HOMP has been successively adapted for cellular complexes (Bodnar et al., 2021a; Hajij et al., 2020), and more recently, for combinatorial complexes (Hajij et al., 2022a;b). Each adaptation is a direct generalization of its predecessor. The HOMP framework extends traditional message-passing neural networks (MPNNs) (Gilmer et al., 2017), widely used in graph learning, to higher-order topological domains.

Despite their widespread adoption in various graph learning applications, MPNNs are known to struggle with expressivity limitations, often failing to distinguish even simple non-isomorphic graphs (Morris et al., 2019; Xu et al., 2018). This realization has led to a substantial body of work dedicated to developing more expressive graph architectures (Morris et al., 2023; Maron et al., 2019; Morris et al., 2019; Bevilacqua et al., 2021; Abboud et al., 2020; Bouritsas et al., 2022). Given the similarity between HOMP and MPNNs, a natural question arises: *What are the limitations of higher-order message-passing architectures in distinguishing topological objects?* This question, highlighted in a recent position paper (Papamarkou et al., 2024), is the main focus of this paper.

---

[*]Equal contribution.

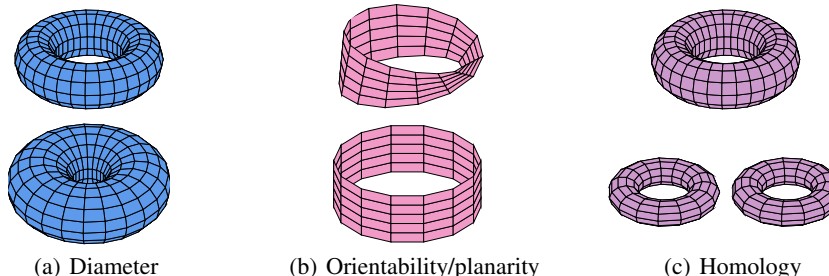

(a) Diameter          (b) Orientability/planarity          (c) Homology

Figure 1: Pairs of HOMP-indistinguishable complexes differing in fundamental metric/topological properties that. In Figure 1(a), tori with different diameters (top 20, bottom 22); in Figure 1(b), a Möbius strip and a cylinder differing in both orientability and planarity; in Figure 1(c), a torus and a pair of disconnected tori which have different homology groups.

We address this question from a topological perspective. First, we introduce a topological criterion designed to identify cases in which a pair of complexes is indistinguishable by HOMP. We then use this criterion to prove HOMP's inability to differentiate between complexes based on several fundamental topological and metric invariants, including diameter, orientability, planarity, and homology groups. These limitations are particularly noteworthy, as TDL's main goal is to leverage topological structure in data. In fact, several methods directly inject information closely related to some of the above properties into pre-existing framewroks (Horn et al., 2021; Chen et al., 2021; Rieck, 2023; Zhang et al., 2023c). Additionally, since many topological data objects are constructed by lifting graph data, we examine HOMP's limitations in expressively leveraging lifting and pooling methods to distinguish graphs.

In the second part of the paper, we introduce a new class of TDL architectures called multi-cellular networks (MCN) designed to address HOMP's expressivity limitations. MCN draws inspiration from higher-order graph architectures (Maron et al., 2019; Morris et al., 2019; Keriven & Peyré, 2019; Azizian & Lelarge, 2020), which successfully resolve expressivity limitations in MPNNs. MCN utilizes the equivariant linear layers introduced in Maron et al. (2018) and integrates them into the HOMP pipeline, resulting in architectures reminiscent of Invariant Graph Networks (IGNs) introduced in the same paper. We prove that MCN can reach *full expressivity* in distinguishing non-isomorphic complexes. Recognizing the scalability challenges of both IGNs and MCN, we propose an alternative called scalable MCN (SMCN). SMCN models apply expressive graph layers – often used as practical alternatives to IGNs – on graph structures defined over the cells of the complex. We prove that SMCN still mitigates many of HOMP's expressivity limitations.

We empirically evaluate SMCN on several real-world (lifted) graph benchmarks and find performance gains over both standard HOMP baselines and expressive GNNs, highlighting the value of expressively leveraging topological information. Additionally, we design three benchmarks to assess TDL architectures' ability to capture topological and metric information. The first, called the Torus Dataset, is a BREC-like (Wang & Zhang, 2024) dataset consisting of pairs of cellular complexes comprising one or more disjoint tori. Models are tasked with separating each pair in a statistically significant way. The two other benchmarks evaluate models based on their ability to predict topological properties of complexes obtained by lifting molecular graphs from ZINC (Sterling & Irwin, 2015).

**Our contributions.** Summarizing, the key contributions of this paper are as follows: (1) We provide a comprehensive analysis of HOMP's expressive power, evaluating its ability to capture topological and metric invariant and leverage lifting and pooling methods. (2) We introduce multi-cellular networks (MCN), a novel class of TDL models, inspired by IGNs, which can provably reach *full expressivity*. (3) We develop SMCN, a scalable version of MCN that addresses HOMP's expressivity limitations while maintaining computational efficiency. (4) We construct three benchmarks for assessing the topological expressivity of TDL architectures. (5) We empirically evaluate the performance of SMCN, demonstrating improvements over both standard TDL methods and expressive graph models, highlighting the benefits of *expressively* leveraging topological information.

## 2 PREVIOUS WORK

**Topological Deep Learning.** TDL architectures enable learning from data supported on topological objects, traditionally focusing on four domains: hypergraphs, simplicial complexes, cellular complexes, and combinatorial complexes. The prominent framework for the latter three is higher-order

message-passing (HOMP). Originally introduced for simplicial complexes (Bodnar et al., 2021b) and later extended to cellular (Bodnar et al., 2021a; Hajij et al., 2020) and combinatorial complexes (Hajij et al., 2022a;b), HOMP architectures achieve strong experimental results and have been shown to enhance the expressive power of MPNNs. Another approach in TDL research incorporates pre-computed topological information into existing models like MPNNs (Horn et al., 2021; Chen et al., 2021; Rieck, 2023) and HOMP (Verma et al., 2024; Buffelli et al., 2024). These methods enhance the expressive power of the base models and show strong experimental results, highlighting the value of integrating topological features. Most prior work is focused on the expressivity of TDL models with respect to *graphs*, the ability of HOMP to capture topological and metric invariants of complexes without relying on pre-computation remains unexplored.

**Expressive power of GNNs.** The expressivity of GNNs is typically evaluated in terms of their separation power, i.e. their ability to assign distinct values to non-isomorphic graphs. Seminal works by Morris et al. (2019) and Xu et al. (2018) demonstrate that the expressive power of MPNNs is equivalent to that of the 1-WL test (Weisfeiler & Leman, 1968). These findings inspired the development of more expressive GNNs, with expressive power surpassing that of the 1-WL test, albeit often requiring greater computational resources. Morris et al. (2019) and Maron et al. (2018) propose architectures that are as expressive as the $k$-WL test with $\mathcal{O}(n^k)$ runtime and memory complexity. Various other expressive GNNs have been introduced in the literature, utilizing techniques such as random features (Abboud et al., 2020), substructure counts (Bouritsas et al., 2022), equivariant polynomials (Maron et al., 2019; Puny et al., 2023), processing sets of subgraphs (Bevilacqua et al., 2021; Frasca et al., 2022; Zhang et al., 2023b; Zhang & Li, 2021; Cotta et al., 2021) and more. We use these frameworks, specifically the architectures proposed in Maron et al. (2019) and Zhang et al. (2023b); Bar-Shalom et al. (2024) to construct efficient and expressive models for combinatorial complexes. Bamberger (2022) offers a perspective on MPNN expressivity through the lens of graph coverings. We extend their work to combinatorial complexes and use it to analyze the topological expressivity of HOMP. For a comprehensive review of expressive graph architectures, refer to the following surveys: Jegelka (2022), Morris et al. (2023); Zhang et al. (2023a).

## 3 PRELIMINARIES

**Notation.** We denote $[n] = \{1, \ldots, n\}$. The size of a set $\mathcal{S}$ is denoted by $|\mathcal{S}|$. $\bigoplus$ and $\bigotimes$ denote aggregation functions, where $\bigoplus$ is *permutation invariant*. Bold lowercase letters denote tuples of integers e.g. $\boldsymbol{k} = (k_0, \ldots, k_\ell)$. $\boldsymbol{e}_i$ denotes the tuple with one at the $i$-th position and zeros elsewhere.

**Combinatorial complexes.** Combinatorial complexes (CCs) are a class of higher-order objects that can flexibly represent many types of hierarchical data. Most topological data domains, including simplicial complexes, cellular complexes, and hypergraphs, can be considered subclasses of combinatorial complexes. Therefore, throughout the paper, all data objects are represented as CCs.

**Definition 3.1** (Combinatorial complex). A combinatorial complex (CC) is a 3-tuple $(S, \mathcal{X}, \mathrm{rk})$ comprising a node set $S$, a cell set $\mathcal{X} \subseteq \mathcal{P}(S) \setminus \emptyset$, and a rank function $\mathrm{rk} : \mathcal{X} \to \mathbb{Z}_{\geq 0}$ such that $\forall s \in S, \{s\} \in \mathcal{X}, \mathrm{rk}(\{s\}) = 0$, and $\forall x, y \in \mathcal{X} \ x \subseteq y \Rightarrow \mathrm{rk}(x) \leq \mathrm{rk}(y)$.

The set of $r$-rank cells ($r$-cells) is called the $r$-skeleton and is denoted by $\mathcal{X}_r = \mathrm{rk}^{-1}(r)$, its size is denoted by $n_r := |\mathcal{X}_r|$; the dimension of a CC is $\ell = \max_{x \in \mathcal{X}} \mathrm{rk}(x)$. We often simplify the notation and use $\mathcal{X}$ to denote the entire CC. For definitions of simplicial and cellular complexes, we refer the reader to Bodnar et al. (2021a) and Bodnar et al. (2021b).

**Neighborhood functions.** Neighborhood functions are a key component in HOMP, facilitating dynamic aggregation of information across cells. Formally, a neighborhood function can be any function $\mathcal{N} : \mathcal{X} \to \mathcal{P}(\mathcal{X})$, but the most common neighborhood functions are

    (1) The $(r_1, r_2)$-adjacency and co-adjacency, defined by

$$\mathcal{A}_{r_1,r_2}(x) = \{y \in \mathcal{X}_{r_1} \mid \exists z \in \mathcal{X}_{r_2} \text{ s.t. } x, y \subseteq z\},$$
$$\mathrm{co}\mathcal{A}_{r_1,r_2}(x) = \{y \in \mathcal{X}_{r_1} \mid \exists z \in \mathcal{X}_{r_2} \text{ s.t. } z \subseteq x, y\}, \tag{1}$$

        for $x \in \mathcal{X}_{r_1}$, and $\mathcal{A}_{r_1,r_2}(x) = \mathrm{co}\mathcal{A}_{r_1,r_2}(x) = \emptyset$ for $x \notin \mathcal{X}_{r_1}$.

    (2) The $(r_1, r_2)$-upper and lower incidence, defined by

$$\mathcal{B}_{r_1,r_2}(x) = \{y \in \mathcal{X}_{r_2} \mid x \subseteq y\}, \quad \mathcal{B}_{r_1,r_2}^\top(x) = \{y \in \mathcal{X}_{r_2} \mid y \subseteq x\}, \tag{2}$$

for $x \in \mathcal{X}_{r_1}$, and $\mathcal{B}_{r_1,r_2}(x) = \mathcal{B}_{r_1,r_2}^{\top}(x) = \emptyset$ for $x \notin \mathcal{X}_{r_1}$.

We call the neighborhood functions defined above *natural neighborhood functions*, a collection we denote by $\mathcal{N}_{\mathrm{nat}}$. See Appendix A.2 for an illustration. Given an enumeration of the cells, a neighborhood function can be represented in matrix form. For example, given a graph $\mathcal{G} = (\mathcal{V}, \mathcal{E})$ viewed as a one dimensional CC through $S = \mathcal{V}$, $\mathcal{X}_0 = \{\{v\} \mid v \in \mathcal{V}\}$ and $\mathcal{X}_1 = \mathcal{E}$, the matrix forms of the neighborhood functions $\mathcal{A}_{0,1}$ and $\mathcal{B}_{0,1}$ are the graph adjacency and incidence matrices respectively.

**Cochain spaces.** Data defined over an $\ell$-dimensional CC can be viewed as a collection of functions $\{\mathbf{h}_r : \mathcal{X}_r \to \mathbb{R}^{d_r}\}_{r=0}^{\ell}$ [1]. Each of these functions is called a *cochain* or a *cell feature map*. The vector space of all cochains over cells of rank $r$ is denoted by $\mathcal{C}^r(\mathcal{X}, \mathbb{R}^{d_r})$ or $\mathcal{C}^r$. The feature associated with a cell $x \in \mathcal{X}_r$ is denoted by $\mathbf{h}_r(x)$, $(\mathbf{h}_r)_x$, or simply $\mathbf{h}_x$.

**Higher-order message-passing.** Higher-order message passing (HOMP) (Hajij et al., 2022b) is a general computational framework for processing information supported on higher-order domains by exchanging messages across cells. Let $\mathcal{N} = \{\mathcal{N}_1, \ldots, \mathcal{N}_k\}$ be a collection of neighborhood functions; given an initial cochain $\mathbf{h}^{(0)} = \mathbf{h}$, HOMP is recursively defined via the following update rule

$$\mathbf{h}_x^{(t+1)} = \beta \left( \bigotimes_{i=1}^{k} \bigoplus_{y \in \mathcal{N}_i(x)} \mathsf{MLP}_{i,\mathrm{rk}(x)}^{(t)} \left( \mathbf{h}_x^{(t)}, \mathbf{h}_y^{(t)} \right) \right), \quad (3)$$

where $\mathbf{h}_x^{(t)}$ is the feature associated with cell $x \in \mathcal{X}$ at layer $t$, and $\beta$ is a nonlinear activation. In the rest of the paper, we assume $\mathcal{N} \subseteq \mathcal{N}_{\mathrm{nat}}$. Similarly to MPNNs, the HOMP framework encompasses many TDL architectures, including architectures for simplicial complexes Bodnar et al. (2021b), cellular complexes Hajij et al. (2020); Bodnar et al. (2021a); Giusti et al. (2023), and combinatorial complexes Hajij et al. (2022a;b).

Figure 2: HOMP Tensor diagram.

**Tensor diagrams.** Hajij et al. (2022a) introduce tensor diagrams, a DAG notation scheme for navigating the rich space of possible HOMP architectures. Tensor diagrams allow for selective aggregation over different neighborhood functions for different cochain spaces in different layers of the network. The nodes of a tensor diagram represent cochain spaces, and the edges represent neighborhood functions. The signal flows from each level of the diagram to the next via the update rule specified in Equation 3, where aggregation is performed only over neighborhood functions associated with the incoming edges [2]. See Figure 2 for an illustration of a tensor diagram and Hajij et al. (2022b) for an in-depth overview.

## 4 EXPRESSIVITY LIMITATIONS OF HIGHER-ORDER MESSAGE-PASSING

The expressivity of graph models is often evaluated in terms of their ability to assign different values to non-isomorphic graphs. Similarly, we study HOMP's ability to distinguish non-isomorphic CCs [3].

### 4.1 A TOPOLOGICAL CRITERION FOR HOMP-INDISTINGUISHABILITY

The main tool we use throughout this section is a topological HOMP-indistinguishability criterion based of the notion of covering spaces, extending the main result of Bamberger (2022) from graph coverings to combinatorial complex coverings.

**Definition 4.1** (CC covering). $\tilde{\mathcal{X}}$ is said to cover $\mathcal{X}$ if there exists a surjective rank-preserving map $\rho : \tilde{\mathcal{X}} \to \mathcal{X}$ which is a local isomorphism with respect to natural neighborhood functions (i.e. $\rho$ bijectively maps the set $\mathcal{N}(x')$ to $\mathcal{N}(\rho(x'))$ for all $x' \in \tilde{\mathcal{X}}$ and $\mathcal{N} \in \mathcal{N}_{\mathrm{nat}}$).

Examples of CC coverings are depicted in Figures 3, 10, and 11. Explicit constructions of covering CCs can be found in Appendix B. The following theorem shows that complexes sharing a cover

---

[1] Generally, $d_{r_1} \neq d_{r_2}$, e.g. atoms (0-cells) and bonds (1-cells) might have a different number of features.

[2] This is equivalent to setting $\mathsf{MLP}_{i,r}^{(t)} \equiv \mathbf{0}$ for neighborhood functions $\mathcal{N}_i$ that are not associated with an incoming edge. Equation 3 in its full generality corresponds to a fully-connected tensor diagram.

[3] $\mathcal{X}$ and $\mathcal{X}'$ are isomorphic if there exists a bijective mapping $\phi : \mathcal{X} \to \mathcal{X}'$ which is both rank-preserving and inclusion-preserving, see Appendix E for a formal definition.

are indistinguishable by HOMP models. See Appendix B for a proof of Theorem 4.2 and further discussion.

**Theorem 4.2** (HOMP-indistinguishability criterion). *Let $\mathcal{X}$ and $\mathcal{X}'$ be CCs such that $|\mathcal{X}_0| = |\mathcal{X}'_0|$. If there exists a CC $\tilde{\mathcal{X}}$ that covers each of the connected components of both $\mathcal{X}$ and $\mathcal{X}'$, then for every HOMP model $\mathsf{M}$, $\mathsf{M}(\mathcal{X}) = \mathsf{M}(\mathcal{X}')$.*

## 4.2 TOPOLOGICAL AND METRIC LIMITATIONS

Although CCs are combinatorial objects, they give rise to various metric and topological spaces. The shortest path distance with respect to any neighborhood function defines a metric on the cells of the CC. In addition, if the complex is cellular or simplicial, it can be canonically associated with a topological space. The topological/metric properties of these spaces are invariants of the underlying CC. We prove HOMP's inability to distinguish between complexes based on the following common invariants: (1) the diameter, which measures how "spread out" the complex is; (2) orientability, which captures whether a consistent "side" or direction can be defined across the entire space; (3) planarity, which captures whether the complex can be embedded in $\mathbb{R}^2$; (4) the homology groups, which encode the structure of "$d$-dimensional holes" [4].

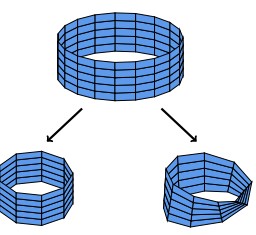

Figure 3: $\mathrm{Cyl}_{h,2p}$ covers both $\mathrm{Cyl}_{h,p}$ and $\mathrm{Möb}_{h,p}$.

**Theorem 4.3** (Topological blindspots). *For any invariant $\boldsymbol{I} \in \{diameter, orientability, planarity, homology\}$ there exists a pair of HOMP-indistinguishable CCs that differ in $\boldsymbol{I}$.*

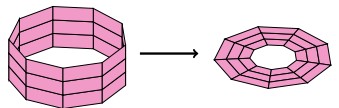

Figure 4: Cylinders are planar.

Figure 1 depicts pairs of HOMP-indistinguishable CCs which differ in each of the above invariants. In addition, Appendix B provides a detailed discussion regarding each invariant, as well as a complete proof of Theorem 4.3. To demonstrate the techniques used in the proof, we include a short proof sketch for the case of orientability and planarity.

*Proof sketch.* Let $\mathrm{Cyl}_{h,p}$ and $\mathrm{Möb}_{h,p}$ be a Cylinder and a Möbius strip with height $h$ and cycle length $p$. The cylinder is orientable and planar, while the Möbius strip is neither (see Figure 4 for an illustration). As illustrated in Figure 3, $\mathrm{Cyl}_{h,2p}$ covers both $\mathrm{Cyl}_{h,p}$ and $\mathrm{Möb}_{h,p}$. Intuitively, $\mathrm{Cyl}_{h,2p}$ covers $\mathrm{Cyl}_{h,p}$ by "wrapping" around it twice, and $\mathrm{Möb}_{h,p}$ by "wrapping" and "twisting" around it (formal construction of both covering maps appears in Appendix B). Since both CCs are connected and have the same number of 0-cells, Theorem 4.2 shows they are HOMP-indistinguishable. □

## 4.3 LIFTING AND POOLING

One benefit of combinatorial complexes is their flexibility in incorporating lifting and pooling[5] methods to construct CCs from graphs. Common graph lifting methods, such as the ones in Bodnar et al. (2021a;b), add meaningful substructures (that standard message-passing cannot detect) as higher-order cells. This results in models that are strictly more expressive than MPNNs. Graph pooling methods, like spectral pooling Ma et al. (2019), Mapper Singh et al. (2007); Hajij et al. (2018); Dey et al. (2016), and DiffPool Ying et al. (2018), coarsen input graphs to enable more efficient learning.

A common feature of many lifting and pooling methods is their ability to generate CCs with a small number of high-order cells that differ in fundamental topological and metric invariants. The sparsity of these cells allows for efficient computation of these invariants, resulting in an efficient way to distinguish the original graphs. However, the following proposition, formally stated and proved in Appendix B.3, suggests that HOMP may still struggle to differentiate between the resulting CCs.

**Proposition 4.4.** *There exist pairs of CCs – generated by standard lifting and pooling methods on graphs (See Figures 10,11 in Appendix B.3) – that HOMP fails to distinguish, even though they differ in basic topological/metric properties. These properties can be efficiently computed due to the sparsity of higher-order cells.*

---

[4]E.g. a circle has a single 1-dimensional hole, a sphere has a single 2-dimensional hole, etc.

[5]In its broadest sense, the term "lifting" includes both substructure lifting and pooling methods. However, as these concepts are often discussed separately in the literature, we distinguish between them, referring to substructure lifting as "lifting" and pooling-based lifting as "pooling".

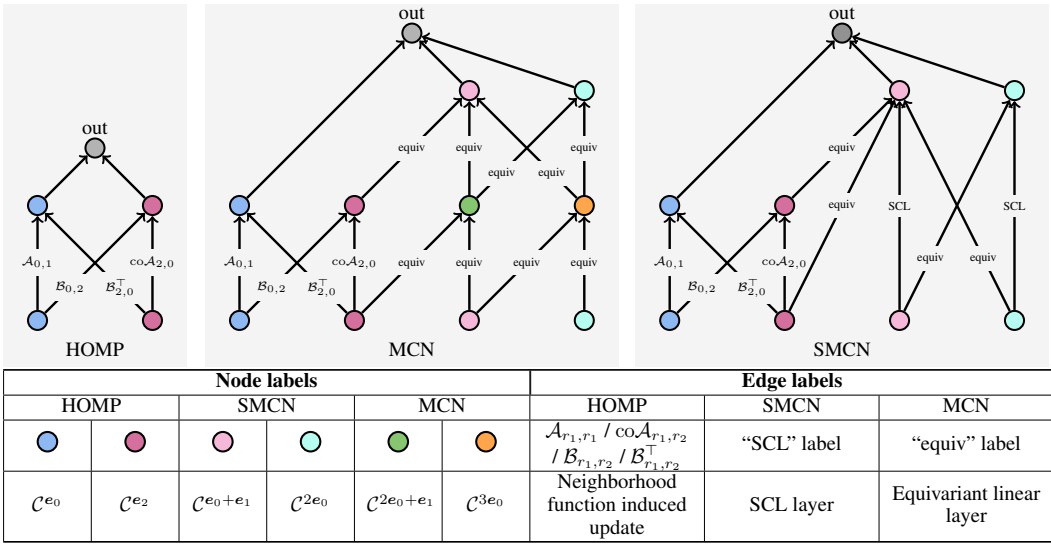

| Node labels | | | | | | Edge labels | | |
|---|---|---|---|---|---|---|---|---|
| **HOMP** | | **SMCN** | | **MCN** | | **HOMP** | **SMCN** | **MCN** |
| 🔵 | 🔴 | 🩷 | 🩵 | 🟢 | 🟠 | $\mathcal{A}_{r_1,r_1}$ / $\mathrm{co}\mathcal{A}_{r_1,r_2}$ / $\mathcal{B}_{r_1,r_2}$ / $\mathcal{B}^\top_{r_1,r_2}$ | "SCL" label | "equiv" label |
| $\mathcal{C}^{e_0}$ | $\mathcal{C}^{e_2}$ | $\mathcal{C}^{e_0+e_1}$ | $\mathcal{C}^{2e_0}$ | $\mathcal{C}^{2e_0+e_1}$ | $\mathcal{C}^{3e_0}$ | Neighborhood function induced update | SCL layer | Equivariant linear layer |

Figure 5: Example tensor diagrams for HOMP, MCN, and SMCN. HOMP only uses nodes labeled with standard cochain spaces. MCN adds nodes labeled with multi-cellular cochain spaces and edges labeled with "equiv" updates. SMCN introduces edges labeled with "SCL". Note that the highest order nodes (green and orange) which can only appear in the MCN diagram are replaced in the SMCN diagram through "SCL" updates of lower order nodes (pink and light blue).

## 5 MULTI-CELLULAR NETWORKS

In graph learning, expressivity limitations similar to those shown in Section 4 have been mitigated by architectures that process features defined over tuples of nodes, and in particular by IGNs (Maron et al., 2018; 2019; Keriven & Peyré, 2019; Azizian & Lelarge, 2020). These models are built by stacking equivariant linear layers between high-order tensor spaces defined over the nodes of the graph, interleaved with pointwise non-linearities. We use a similar approach, introducing multi-cellular cochain spaces (as an analogue to IGN tensor spaces) and incorporating their induced equivariant linear updates into the HOMP framework. For an overview of IGNs, see Appendix A.1.

**Multi-cellular cochain spaces.** Given an $\ell$-dimensional CC $\mathcal{X}$ and an $(\ell+1)$-tuple $\boldsymbol{k} \in \mathbb{N}^{\ell+1}$, a $\boldsymbol{k}$-order multi-cellular cochain is a function $\mathbf{h}_{\boldsymbol{k}} : \mathcal{X}_0^{k_0} \times \cdots \times \mathcal{X}_\ell^{k_\ell} \to \mathbb{R}^d$. The vector space of multi-cellular cochains, denoted by $\mathcal{C}^{\boldsymbol{k}}(\mathcal{X}, \mathbb{R}^d)$ or $\mathcal{C}^{\boldsymbol{k}}$, is called a multi-cellular cochain space. Multi-cellular cochain spaces are a natural generalization of standard cochain spaces, providing a way to represent other types of CC data. E.g. (1) $\mathcal{C}^{e_i} \cong \mathcal{C}^i$; (2) $\mathcal{B}_{r_1,r_2}$ can be represented as a multi-cellular cochain $\in \mathcal{C}^{e_{r_1}+e_{r_2}}$; (3) $(\mathrm{co})\mathcal{A}_{r_1,r_2}$ can be represented as a multi-cellular cochain $\in \mathcal{C}^{2e_{r_1}}$ (see appendix C for details). Moreover, multi-cellular cochain spaces recover many linear spaces studied in several previous works. For example, $\mathcal{C}^{k e_r}$ matches the features space of a $k$-IGN layer operating on $r$-cells, and $\mathcal{C}^{e_{r_1}+e_{r_2}}$ corresponds to the input space of the exchangeable matrix layers introduced in Hartford et al. (2018).

**Symmetry group.** Given enumerations of the sets $\mathcal{X}_0 = \{x_1^0, \ldots, x_{n_0}^0\}, \ldots, \mathcal{X}_\ell = \{x_1^\ell, \ldots, x_{n_\ell}^\ell\}$, a multi-cellular cochain $\mathbf{h} \in \mathcal{C}^{\boldsymbol{k}}(\mathcal{X}, \mathbb{R}^d)$ can be identified with a tensor $\mathbf{A_h}$ defined by

$$(\mathbf{A_h})_{\boldsymbol{i}_0,\ldots,\boldsymbol{i}_\ell,:} = \mathbf{h}\big(x^0_{(\boldsymbol{i}_0)_1}, \ldots, x^0_{(\boldsymbol{i}_0)_{k_0}}, \ldots, x^\ell_{(\boldsymbol{i}_\ell)_1}, \ldots, x^\ell_{(\boldsymbol{i}_\ell)_{k_\ell}}\big) \tag{4}$$

for multi-indices $\boldsymbol{i}_1 \in \{1, \ldots, n_0\}^{k_0}, \ldots, \boldsymbol{i}_\ell \in \{1, \ldots, n_\ell\}^{k_\ell}$. Therefore, $\mathcal{C}^{\boldsymbol{k}}$ can be identified with the tensor space $\mathbb{R}^{n_0^{k_0} \times \cdots \times n_\ell^{k_\ell} \times d}$. The group $G = S_{n_0} \times \cdots \times S_{n_\ell}$ acts on $\mathbf{h} \in \mathcal{C}^{\boldsymbol{k}}$ by

$$(\boldsymbol{\sigma} \cdot \mathbf{h})(\boldsymbol{x}^0, \ldots, \boldsymbol{x}^\ell) = ((\sigma_0, \ldots, \sigma_\ell) \cdot \mathbf{h})(\boldsymbol{x}^0, \ldots, \boldsymbol{x}^\ell) = \mathbf{h}(\sigma_0 \cdot \boldsymbol{x}^0, \ldots, \sigma_\ell \cdot \boldsymbol{x}^\ell), \tag{5}$$

where if $\boldsymbol{x}^r = (x^r_{j_1}, \ldots, x^r_{j_{k_r}}) \in \mathcal{X}_r^{k_r}$ is a tuple of cells, $\sigma_r \cdot \boldsymbol{x}^r = (x^r_{\sigma_r^{-1}(j_1)}, \ldots, x^r_{\sigma_r^{-1}(j_{k_r})})$. In simple terms, the group $G$ acts on $\mathbf{h}$ by reordering the cells of each rank independently. Therefore, to ensure independence of cell ordering, we aim to construct $G$-invariant architectures.

**Equivariant updates.** Since the space $\mathcal{C}^{\boldsymbol{k}}$ can be identified with $\mathbb{R}^{n_0^{k_0} \times \cdots \times n_\ell^{k_\ell} \times d}$, we utilize the basis of equivariant linear layers $\mathbb{R}^{n_0^{k_0} \times \cdots \times n_\ell^{k_\ell} \times d} \to \mathbb{R}^{n_0^{k_0'} \times \cdots \times n_\ell^{k_\ell'} \times d'}$, constructed in Maron et al. (2018), to describe the space of equivariant linear maps $\mathcal{C}^{\boldsymbol{k}} \to \mathcal{C}^{\boldsymbol{k}'}$. Using this basis, denoted $\{L_\gamma\}_{\gamma \in \Gamma(\boldsymbol{k}, \boldsymbol{k}', d, d')}$ (Here $\Gamma(\boldsymbol{k}, \boldsymbol{k}', d, d')$ is an index set defined in Appendix C), we follow the construction of well-known permutation-invariant architectures (Maron et al., 2018; Zaheer et al., 2017; Hartford et al., 2018; Bronstein et al., 2021), and define learnable equivariant layers

$$F(\mathbf{h}) = \beta \left( \sum_{\gamma \in \Gamma(\boldsymbol{k}, \boldsymbol{k}', d, d')} w_\gamma L_\gamma(\mathbf{A_h}) \right), \tag{6}$$

where $\{w_\gamma\}_{\gamma \in \Gamma(\boldsymbol{k}, \boldsymbol{k}', d, d')}$ are learnable parameters and $\beta$ is a non-linearity.

**MCN.** We incorporate equivariant layers to the HOMP framework by adding new node and edge labels to the tensor diagram scheme (as depicted in Figure 5), defining a new class of TDL architectures we call multi-cellular networks (MCNs). At each layer of an MCN tensor diagram, if $v$ is a node labeled by $\mathcal{C}^{\boldsymbol{k}}$, we compute a multi-cellular cochain $\mathbf{h}^{(v)} \in \mathcal{C}^{\boldsymbol{k}}$ by $\mathbf{h}_{\boldsymbol{x}}^{(v)} = \bigotimes_{u \in \mathrm{pred}(v)} \boldsymbol{m}_{u,v}(\boldsymbol{x})$, where $\mathrm{pred}(v)$ denotes the set of predecessor nodes in the diagram, and messages $\boldsymbol{m}_{u,v} \in \mathcal{C}^{\boldsymbol{k}}$ are computed based on the label of the edge $(u, v)$. If the edge is labeled with "equiv", the message is computed as described in Equation 6. For edges labeled by neighborhood functions, the message follows the standard tensor diagram update rule. A formal definition of the MCN framework can be found in Appendix C. Using nodes labeled by higher-order multi-cellular cochain spaces and equivariant updates improves expressivity. In fact, by using multi-cellular cochain spaces of high order, MCN can reach full expressivity, as proved in Appendix E.

**Proposition 5.1** (MCN is fully expressive). *If $\mathcal{X}$ and $\mathcal{X}'$ are non-isomorphic CCs, there exists an MCN model* $\mathsf{M}$ *such that* $\mathsf{M}(\mathcal{X}) \neq \mathsf{M}(\mathcal{X}')$.

## 6 SCALABLE MULTI-CELLULAR NETWORKS

Despite its strong expressive power, implementing MCN in full generality is impractical as the computational complexity and the size of the basis $|\Gamma(\boldsymbol{k}, \boldsymbol{k}', d, d')|$ grow exponentially with $\boldsymbol{k}$ and $\boldsymbol{k}'$. In this section, we design scalable MCN (SMCN), a more efficient version of MCN that still mitigates many of HOMP's expressivity limitations. Below is an overview and motivation for the SMCN architecture. A formal construction and computational runtime analysis can be found in Appendix D; an implementation guide and empirical runtime evaluation are provided in Appendix G.

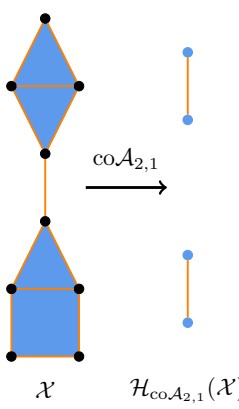

$$\mathrm{co}\mathcal{A}_{2,1}$$

$$\mathcal{X} \qquad \mathcal{H}_{\mathrm{co}\mathcal{A}_{2,1}}(\mathcal{X})$$

Figure 6: $\mathcal{H}_{\mathrm{co}\mathcal{A}_{2,1}}$.

First, we restrict SMCN to multi-cellular cochain spaces $\mathcal{C}^{\boldsymbol{k}}$ with $\sum_{i=0}^{\ell} k_i \leq 3$. Of these, the spaces whose updates incur the heaviest computational overhead are $\mathcal{C}^{3\boldsymbol{e}_r}$ and $\mathcal{C}^{2\boldsymbol{e}_{r_1} + \boldsymbol{e}_{r_2}}$. We replace the updates induced by these spaces with new updates inspired by expressive GNNs, providing a middle ground between expressive power and scalability. These GNNs are applied to graph structures, called augmented Hasse graphs, that capture relational information between cells.

**Definition 6.1** (Augmented Hasse graph). The augmented Hasse graph of $\mathcal{X}$ w.r.t. $(\mathrm{co})\mathcal{A}_{r_1, r_2}$ [6], is defined by $\mathcal{H}_{(\mathrm{co})\mathcal{A}_{r_1, r_2}} = (\mathcal{V}, \mathcal{E})$, where $\mathcal{V} = \mathcal{X}_{r_1}$ and $\mathcal{E} = \{(x, y) \mid y \in (\mathrm{co})\mathcal{A}_{r_1, r_2}(x)\}$.

See Figure 6 for an illustration of an augmented Hasse graph.

**Replacing $\mathcal{C}^{2\boldsymbol{e}_{r_1} + \boldsymbol{e}_{r_2}}$ with $\mathcal{C}^{\boldsymbol{e}_{r_1} + \boldsymbol{e}_{r_2}}$.** Recall that $\mathcal{C}^{2\boldsymbol{e}_{r_1} + \boldsymbol{e}_{r_2}}$ can be identified with $\mathbb{R}^{n_{r_1}^2 \times n_{r_2} \times d}$. Under the action of $S_{n_{r_1}} \times S_{n_{r_2}}$, a tensor $\mathbf{H} \in \mathbb{R}^{n_{r_1}^2 \times n_{r_2} \times d}$ can be viewed as a bag of tensors $\{\mathbf{H}_k \in \mathbb{R}^{n_{r_1} \times n_{r_1} \times d}\}_{k \in [n_{r_2}]}$, each of which is considered up-to $S_{n_{r_1}}$ permutations. These are the exact objects processed by Subgraph GNNs (Bevilacqua et al., 2021; Frasca et al., 2022; Zhang et al., 2023b), which operate on a set of adjacency matrices corresponding to different subgraphs defined

---

[6]$(\mathrm{co})\mathcal{A}_{r_1, r_2}$ represents either an adjacency or a co adjacency neighborhood function.

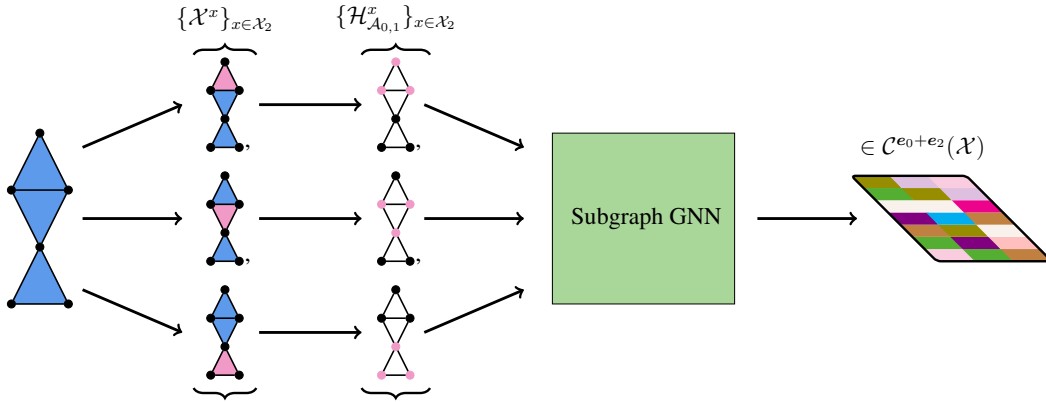

Figure 7: Illustration of an SCL update on $\mathcal{C}^{e_0+e_2}$. We construct a bag of copies of $\mathcal{X}$ with marked 2-cells. This bag is processed by applying a subgraph GNN to the bag of the corresponding marked $\mathcal{A}_{0,1}$ augmented Hasse graphs. Here, $\mathcal{X}^x$ represents the complex $\mathcal{X}$ with a distinct "marking" feature added to the cell features of cell $x$. Similarly, $\mathcal{H}^x_{\mathcal{A}_{0,1}}$ denotes the Augmented Hasse graph $\mathcal{H}_{\mathcal{A}_{0,1}}$ with a marking added to the nodes corresponding to the elements of cell $x$.

over a fixed set of nodes. Subgraph GNNs are strictly more expressive than MPNNs, demonstrate strong experimental performance, and have quadratic runtime complexity as opposed to $\mathcal{O}(n^2_{r_1} \cdot n_{r_2})$ for $\mathcal{C}^{2e_{r_1}+e_{r_2}} \to \mathcal{C}^{2e_{r_1}+e_{r_2}}$ equivariant updates.

Following the above discussion, we define the subcomplex layer (SCL) updates. Let $(u, v)$ be tensor diagram edge labeled by "SCL", connecting layers $t$ and $t+1$ (in which case $v$ and $u$ are both labeled by $\mathcal{C}^{e_{r_1}+e_{r_2}}$). The message $\boldsymbol{m}_{u,v} \in \mathcal{C}^{e_{r_1}+e_{r_2}}$ is computed by

$$\boldsymbol{m}_{u,v}(x,y) = \bigotimes_{r=0,r'=0}^{\ell} \mathsf{MLP}^{r,r'}\left(\mathbf{h}^{(t)}_{x,y}, \mathbf{h}^{(t)}_{(\mathrm{co})\mathcal{A}_{r_1,r}(x),y}, \mathbf{h}^{(t)}_{x,(\mathrm{co})\mathcal{A}_{r_2,r'}(y)}, \mathbf{h}_{x,\mathcal{B}_{r_1,r_2}(x)}, \mathbf{h}_{\mathcal{B}^{\top}_{r_2,r_1}(y),y}\right),$$

where $\mathbf{h}_{\mathcal{Q}_1,y} := \sum_{x' \in \mathcal{Q}_1} \mathbf{h}_{x',y}$ and $\mathbf{h}_{x,\mathcal{Q}_2} := \sum_{y' \in \mathcal{Q}_2} \mathbf{h}_{x,y'}$. This update can be viewed as applying a subgraph update to bags of augmented Hasse graphs, as shown in Figure 7. Additional details are provided in Appendix D. For a review of subgraph networks, see Appendix A.1.

**Replacing $\mathcal{C}^{3e_r}$ with $\mathcal{C}^{2e_r}$.** Since $\mathcal{C}^{3e_r}$ can be identified with $\mathbb{R}^{n^3_r \times d}$, equivariant linear layers of the form $L : \mathcal{C}^{3e_r} \to \mathcal{C}^{3e_r}$ can be identified with 3-IGN layers acting on augmented Hasse graphs of the form $\mathcal{H}_{(\mathrm{co})\mathcal{A}_{r,r'}}$. The GNN literature offers several candidates for efficient 3-IGN substitutes. The first option we considered is PPGN (Maron et al., 2019), which matches 3-IGN's 3-WL expressive power with a runtime of $\mathcal{O}(|\mathcal{V}|^{2 \cdot s})$. Another option is using subgraph networks with node marking, which results in the SCL update rule above with $r_1 = r_2 = r$. These networks have a runtime of $\mathcal{O}(|\mathcal{V}| \cdot |\mathcal{E}|)$ and are strictly more expressive than MPNNs ($> 2$-WL), but are less powerful than 3-IGNs (Frasca et al., 2022). We experimented with both versions and found no significant performance improvement using PPGN. Therefore, we continue with the subgraph version, but note that – since PPGN can implement subgraph networks – all theoretical results hold for the PPGN case as well.

### 6.1 EXPRESSIVE POWER OF SCALABLE MULTI-CELLULAR NETWORKS

**Topological and metric properties.** In Section 4.2, we proved that HOMP models cannot distinguish CCs based on diameter, orientability, planarity or homology. We now examine SMCN with respect to each of these limitations. First, SMCN fully mitigates HOMP's inability to compute diameters.

**Proposition 6.2.** *Any pair of CCs with different diameters can be distinguished by an SMCN model.*

Next, SMCN can distinguish between a cylinder and a Möbius strip, implying that it is strictly better than HOMP at detecting planarity and orientability.

**Proposition 6.3.** *There exists an SMCN model that separates the Möbius strip and the cylinder.*

Finally, we offer two results demonstrating SMCN's ability to distinguish CCs based on their homology groups. The first result examines the 0-th homology group.

**Proposition 6.4.** *Any pair of CCs with distinct $0$-th homology groups can be distinguished by SMCN.*

The second result generalizes to homology groups of any order in the case of two-dimensional surfaces embedded in $\mathbb{R}^3$.

**Proposition 6.5.** *Let $\mathcal{X}, \mathcal{X}'$ be a pair of CCs whose underlying topology corresponds to a 2-dimensional surface (with or without boundary) embeddable in $\mathbb{R}^3$. If $\mathcal{X}$ and $\mathcal{X}'$ differ in any homology group, there exists an SMCN model that distinguishes them.*

A full exploration of SMCN's ability to capture homology groups of any order, orientability and planarity is left for future work. Collectively, Propositions 6.2 – 6.5 suggest that SMCN is strictly better than HOMP at leveraging topological properties of CCs. Rigorous formulations and proofs of these propositions appear in Appendix F. The following is a proof sketch for Proposition 6.3.

*Proof sketch of Proposition 6.3.* The key to distinguishing $\mathrm{Cyl}_{h,p}$ and $\mathrm{M\ddot{o}b}_{h,p}$ is their boundary 1-cells. In our case, these are the 1-cells that are incident to exactly one 2-cell, i.e. their $\mathcal{B}_{1,2}$-degree is 1, so they can easily be detected by an SMCN model. As illustrated in Figure 8, the boundary of $\mathrm{Cyl}_{h,p}$ forms two cycles of length $p$ while the boundary of $\mathrm{M\ddot{o}b}_{h,p}$ forms a single cycle of length $2p$. This is a standard example of a pair of graphs that are indistinguishable by MPNNs but are distinguishable by expressive graph models such as subgraph networks. We can use an SCL update to simulate a subgraph network on the boundaries, separating the two CCs. $\square$

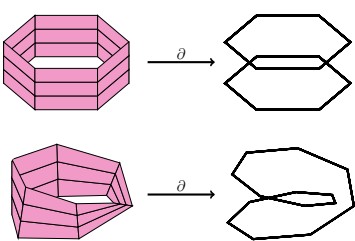

Figure 8: Boundary 1-cells.

**Lifting and Pooling.** In Section 4.3, we discuss HOMP's inability to expressively leverage the sparse higher-order cells generated by common lifting and pooling methods. The next proposition, proved in Appendix F.2, suggests that SMCN is able to leverage this information to a greater extent.

**Proposition 6.6.** *There exist CCs, generated from graphs by standard lifting and pooling methods, that HOMP cannot distinguish but SMCN can. The SMCN model can be constructed to have runtime $\mathcal{O}(m_{\deg} \cdot n_0 \cdot n_2)$, where $m_{\deg}$ is the maximal degree w.r.t. any natural neighborhood function.*

## 7 EXPERIMENTS

The lack of CC benchmarks has been recognized as a challenge in TDL (Papamarkou et al., 2024). To address this, we introduce three novel CC benchmarks designed to assess the ability of TDL models to capture topological/metric properties, and evaluate both SMCN and other HOMP architectures on them. In addition, we adopt the setup of Bodnar et al. (2021a), applying cyclic lifting on real-world graph benchmarks. For an in-depth discussion of experimental details, see Appendix G. A comparison of the expressive power of SMCN and all baselines is available in Appendix F.3.

**Torus dataset.** The torus dataset consists of pairs of CCs, each comprising one or more disjoint tori (see Definition B.4). These pairs are chosen to be HOMP-indistinguishable, despite differing in basic metric/topological properties: they either have distinct homology groups, or they differ in the diameters of some of the components. Models are evaluated by counting the number of pairs they can to separate in a statistically significant way, following the protocols outlined in (Wang & Zhang, 2024). In our experiments, HOMP was unable to distinguish *any* of the pairs, while SMCN was able to distinguish *all* pairs.

**Predicting topological and metric properties.** We construct two additional benchmarks in which models are tasked with predicting topological and metric properties of CCs lifted from ZINC (Sterling & Irwin, 2015) molecular graphs. The predicted properties are the $(0, 1, 2)$-cross-diameter (Equation 24), and the second-order Betti number (rank of the second homology group). In Appendix B we show that HOMP is incapable of fully capturing either property. Table 2 presents both the MSE[7] and the accuracy of predicting the target values (18 possible values for cross-diameter and 6 for Betti numbers) across three TDL models: SMCN, CIN, and a custom HOMP architecture tailored for this prediction task.

---

[7]The targets are normalized to have a standard deviation of 1.

Table 1: SMCN outperforms MPNNs , HOMP and expressive GNNs on graph regression and classification tasks. SMCN results are reported over 5 runs with seed 1-5.

| Model | Reference | ZINC MAE ($\downarrow$) | MOLHIV ROC-AUC ($\uparrow$) | MOLESOL RMSE ($\downarrow$) |
|---|---|---|---|---|
| GCN | Kipf & Welling (2016) | $0.321 \pm 0.009$ | $76.06 \pm 0.97$ | $1.114 \pm 0.036$ |
| GIN | Xu et al. (2018) | $0.163 \pm 0.004$ | $75.58 \pm 1.40$ | $1.173 \pm 0.057$ |
| CIN | Bodnar et al. (2021a) | $0.079 \pm 0.006$ | $80.94 \pm 0.57$ | $1.288 \pm 0.026$ |
| CIN++ | Giusti et al. (2023) | $0.077 \pm 0.004$ | $80.63 \pm 0.94$ | $-$ |
| CIN + CycleNet | Yan et al. (2024) | $0.068$ | $-$ | $-$ |
| Cellular Transformer | Ballester et al. (2024) | $0.080$ | $79.46$ | $-$ |
| PPGN | Maron et al. (2019) | $0.079 \pm 0.005$ | $-$ | $-$ |
| PPGN++ (6) | Puny et al. (2023) | $0.071 \pm 0.001$ | $-$ | $-$ |
| DS-GNN | Bevilacqua et al. (2023) | $0.087 \pm 0.003$ | $76.54 \pm 1.37$ | $0.847 \pm 0.015$ |
| DSS-GNN | Bevilacqua et al. (2021) | $0.102 \pm 0.003$ | $76.78 \pm 1.66$ | $-$ |
| SUN | Frasca et al. (2022) | $0.083 \pm 0.003$ | $80.03 \pm 0.55$ | $-$ |
| GNN-SSWL | Zhang et al. (2023b) | $0.082 \pm 0.003$ | $-$ | $-$ |
| GNN-SSWL+ | Zhang et al. (2023b) | $0.070 \pm 0.005$ | $79.58 \pm 0.35$ | $0.837 \pm 0.019$ |
| Subgraphormer | Bar-Shalom et al. (2023) | $0.067 \pm 0.007$ | $80.38 \pm 1.92$ | $0.832 \pm 0.043$ |
| Subgraphormer + PE | Bar-Shalom et al. (2023) | $0.063 \pm 0.001$ | $79.48 \pm 1.28$ | $0.826 \pm 0.010$ |
| SMCN (ours) | This paper | $\mathbf{0.060 \pm 0.004}$ | $\mathbf{81.16 \pm 0.90}$ | $\mathbf{0.809 \pm 0.037}$ |

The benchmarks detailed above empirically verify SMCN's superior ability to capture topological/metric properties of CCs. This is demonstrated for both synthetically generated data as well lifted molecular graphs, complementing theoretical results from Sections 4 and 6.1, demonstrating that the expressivity gains of SMCN lead to improved learning of topological/metric invariants.

**Real-world graph benchmarks.** We evaluate SMCN on ZINC-12K (Sterling & Irwin, 2015), MOL-HIV, and MOLESOL (Hu et al., 2020). We compare SMCN to several HOMP baselines as well as a range of expressive graph architectures. As seen in Table 1, SMCN outperforms both HOMP architectures and expressive graph methods across all three benchmarks, underscoring the value of expressively leveraging higher-order topological information on graphs.

## 8 CONCLUSION

In the first part of the paper, we analyzed the expressivity limitations of HOMP from a topological perspective, proving its inability to capture the diameter, orientability, planarity, and homology of input CCs.

Table 2: Accuracy and normalized MSE scores of predicting the cross-diameter and the second Betti number of lifted ZINC graphs.

| Model | Cross-diameter Accuracy ($\uparrow$) / MSE ($\downarrow$) | 2nd Betti number Accuracy ($\uparrow$) / MSE ($\downarrow$) |
|---|---|---|
| CIN | $34.78 \pm 3.00\%/0.3421 \pm 0.0691$ | $42.15 \pm 25.22\%/0.3405 \pm 0.3799$ |
| Custom HOMP | $67.87 \pm 12.26\%/0.0684 \pm 0.0323$ | $81.76 \pm 10.06\%/0.0391 \pm 0.0208$ |
| SMCN | $\mathbf{92.76 \pm 0.53\%/0.011 \pm 0.0008}$ | $\mathbf{99.61 \pm 0.12\%/0.0024 \pm 0.0005}$ |

Additionally, we showed that there exist CCs generated through common graph lifting methods which are HOMP-indistinguishable despite differing in easy-to-compute topological invariants. In the second part of the paper, we introduced MCN, inspired by $k$-IGNs, and its more scalable version, SMCN. We proved that, analogously to IGNs, MCN can reach full expressivity for CCs. We additionally showed that SMCN tractably addresses many of HOMP's expressivity limitations. Finally, we presented three novel benchmarks designed to evaluate TDL architectures' ability to capture topological/metric information. We evaluated SMCN on both benchmarks as well as real-world graph benchmarks. SMCN outperformed HOMP architectures on expressivity benchmarks, empirically supporting our theoretical findings. On the real-world graph benchmarks, SMCN outperformed both HOMP architectures and expressive graph architectures, demonstrating the value of expressively leveraging higher-order topological information.

**Limitations and future work.** The components that make SMCN more expressive have runtime that scales super-linearly in the number of cells, making SMCN intractable for larger CCs. Future research may aim to design more scalable alternatives to SMCN. Additionally, although we have shown that SMCN is strictly better than HOMP in distinguishing CCs based on orientability and homology, it remains unclear if it is able to fully capture these properties. Future work can explore SMCN limitations in expressing topological and metric invariants. Finally, future research can aim to develop more complex benchmarks that include a broader range of topological properties.

ACKNOWLEDGMENTS

The authors would like to thank Beatrice Bevilacqua for insightful discussions. YG is supported by the the UKRI Engineering and Physical Sciences Research Council (EPSRC) CDT in Autonomous and Intelligent Machines and Systems (grant reference EP/S024050/1). FF is funded by the Andrew and Erna Finci Viterbi Post-Doctoral Fellowship. FF partly performed this work while visiting the Machine Learning Research Unit at TU Wien led by Prof. Thomas Gärtner. MB is supported by EPSRC Turing AI World-Leading Research Fellowship No. EP/X040062/1 and EPSRC AI Hub on Mathematical Foundations of Intelligence: An "Erlangen Programme" for AI No. EP/Y028872/1. HM is a he Robert J. Shillman Fellow, and is supported by the Israel Science Foundation through a personal grant (ISF 264/23) and an equipment grant (ISF 532/23).

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

## A  BACKGROUND

### A.1  OVERVIEW OF EXPRESSIVE GNN ARCHITECTURES USED IN THE PAPER

The expressivity of GNNs is typically evaluated in terms of their separation power, i.e. their ability to assign distinct values to non-isomorphic graphs. Seminal works by Morris et al. (2019) and Xu et al. (2018) demonstrate that the expressive power of MPNNs is equivalent to that of the 1-WL test (Weisfeiler & Leman, 1968). These findings inspired the development of more expressive GNNs, with expressive power surpassing that of the 1-WL test, albeit often requiring greater computational resources. In this paper we focus on two such architecturres: invariant graph networks (IGNs) (Maron et al., 2018; 2019) and subgraph neural networks Bevilacqua et al. (2021); Frasca et al. (2022); Zhang et al. (2023b).

**Invariant graph networks.**  Maron et al. (2018) proposed a principled approach to design expressive GNN architectures by leveraging the inherent symmetries of graphs.

More specifically, given a graph $G$ with adjacency matrix $\boldsymbol{A} \in \mathbb{R}^{n \times n}$ and node feature matrix $X \in \mathbb{R}^{n \times d}$, an IGN first encodes this graph as a tensor $\mathbf{T} \in \mathbb{R}^{n^2 \times (d+1)}$ where $T_{:,:,1}$ holds the adjacency matrix $A$ and the last $d$ channels hold the node features on their diagonal, i.e. $T_{i,i,j} = X_{i,j}$ and $T_{i_1,i_2,j} = 0$ for $i_1 \neq i_2$. The symmetry group $S_n$ acts naturally on $\mathbb{R}^{n^2 \times (d+1)}$ by:

$$\sigma \cdot \mathbf{T}_{i,j,k} = \mathbf{T}_{\sigma^{-1}(i), \sigma^{-1}(j), k} \quad \sigma \in S_n. \tag{7}$$

Notice that for any graph tensor $\mathbf{T}$ and permutation $\sigma \in S_n$, the tensors $\mathbf{T}$ and $\sigma \cdot \mathbf{T}$ represent the same graph. This action can be easily generalied to the tensor space $\mathbb{R}^{n^k \times c}$ by:

$$\sigma \mathbf{T}_{i_1, \ldots, i_k m j} = \mathbf{T}_{\sigma^{-1}(i_1), \ldots, \sigma^{-1}(i_k), j}. \tag{8}$$

For any integers $k, k', c, c'$ Maron et al. (2018) finds a basis to the space of all linear maps $L : \mathbb{R}^{n^k \times c} \to \mathbb{R}^{n^{k'} \times c'}$ which satisfy

$$L(\sigma \cdot \mathbf{T}) = \sigma \cdot L(\mathbf{T}). \tag{9}$$

These are called equivariant linear maps. a $k$-IGN stacks lyaers is of the form

$$U(T) = \beta\left(\sum_{\gamma \in \Gamma} w_\gamma L_\gamma(\mathbf{T})\right) \tag{10}$$

where $\{L_\gamma\}_{\gamma \in \Gamma}$ is a basis of the space of equivariant layers from $\mathbb{R}^{n^{k_1} \times c_1}$ to $\mathbb{R}^{n^{k_2} \times c_2}$ for some $k_1 1, k_2 \leq k$ and $c_1, c_2 \in \mathbb{N}$, $\{w_\gamma\}_{\gamma \in \Gamma}$ are learnable parameters and $\beta$ is a non-linear activation

function. The k-IGN architecture was proven in Maron et al. (2019) to be as expressive as the $k$-WL test, extending the capabilities of MPNNs, which were shown to possess expressivity equivalent to the 1-WL test. Despite this $l$-IGNs have a runtime complexity of $\mathcal{O}(n^k)$ making them inpractical to use. To address this, other expressive GNNs have been proposed, offering a balance between the computational complexity and expressive power of 3-IGNs and MPNNs. One such family of architectures is subgraph neural networks.

**Subgraph neural networks.** Subgraph neural networks, (You et al., 2021; **?**; Cotta et al., 2021; Bevilacqua et al., 2021), rely on a predefined policy that transforms an input graph into a set of graphs, with each graph in the set representing an augmented version of the original. Some policies include node deletion, where each graph in the set is created by removing a single node from the original graph; $k$-ego policies, where each graph is generated by extracting the $k$-neighborhood of a specific node; and node marking, where each graph is obtained by assigning a unique node feature to a single node in the original graph. Subgraph GNNs then process sets of graphs by independently applying MPNN updates to each graph in the set, while also incorporating cross-graph updates to exchange information between the graphs. Bevilacqua et al. (2021) has shown that subgraph GNNs are strictly more expressive then MPNNs, while Frasca et al. (2022) has shown they are strictly less expressive then 3-IGNS. With a runtime complexity of $\mathcal{O}(d \cdot n^2)$ where $d$ is the maximum degree of the input graph, subgraph GNNs offer a compelling trade-off: they are more expressive than MPNNs while remaining more scalable than 3-IGNs. Subgraph GNNs have demonstrated strong empirical performance in studies such as Bevilacqua et al. (2021), Frasca et al. (2022), and Zhang et al. (2023b), among others, establishing them as a robust and effective choice for GNN architectures. For an in depth discussion on subgraph neural networks see Bronstein (2021).

### A.2 Neighborhood Function Illustration

The following is an example of the standard neighborhood functions introduced in Section 3. For the CC in Figure 9 the following relations hold:

- $\{A\} \in \mathcal{A}_{0,1}(\{B\})$.
- $\{A\} \notin \mathcal{A}_{0,1}(\{D\})$.
- $\{A\} \in \mathcal{A}_{0,2}(\{D\})$.
- $\{C, D\} \in \mathrm{co}\mathcal{A}_{1,0}(\{A, C\})$.
- $\{C, D\} \notin \mathrm{co}\mathcal{A}_{1,0}(\{A, B\})$.
- $\{C, D\} \in \mathcal{A}_{1,2}(\{A, B\})$.
- $\{C, D, E\} \in \mathrm{co}\mathcal{A}_{2,0}(\{E, F, H\})$.
- $\{C, D, E\} \notin \mathrm{co}\mathcal{A}_{2,1}(\{E, F, H\})$.
- $\{A, B, C, D\} \in \mathrm{co}\mathcal{A}_{2,1}(\{C, D, E\})$.
- $\{B, D\} \in \mathcal{B}_{0,1}(\{D\})$.
- $\{F, G, H\} \in \mathcal{B}_{0,2}(\{G\})$.
- $\{B\} \in \mathcal{B}_{1,0}^{\top}(\{B, D\})$.
- $\{B\} \notin \mathcal{B}_{1,0}^{\top}(\{C, D\})$.
- $\{B\} \in \mathcal{B}_{2,0}^{\top}(\{A, B, C, D\})$.
- $\{B\} \notin \mathcal{B}_{2,0}^{\top}(\{C, D, E\})$.

HOMP can be viewed as performing parallel message passing on the connectivity structures defined by these neighborhood functions.

## B Expressivity Limitations of Higher-Order Message-Passing

### B.1 A Topological Criterion for HOMP Indistinguishability

In this section we formally restate and prove Theorem 4.2.

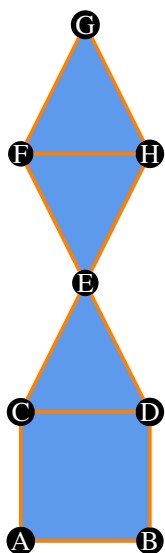

Figure 9: CC example for neighborhood function illustration. Nodes are labeled $A, \ldots, G$, cells are comprised of subsets of nodes, 1-cells are represented by orange edges, and 2-cells are represented as blue faces. E.g. $\{E, F, H\}$ is a 2-cell, $\{A, C\}$ is a 1-cell.

**Theorem B.1** (HOMP-indistinguishability criterion, restatement of Theorem 4.2). *Let $\mathcal{X}$ and $\mathcal{X}'$ be CCs of dimension $\ell$ with no cell features such that $|\mathcal{X}_0| = |\mathcal{X}_0'|$. If $\mathcal{X}$ and $\mathcal{X}'$ admit decompositions into connected components*

$$\mathcal{X} = \bigsqcup_{\mathcal{Z} \in C(\mathcal{X})} \mathcal{Z}, \quad \mathcal{X}' = \bigsqcup_{\mathcal{Z}' \in C(\mathcal{X}')} \mathcal{Z}', \tag{11}$$

*such that $\exists \tilde{\mathcal{X}}$ that is covers each of the connected components $\mathcal{Z} \in C(\mathcal{X}), \mathcal{Z}' \in C(\mathcal{X}')$, then for every HOMP model M, $\mathsf{M}(\mathcal{X}) = \mathsf{M}(\mathcal{X}')$.*

A combinatorial complex $\mathcal{X}$ is said to be *connected* if its Hasse graph, defined by $\mathcal{G} = (\mathcal{V}, \mathcal{E})$ with $\mathcal{V} = \mathcal{X}$ and $\mathcal{E} = \{(x, y) \mid x \subseteq y, \mathrm{rk}(x) = \mathrm{rk}(y) - 1\}$, is connected. To prove Theorem B.1, we first state and prove two lemmas.

**Lemma B.2.** *Let $\rho : \tilde{\mathcal{X}} \to \mathcal{X}$ be a covering map. In addition, let M be a HOMP model with $T$ layers, and let $\mathbf{h}_x^{(t)}$ and $\tilde{\mathbf{h}}_{x'}^{(t)}$ denote the cell feature maps of $\mathcal{X}$ and $\tilde{\mathcal{X}}$ at layer $t$ evaluated on cells $x \in \mathcal{X}$ and $x' \in \tilde{\mathcal{X}}$ respectively. Under these conditions, $\tilde{\mathbf{h}}_{x'}^{(t)} = \mathbf{h}_{\rho(x')}^{(t)}$, for $t = 0, \ldots, T$, $x' \in \tilde{\mathcal{X}}$.*

*Proof.* We use induction on $t$. For $t = 0$, as both CCs have no initial cellular feature maps, HOMP initializes $\mathbf{h}_x^{(0)}, \tilde{\mathbf{h}}_{x'}^{(0)}$ by assigning a constant feature to all cells and the claim holds trivially. Assume the claim holds for some $t \in \{0, \ldots, T\}$. The HOMP update rule reads:

$$\begin{aligned}
\mathbf{h}_x^{(t+1)} &= \beta \left( \bigotimes_{\mathcal{N} \in \boldsymbol{\mathcal{N}}_{\mathrm{nat}}} \bigoplus_{y \in \mathcal{N}(x)} \mathsf{MLP}_{\mathcal{N},\mathrm{rk}(x)}^{(t)}(\mathbf{h}_x^{(t)}, \mathbf{h}_y^{(t)}) \right), \\
\tilde{\mathbf{h}}_{x'}^{(t+1)} &= \beta \left( \bigotimes_{\mathcal{N} \in \boldsymbol{\mathcal{N}}_{\mathrm{nat}}} \bigoplus_{y' \in \mathcal{N}(x')} \mathsf{MLP}_{\mathcal{N},\mathrm{rk}(x')}^{(t)}(\tilde{\mathbf{h}}_{x'}^{(t)}, \tilde{\mathbf{h}}_{y'}^{(t)}) \right).
\end{aligned} \tag{12}$$

Since $\rho$ is a covering map, $\mathcal{N}(x')$ is bijectively mapped to $\mathcal{N}(\rho(x'))$ for every $x' \in \tilde{\mathcal{X}}$ and every neighborhood function $\mathcal{N} \in \boldsymbol{\mathcal{N}}_{\mathrm{nat}}$. Additionally, $\mathrm{rk}(\rho(x')) = \mathrm{rk}(x')$. This, along with the fact that $\bigoplus$ is permutation invariant, and the induction hypothesis implies that:

$$\bigoplus_{y' \in \mathcal{N}(x')} \mathsf{MLP}_{\mathcal{N},\mathrm{rk}(x')}^{(t)}(\tilde{\mathbf{h}}_{x'}^{(t)}, \tilde{\mathbf{h}}_{y'}^{(t)}) = \bigoplus_{y \in \mathcal{N}(\rho(x'))} \mathsf{MLP}_{\mathcal{N},\mathrm{rk}(\rho(x'))}^{(t)}(\mathbf{h}_{\rho(x')}^{(t)}, \mathbf{h}_y^{(t)}). \tag{13}$$

Thus, combining Equation 12 and Equation 13, we get $\tilde{\mathbf{h}}_{x'}^{(t+1)} = \mathbf{h}_{\rho(x')}^{(t+1)}$. □

**Lemma B.3.** *If $\mathcal{X}$ is connected and $\rho : \tilde{\mathcal{X}} \to \mathcal{X}$ is a covering map, $\forall x \in \mathcal{X}, |\rho^{-1}(x)| = \frac{|\tilde{\mathcal{X}}_0|}{|\mathcal{X}_0|}$.*

*Proof.* Since $\rho$ is surjective and rank-preserving, the above is equivalent to $\forall x, y \in \mathcal{X}, |\rho^{-1}(y)| = |\rho^{-1}(x)|$. Since $\mathcal{X}$ is connected, it suffices to show that this equality holds for any $x, y \in \mathcal{X}$ such that $y \in \mathcal{N}(x)$ for some function $\mathcal{N} \in \boldsymbol{\mathcal{N}}_{\text{nat}}$. We first show that for any natural neighborhood function $\mathcal{N} \in \boldsymbol{\mathcal{N}}_{\text{nat}}$ and cell $x \in \mathcal{X}$ the sets $\{\mathcal{N}(x') \mid x' \in \rho^{-1}(x)\}$ are pairwise disjoint. To see this, assume by contradiction that for a pair of cells $x'_1, x'_2 \in \rho^{-1}(x)$ we have $\mathcal{N}(x'_1) \cap \mathcal{N}(x'_2) \neq \emptyset$. If $z' \in \mathcal{N}(x'_1) \cap \mathcal{N}(x'_2)$, then there is a neighborhood function $\mathcal{N}^* \in \boldsymbol{\mathcal{N}}_{\text{nat}}$ such that $x'_1, x'_2 \in \mathcal{N}^*(z')$. Given that $\rho(x'_1) = \rho(x'_2)$, this would imply that $\rho$ is not injective on $\mathcal{N}^*(z')$, contradicting the definition of a covering map. Now, since $y \in \mathcal{N}(x)$, for any $x' \in \rho^{-1}(x)$ there exists a $y' \in \mathcal{N}(x')$ such that $\rho(y') = y$. Since the set $\{\mathcal{N}(x') \mid x' \in \rho^{-1}(x)\}$ is pairwise disjoint this implies that $|\rho^{-1}(y)| \geq |\rho^{-1}(x)|$. Since $y \in \mathcal{N}(x)$, there exists a neighborhood function $\mathcal{N}^* \in \boldsymbol{\mathcal{N}}_{\text{nat}}$ such that $x \in \mathcal{N}^*(y)$, implying by the same reasoning above that $|\rho^{-1}(y)| \leq |\rho^{-1}(x)|$. We thus have $|\rho^{-1}(y)| = |\rho^{-1}(x)|$ which concludes the proof. □

We are now ready to prove Theorem B.1.

*Proof.* Let $\tilde{\mathcal{X}}$ be a combinatorial complex that covers all connected components $\mathcal{Z} \in C(\mathcal{X})$ and $\mathcal{Z}' \in C(\mathcal{X}')$ via maps the maps $\{\rho_{\mathcal{Z}}\}_{\mathcal{Z} \in C(\mathcal{X})}$ and $\{\rho_{\mathcal{Z}'}\}_{\mathcal{Z}' \in C(\mathcal{X}')}$ respectively. Let M be a HOMP model with $T$ layers and let $\mathbf{h}^{(t)}$, $\mathbf{h}'^{(t)}$, and $\tilde{\mathbf{h}}^{(t)}$ denote the cell feature maps of $\mathcal{X}$, $\mathcal{X}'$, and $\tilde{\mathcal{X}}$ respectively at layer $t$. Lemma B.2 implies that for every $\mathcal{Z} \in C(\mathcal{X})$, $\mathcal{Z}' \in C(\mathcal{X}')$ and every $z \in \mathcal{Z}$, $z' \in \mathcal{Z}'$ we have

$$
\begin{aligned}
\mathbf{h}_z^{(T)} &= \tilde{\mathbf{h}}_y^{(T)} \quad \forall y \in \rho_{\mathcal{Z}}^{-1}(z), \\
\mathbf{h}_{z'}'^{(T)} &= \tilde{\mathbf{h}}_y^{(T)} \quad \forall y \in \rho_{\mathcal{Z}'}^{-1}(z').
\end{aligned}
\tag{14}
$$

This implies that the sets of unique values corresponding to the multisets $\{\!\{\mathbf{h}_x^{(T)} \mid x \in \mathcal{X}\}\!\}$, $\{\!\{\mathbf{h}_{x'}'^{(T)} \mid x' \in \mathcal{X}'\}\!\}$ and $\{\!\{\tilde{\mathbf{h}}_y^{(T)} \mid y \in \tilde{\mathcal{X}}\}\!\}$ are the same. Let $n_y, n_y', \tilde{n}_y$ be the number of times the value $\tilde{\mathbf{h}}_y^{(T)}$ appear in the multisets $\{\!\{\mathbf{h}_x^{(T)} \mid x \in \mathcal{X}\}\!\}$, $\{\!\{\mathbf{h}_{x'}'^{(T)} \mid x' \in \mathcal{X}'\}\!\}$ and $\{\!\{\tilde{\mathbf{h}}_y^{(T)} \mid y \in \tilde{\mathcal{X}}\}\!\}$ respectively. Since each $\mathcal{Z}, \mathcal{Z}'$ are connected, we can use Lemma B.3 to get that $\forall z \in \mathcal{Z}, \forall z' \in \mathcal{Z}', |\rho_{\mathcal{Z}}^{-1}(z)| = \frac{|\tilde{\mathcal{X}}_0|}{|\mathcal{Z}_0|}$ and $|\rho_{\mathcal{Z}'}^{-1}(z')| = \frac{|\tilde{\mathcal{X}}_0|}{|\mathcal{Z}'_0|}$. This implies that $\forall y \in \tilde{\mathcal{X}}$

$$
n_y = \tilde{n}_y \cdot \left( \sum_{\mathcal{Z} \in C(\mathcal{X})} \frac{|\mathcal{Z}_0|}{|\tilde{\mathcal{X}}_0|} \right), \quad n_y' = \tilde{n}_y \cdot \left( \sum_{\mathcal{Z}' \in C(\mathcal{X}')} \frac{|\mathcal{Z}'_0|}{|\tilde{\mathcal{X}}_0|} \right).
\tag{15}
$$

Since $\sum_{\mathcal{Z} \in C(\mathcal{X})} |\mathcal{Z}_0| = |\mathcal{X}_0| = |\mathcal{X}'_0| = \sum_{\mathcal{Z}' \in C(\mathcal{X}')} |\mathcal{Z}'_0|$, this implies that $\forall y \in \tilde{\mathcal{X}}, n_y = n_y'$. We have shown the set of unique values corresponding to multisets $\{\!\{\mathbf{h}_x^{(T)} \mid x \in \mathcal{X}\}\!\}$ and $\{\!\{\mathbf{h}_x'^{(T)} \mid x' \in \mathcal{X}'\}\!\}$ is the same, and that the number of times each value appears in the multisets is the same, thus the two multisets are equal. Since the readout of a HOMP model can is a function this multiset, $\mathcal{X}$ and $\mathcal{X}'$ are indistinguishable by HOMP. □

### B.2 Topological and Metric Limitations

In this section, we rigorously state and prove all results regarding HOMP's inability to express topological/metric properties, presented in Section B. We begin by defining the $\ell$-dimensional torus CCs. As we will later see, this class provides us with examples of indistinguishable CCs that differ in both the diameter and all homology groups.

**$\ell$-dimensional torus CCs.** An $\ell$ dimensional torus is a Cartesian product of $\ell$ cycles. More formally:

**Definition B.4** ($\ell$-dimensional torus CCs). For a sequence of integers $p_1, \ldots, p_\ell$, the torus $T_{p_1, \ldots, p_\ell}$ is a combinatorial complex $(S, \mathcal{X}, \text{rk})$ defined by:

$$
S = [p_1] \times \cdots \times [p_\ell],
\tag{16}
$$

$$\mathcal{X}_r = \{s_k \mid s \in S, k \in \{0, 1\}^\ell, k_1 + \cdots + k_\ell = r\}, \tag{17}$$

where $s_k$ is defined by:

$$s_k = \{s + k' \mid k' \in \{0, 1\}^\ell, k' \le k\}. \tag{18}$$

The sum $s + k'$ is coordinate-wise, where at coordinate $j$ result is taken modulo $p_j$, and $k' \le k$ if $k'_j \le k_j, \forall j \in \{1, \ldots, \ell\}$.

By slight abuse of notation, we sometimes refer to the set of cells of the torus by $T_{p_1, \ldots, p_\ell}$ as well. We note that the torus $T_{p_1, \ldots, p_\ell}$ as defined above is only one possible realization of the $\ell$-dimensional torus as a combinatorial complex. An example of a two-dimensional torus can be seen in Figure 1(a). As the next lemma shows, all $\ell$ dimensional tori are locally isometric.

**Lemma B.5.** *Let $T_{p_1, \ldots, p_\ell}$ and $T_{p'_1, \ldots, p'_\ell}$ be two $\ell$-dimensional tori such that $\forall j \in \{1, \ldots, \ell\}$, $p_j, p'_j \ge 3$. The torus $T_{p_1 \cdot p'_1, \ldots, p_\ell \cdot p'_\ell}$ covers both $T_{p_1, \ldots, p_\ell}$ and $T_{p'_1, \ldots, p'_\ell}$.*

*Proof.* Denote $p = (p_1, \ldots, p_\ell)$, $p' = (p'_1, \ldots, p'_\ell)$, $\tilde{p} = (\tilde{p}_1, \ldots, \tilde{p}_\ell) = (p_1 \cdot p'_1, \ldots, p_\ell \cdot p'_\ell)$. Additionally, denote by $S, S', \tilde{S}$, and $\mathcal{X}, \mathcal{X}', \tilde{\mathcal{X}}$ the nodes and cell sets of $T_p, T_{p'}$ and $T_{\tilde{p}}$ respectively. Define $\rho : \tilde{S} \to S, \rho' : \tilde{S} \to S'$ by:

$$\begin{aligned} \rho(\tilde{s}) &= \tilde{s} \mod p, \\ \rho'(\tilde{s}) &= \tilde{s} \mod p', \end{aligned} \tag{19}$$

where $\tilde{s} \mod p := (\tilde{s}_1 \mod p_1, \ldots, \tilde{s}_\ell \mod p_\ell)$. We extend $\rho$ and $\rho'$ to $\tilde{\mathcal{X}}$ by $\rho(x) = \{\rho(s) \mid s \in x\}$. We now prove that $\rho$ is a covering map. We start by showing that $\forall r \in \{0, \ldots \ell\}$, $\rho(\tilde{\mathcal{X}}_r) = \mathcal{X}_r$ (i.e. $\rho$ is rank-preserving). Recall that all elements of $\tilde{\mathcal{X}}_r$ are of the form $\tilde{s}_k$ for some $\tilde{s} \in \tilde{S}$ and $k \in \{0, 1\}^\ell$ such that $k_1 + \cdots + k_\ell = r$. Since $p < \tilde{p}$, for every $k' \le k$:

$$(\tilde{s} + k' \mod \tilde{p}) \mod p = (\tilde{s} \mod p) + (k' \mod p) = \rho(\tilde{s}) + k' \mod p. \tag{20}$$

Therefore,

$$\rho(\tilde{s}_k) = \rho(\tilde{s})_k \in \mathcal{X}_r \tag{21}$$

and $\rho$ is rank-preserving. To show that $\rho$ is a covering map, all that remains is to show that it preserves natural neighborhood functions and that it is surjective. For the former, notice that since $\rho$ is defined on the node set $\tilde{S}$, for every $x, y, z \in \tilde{\mathcal{X}}$ we have:

- $x \subseteq y \Rightarrow \rho(x) \subseteq \rho(y)$.
- $x, y \subseteq z \Rightarrow \rho(x), \rho(y) \subseteq \rho(z)$.
- $z \subseteq x, y \Rightarrow \rho(z) \subseteq \rho(x), \rho(y)$.

Thus, $\rho$ preserves all natural neighborhood functions. Finally, since $p_1, \ldots, p_\ell \ge 3$ it is easy to check that for any $x, y \in \tilde{\mathcal{X}}$ and $\mathcal{N} \in \mathcal{N}_{\mathrm{nat}}$:

$$y \in \mathcal{N}(x) \Rightarrow \rho(x) \ne \rho(y). \tag{22}$$

This implies that $\rho$ is a covering map. An equivalent argument shows that $\rho'$ is also a covering map, completing the proof. $\square$

Lemma B.5 gives rise to the following useful corollary.

**Corollary B.6.** *If $T_{p_1, \ldots, p_\ell}$ and $T_{p'_1, \ldots, p'_\ell}$ are $\ell$-dimensional tori such that $p_1 \cdots p_\ell = p'_1 \cdots p'_\ell$ (i.e. $T_{p_1, \ldots, p_\ell}$ and $T_{p'_1, \ldots, p'_\ell}$ have the same number of 0-cells) and $\forall j \in \{1, \ldots, \ell\}$, $p_j, p'_j \ge 3$, then for every HOMP model $\mathsf{M}$, $\mathsf{M}(T_{p_1, \ldots, p_\ell}) = \mathsf{M}(T_{p'_1, \ldots, p'_\ell})$.*

*Proof.* Both $T_{p_1, \ldots, p_\ell}$ and $T_{p'_1, \ldots, p'_\ell}$ are connected, have the same number of 0-cells ($(T_{p_1, \ldots, p_\ell})_0 = p_1 \cdots p_\ell = p'_1 \cdots p'_\ell = (T_{p'_1, \ldots, p'_\ell})_0$), and are covered by $T_{p_1 \cdot p'_i, \ldots, p_\ell \cdot p'_\ell}$. Therefore, Theorem B.1 implies that $T_{p_1, \ldots, p_\ell}$ and $T_{p'_1, \ldots, p'_\ell}$ are indistinguishable by HOMP. $\square$

Note, that tori with the same number of nodes can still differ on a number of topological and metric properties. In the following we use the family of $\ell$ dimensional tori to produce examples of topologically/metrically distinct CCs that are indistinguishable by HOMP.

**Diameter.** For a given adjacency neighborhood function $(\text{co})\mathcal{A}_{r_1,r_2}$, the $(r_1, r_2)$-diameter of a combinatorial complex $\mathcal{X}$ is defined by:

$$\text{diam}_{(\text{co})\mathcal{A}_{r_1,r_2}}(\mathcal{X}) = \max_{x,x' \in \mathcal{X}_{r_1}} d_{(\text{co})\mathcal{A}_{r_1,r_2}}(x, x'), \tag{23}$$

where $d_{(co)\mathcal{A}_{r_1,r_2}}$ is the shortest path distance with respect to neighborhood function $(\text{co})\mathcal{A}_{r_1,r_2}$. Additionally, for $k \in \{1, \ldots, \ell\}$, the $(r_1, r_2, k)$ cross diameter is defined by:

$$\text{diam}^k_{(\text{co})\mathcal{A}_{r_1,r_2}}(\mathcal{X}) = \max_{\substack{x \in \mathcal{X}_{r_1} \\ y \in \mathcal{X}_k}} \min_{x' \subseteq y} d_{(\text{co})\mathcal{A}_{r_1,r_2}}(x, x'). \tag{24}$$

In this section we show that HOMP is unable to compute diameters of CCs, using $\ell$-dimensional tori as a counter example. Corollary B.6 implies that any pair of $\ell$-dimensional tori with the same number of nodes (0-cells) is indistinguishable by HOMP, therefore it is enough to construct such tori with different diameters. E.g. the tori $T_{4,4,32}$ and $T_{8,8,8}$ have the same number of 0-cells but different diameters and cross-diameters for any (co)adjacency function and $k = 1, 2, 3$. This can be extended to tori of any dimensions. More formally we have the following proposition for the $(0, 1)$-diameter.

**Proposition B.7.** *If $T_{p1,\ldots,p_\ell}$ and $T_{p'_1,\ldots,p'_\ell}$ are $\ell$-dimensional tori satisfying*

1. $p_1 \cdots p_\ell = p'_1 \cdots p'_\ell$,

2. $\forall j \in \{1, \ldots, \ell\}$, $p_j, p'_j \geq 3$, and

3. $\sum_{j=1}^{\ell} \lfloor \frac{p_j}{2} \rfloor \neq \sum_{j=1}^{\ell} \lfloor \frac{p'_j}{2} \rfloor$,

*then*

$$diam_{\mathcal{A}_{0,1}}(T_{p_1,\ldots,p_\ell}) \neq diam_{\mathcal{A}_{0,1}}(T_{p'_1,\ldots,p'_\ell}) \tag{25}$$

*but for any HOMP model* $\mathsf{M}$,

$$\mathsf{M}(T_{p_1,\ldots,p_\ell}) = \mathsf{M}(T_{p'_1,\ldots,p'_\ell}). \tag{26}$$

*Proof.* Conditions 1 and 2 imply that $T_{p_1,\ldots,p_\ell}$ and $T_{p'_1,\ldots,p'_\ell}$ are indistinguishable by HOMP. To see that they have different diameters, observe that the graph induced on the nodes of $T_{p_1,\ldots,p_\ell}$ by the adjacency neighborhood $\mathcal{A}_{0,1}$ is the Cartesian product of the cyclic graphs $\text{Cyc}(p_1), \ldots, \text{Cyc}(p_\ell)$. Consequently, since the diameter of a Cartesian product is equal to the sum of diameters over the factors of the product, we have:

$$\begin{aligned}
\text{diam}_{\mathcal{A}_{0,1}}(T_{p_1,\ldots,p_\ell}) = \sum_{j=1}^{\ell} \text{diam}(\text{Cyc}(p_j)) = \sum_{j=1}^{\ell} \left\lfloor \frac{p_j}{2} \right\rfloor &\neq \sum_{j=1}^{\ell} \left\lfloor \frac{p'_j}{2} \right\rfloor \\
&= \sum_{j=1}^{\ell} \text{diam}(\text{Cyc}(p'_j)) \\
&= \text{diam}_{\mathcal{A}_{0,1}}(T_{p'_1,\ldots,p'_\ell}).
\end{aligned} \tag{27}$$

$\square$

**Homology and Betti numbers.** The $r$-th homology group of a cellular complex [8] encodes the structure of "$r$-dimensional holes" in the space (e.g. a circle has a single 1-dimensional hole, a sphere has a single 2-dimensional hole, etc). We denote the $r$-th homology of a CC $\mathcal{X}$ by $H_r(\mathcal{X})$. The *rank* of the $r$-th homology group (i.e. the size of the minimal generating set) is called the $r$-th *Betti number*, denoted by $b_r(\mathcal{X})$.

**Proposition B.8** (HOMP cannot distinguish complexes based on homology)**.** *Let $T = T_{p_1,\ldots,p_\ell}$ be an $\ell$-dimensional torus and $T' = T_{p_1^1,\ldots,p_\ell^1} \sqcup T_{p_1^2,\ldots,p_\ell^2}$ be a disjoint union of two disconnected tori. If $p_1 \cdots p_\ell = p_1^1 \cdots p_\ell^1 + p_1^2 \cdots p_\ell^2$ and $\forall j \in \{1, \ldots, \ell\}$, $p_j, p_j^1, p_j^2 \geq 3$, then $T$ and $T'$ are HOMP-indistinguishable but have different homology groups and Betti number of all orders: $\forall r \in \{0, \ldots, \ell\}$, $H_r(T) \neq H_r(T')$, $b_r(T) \neq b_r(T')$.*

---

[8]Homology is not defined for general combinatorial complexes, only for simplicial / cellular complexes.

*Proof.* First, Lemma B.5 implies that the $T, T_1$, and $T_2$ have a common cover. Thus, since $T$ and $T'$ have the same number of cells, Theorem B.1 implies they are HOMP-indistinguishable. Additionally, for every $H_r(T) = \mathbb{Z}^{\binom{\ell}{r}}$ (see e.g. Hatcher (2002)) and since, $T'$ is a disjoint union of $T_1$ and $T_2$, $H_r(T') = H_r(T_1) \times H_r(T_2) = \mathbb{Z}^{\binom{\ell}{r}} \times \mathbb{Z}^{\binom{\ell}{r}} = \mathbb{Z}^{2\binom{\ell}{r}}$. Therefore, $\forall r \in \{0, \ldots, \ell\}$, $H_r(T) \neq H_r(T')$ and $b_r(T) = \binom{\ell}{r} \neq 2\binom{\ell}{r} = b_r(T')$. □

**Orientability.** We now turn our attention to HOMP's capability to to detect another common topological property: orientability. Loosely speaking, a surface is orientable if one can distinguish between an "inner" and an "outer" side of the surface. A common example of two locally isomorphic surfaces where one is orientable and the other is not is the Möbius strip and a cylinder. For an indepth discussion about orientability and the Möbius strip see Hatcher (2002). We now realize both of these surfaces as cellular complexes. A visualization of the construction can be seen in Figure 1(b). We begin by defining two auxiliary functions.

**Definition B.9.** For $h, p \in \mathbb{N}$ define $\rho_{\mathrm{cyl}}^{h,p}, \rho_{\mathrm{m\ddot{o}b}}^{h,p} : \mathbb{Z}^2 \to \mathbb{Z}^2$ by

$$\rho_{\mathrm{cyl}}^{h,p}(\boldsymbol{s}) = (s_1, s_2 \mod p) \tag{28}$$

$$\rho_{\mathrm{m\ddot{o}b}}^{h,p}(\boldsymbol{s}) = \begin{cases} s_1, s_2 \mod r & s_2 \mod 2p \leq p \\ (h+1-s_1, s_2 \mod r) & s_2 \mod 2p > p. \end{cases} \tag{29}$$

Using $\rho_{\mathrm{cyl}}^{h,p}$ and $\rho_{\mathrm{m\ddot{o}b}}^{h,p}$ we can costruct the cylinder and the Möbius strip.

**Definition B.10** (Cylinder as CC). Given two integers $h, p$, the cylinder $\mathrm{Cyl}_{h,p}$ is a 2-dimensional combinatorial complex $(S, \mathcal{X}, \mathrm{rk})$ defined by:

$$S = [h] \times [p], \tag{30}$$

$$\mathcal{X}_r = \{\boldsymbol{s_k} \mid \boldsymbol{s} \in S, \boldsymbol{k} \in \{0,1\}^2, k_1 + k_2 = r, \rho_{\mathrm{cyl}}^{h,p}(\boldsymbol{s} + \boldsymbol{k}) \in S\}, \tag{31}$$

$$\mathcal{X} = \mathcal{X}_0 \cup \mathcal{X}_1 \cup \mathcal{X}_2, \tag{32}$$

where $\boldsymbol{s_k}$ is defined by:

$$\boldsymbol{s_k} = \{\rho_{\mathrm{cyl}}^{h,p}(\boldsymbol{s} + \boldsymbol{k'}) \mid \boldsymbol{k'} \in \{0,1\}^2, \boldsymbol{k'} \leq \boldsymbol{k}\}. \tag{33}$$

**Definition B.11** (Möbius strip as a CC). Given two integers $h, p$, the Möbius strip $\mathrm{M\ddot{o}b}_{h,p}$ is a 2-dimensional combinatorial complex $(S, \mathcal{X}, \mathrm{rk})$ defined by:

$$S = [h] \times [p], \tag{34}$$

$$\mathcal{X}_r = \{\boldsymbol{s_k} \mid \boldsymbol{s} \in S, \boldsymbol{k} \in \{0,1\}^2, k_1 + k_2 = r, \rho_{\mathrm{m\ddot{o}b}}^{h,p}(\boldsymbol{s} + \boldsymbol{k}) \in S\}, \tag{35}$$

$$\mathcal{X} = \mathcal{X}_0 \cup \mathcal{X}_1 \cup \mathcal{X}_2, \tag{36}$$

where $\boldsymbol{s_k}$ is defined by:

$$\boldsymbol{s_k} = \{\rho_{\mathrm{m\ddot{o}b}}^{h,p}(\boldsymbol{s} + \boldsymbol{k}) \mid \boldsymbol{k'} \in \{0,1\}^2, \boldsymbol{k'} \leq \boldsymbol{k}\}. \tag{37}$$

We now show HOMP is unable to distinguish between CCs based on orientability:

**Proposition B.12** (HOMP cannot detect orientability). *For any two integers $h, p \in \mathbb{N}$ such that $h, p \geq 3$, and for every HOMP model $M$, $\mathrm{Cyl}_{h,p}$ and $\mathrm{M\ddot{o}b}_{h,p}$ are HOMP-indistinguishable, but $\mathrm{Cyl}_{h,p}$ is orientable as a topological space while $\mathrm{M\ddot{o}b}_{h,r}$ is not.*

*Proof.* First, the fact that the cylinder is orientable, whereas the Möbius strip is not is well known (see e.g. Hatcher (2002) for proof). As for HOMP-indistinguishably, consider the wide cylinder $\mathrm{Cyl}_{h,2p}$ with height $h$ and perimeter $2p$. We show that $\mathrm{Cyl}_{h,2p}$ covers both $\mathrm{Cyl}_{h,p}$ and $\mathrm{M\ddot{o}b}_{h,p}$. Since the two CCs are connected and have the same number of nodes, Theorem B.1 implies that they are HOMP-indistinguishable. Denote by $\tilde{S}, S^{\mathrm{cyl}}, S^{\mathrm{m\ddot{o}b}}$ and $\tilde{\mathcal{X}}, \mathcal{X}^{\mathrm{cyl}}, \mathcal{X}^{\mathrm{m\ddot{o}b}}$ the sets of nodes and cells of $\mathrm{Cyl}_{h,2p}, \mathrm{Cyl}_{h,p}$ and $\mathrm{M\ddot{o}b}_{h,p}$ respectively. Define $\rho : \tilde{S} \to S^{\mathrm{cyl}}$ and $\rho' : \tilde{S} \to S^{\mathrm{m\ddot{o}b}}$ by $\rho = \rho_{\mathrm{cyl}}^{h,p}\big|_{\tilde{S}}$ and $\rho' = \rho_{\mathrm{m\ddot{o}b}}^{h,p}\big|_{\tilde{S}}$. It's easy to verify that $\rho(\tilde{S}) = S^{\mathrm{cyl}}$ and $\rho'(\tilde{S}) = S^{\mathrm{m\ddot{o}b}}$, thus $\rho$ and

$\rho'$ are well defined and surjective. $\rho, \rho'$ induce maps $\mathcal{P}(\tilde{S}) \to \mathcal{P}(S^{\text{cyl}})$ and $\mathcal{P}(\tilde{S}) \to \mathcal{P}(S^{\text{möb}})$; by abuse of notation we refer to these maps by $\rho, \rho'$ as well. To show that $\rho$ and $\rho'$ are covering maps, we first show that they are rank-preserving (i.e. that $\rho(\tilde{\mathcal{X}}_r) = \mathcal{X}_r^{\text{cyl}}$ and $\rho'(\tilde{\mathcal{X}}_r) = \mathcal{X}_r^{\text{möb}}$), and then show that they are local isomorphisms. Recall that all elements of $\tilde{\mathcal{X}}_r$ are of the form $\tilde{s}_{\boldsymbol{k}}$ for some $\tilde{s} \in \tilde{S}$ and $\boldsymbol{k} \in \{0, 1\}^2$ such that $k_1 + k_2 = r$. For every $\boldsymbol{k}' \leq \boldsymbol{k}$

$$\rho(\rho_{\text{cyl}^{h,2p}}(\tilde{s} + \boldsymbol{k}')) = \rho_{\text{cyl}^{h,p}}(\rho(\tilde{s}) + \boldsymbol{k}'), \tag{38}$$

so $\rho(\tilde{s}_{\boldsymbol{k}}) = \rho(\tilde{s})_{\boldsymbol{k}}$. Additionally,

$$\rho'(\rho_{\text{cyl}}^{h,2p}(\tilde{s} + \boldsymbol{k}')) = \begin{cases} \rho_{\text{möb}}^{h,p}(\rho'(\tilde{s}) + \boldsymbol{k}') & \tilde{s}_1 \leq p \\ \rho_{\text{möb}}^{h,p}(\rho'(\tilde{s}) + (-k_1', k_2')) & \tilde{s}_1 > p. \end{cases} \tag{39}$$

so

$$\rho'(\tilde{s}_{\boldsymbol{k}}) = \begin{cases} \rho'(\tilde{s})_{\boldsymbol{k}} & \tilde{s}_1 \leq p \\ (\rho'(\tilde{s}) + (-1, 0))_{\boldsymbol{k}} & \tilde{s}_1 > p. \end{cases} \tag{40}$$

By the definitions $\tilde{\mathcal{X}}_r$, $\mathcal{X}^{\text{cyl}}$ and $\mathcal{X}^{\text{möb}}$ we now have $\rho(\tilde{\mathcal{X}}_r) = \mathcal{X}_r^{\text{cyl}}$ and $\rho'(\tilde{\mathcal{X}}_r) = \mathcal{X}_r^{\text{möb}}$ as needed. Since $\rho$ and $\rho'$ are extended to $\mathcal{P}(\tilde{S})$ from $\tilde{S}$, for every $x, y, z \in \tilde{\mathcal{X}}$

- $x \subseteq y \Rightarrow \rho(x) \subseteq \rho(y)$ and $\rho'(x) \subseteq \rho'(y)$.

- $x, y \subseteq z \Rightarrow \rho(x), \rho(y) \subseteq \rho(z)$ and $\rho'(x), \rho'(y) \subseteq \rho'(z)$

- $z \subseteq x, y \Rightarrow \rho(z) \subseteq \rho(x), \rho(y)$ and $\rho'(z) \subseteq \rho'(x), \rho'(y)$.

Therefore, $\rho$ and $\rho'$ preserve all natural neighborhood functions. Finally, since $h, p \geq 3$, for $x, y \in \tilde{\mathcal{X}}$ and $\mathcal{N} \in \boldsymbol{\mathcal{N}}_{\text{nat}}$, $y \in \mathcal{N}(x) \Rightarrow \rho(x) \neq \rho(y)$ and $\rho'(x) \neq \rho'(y)$. This implies that $\rho$ and $\rho'$ are local isomorphisms, completing the proof. □

**Planarity.** A topological space is considered planar if it can be continuously embedded in $\mathbb{R}^2$. Proposition B.12 provides us with the following corollary.

**Corollary B.13** (HOMP cannot detect planarity). *There exist pairs of cellular complexes $\mathcal{X}, \mathcal{X}'$ such that the induced topology of $\mathcal{X}$ is planar while the induced topology of $\mathcal{X}'$ is not, but $\mathcal{X}$ and $\mathcal{X}'$ are HOMP-indistinguishable.*

*Proof.* The CCs $\text{Cyl}_{h,p}$ and $\text{Möb}_{h,p}$ for $p, h \geq$ are HOMP-indistinguishable according to Proposition B.12. The Möbius strip is not planar (see e.g., Hatcher (2002)), whereas the cylinder is. □

### B.3 LIFTING AND POOLING

In this section, we rigorously state and prove Proposition 4.4. We begin by focusing on lifting operations, proving Proposition 4.4 for triangular lifting, as used in Bodnar et al. (2021b) and Bodnar et al. (2021a). Next, we address pooling operations, proving the proposition for MOG pooling (Hajij et al., 2018), which was used to in conjunction with HOMP in Hajij et al. (2022b). While we only provide proofs for these triangular lifting and MOG, we note that this phenomenon generalizes to other lifting and pooling methods as well.

**Lifting.** We first define triangular lifting on graphs, denoted by 3-CL.

**Definition B.14** (Triangular lifting). The triangular lift of a graph $\mathcal{G} = (\mathcal{V}, \mathcal{E})$ is a combinatorial complex denoted by $3 - \text{CL}(\mathcal{G})$, with $S = \mathcal{V}$, $\mathcal{X}_0 = \{\{v\} \mid v \in \mathcal{V}\}$, $\mathcal{X}_1 = \mathcal{E}$, and $\mathcal{X}_2 = \{\{x, y, z\} \mid x \sim y, x \sim z, y \sim z\}$.

We now formally state Proposition 4.4 for triangular lifting

**Proposition B.15.** *There exist pairs of graphs $\mathcal{G}$ and $\mathcal{G}'$ such that the combinatorial complexes $\mathcal{X} = 3 - \text{CL}(\mathcal{G})$ and $3\mathcal{X}' = 3 - \text{CL}(\mathcal{G}')$ are indistinguishable by HOMP. This occurs despite the fact that the cross diameter $\text{diam}_{\mathcal{A}_{0,1}}^2(\mathcal{X})$ is finite while $\text{diam}_{\mathcal{A}_{0,1}}^2(\mathcal{X}')$ is infinite.*

*Proof.* Given $n, k \in \mathbb{N}$ where $k > 3$, the star graph $\text{Star}_{n,k}$ is constructed as follows. We begin with a cyclic graph of length $n \cdot k$ with nodes $a_1, \ldots, a_{n \cdot k}$ where $a_i \sim a_j \Leftrightarrow i - j = \pm 1 \mod n \cdot k$. Secondly, we add $k$ additional nodes, denoted as $b_1, \ldots, b_k$, and connect them to the existing graph via $b_i \sim a_{n \cdot i}$ and $b_i \sim a_{n \cdot i + 1}$, with all index computations carried out modulo $n \cdot k$. We consider a pair of graphs $\mathcal{G}$ and $\mathcal{G}'$ defined by $\mathcal{G} = \text{Star}_{n,2k}$ and $\mathcal{G}' = \text{Star}_{n,k} \sqcup \text{Star}_{n,k}$ where $\sqcup$ represents disjoint union. See Figure 10 for an illustration of the case $k = 3, n = 2$.

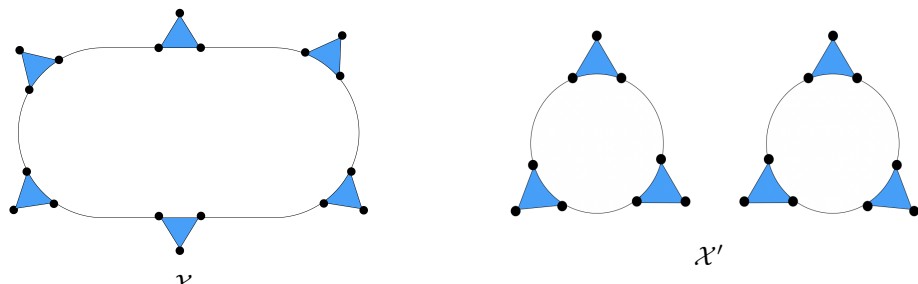

Figure 10: A pair of indistinguishable CCs produced by triangular lifting. The left-hand CC covers each connected component of the right-hand CC.

Since $n \cdot k > 3$, the only triangles in $\mathcal{G}$ and $\mathcal{G}'$ are of the form $\{b_i, a_{i \cdot n}, a_{i \cdot n + 1}\}$. Denote the combinatorial complexes constructed from $\mathcal{G}$ and $\mathcal{G}'$ by applying triangular lifting as $\mathcal{X}$ and $\mathcal{X}'$, respectively. Additionally, denote $3-\text{CL}(\text{Star}_{n,k})$ by $\mathcal{X}^*$. Since $\mathcal{X}'$ consists of two disconnected copies of $\mathcal{X}^*$, and the complexes $\mathcal{X}'$ and $\mathcal{X}$ are of equal size, Theorem B.1 implies that in order to show HOMP cannot distinguish between $\mathcal{X}$ and $\mathcal{X}'$, it suffices to show that $\mathcal{X}$ is a cover of $\mathcal{X}^*$. Letting $S$ ad $S^*$ be the node sets corresponding to CCs $\mathcal{X}, \mathcal{X}^*$, we construct a covering map $\rho : S \to S^*$ defined by:

$$\begin{aligned} \rho(a_i) &= a'_{i \bmod n \cdot k} \\ \rho(b_i) &= b'_{i \bmod k}. \end{aligned} \tag{41}$$

Note that $\rho$ induces a map from $\mathcal{P}(S)$ to $\mathcal{P}(S^*)$ where $\mathcal{P}(\cdot)$ denotes the power set. We abuse notation and refer to this function by $\rho$ as well. We notice that $\rho$ is surjective, and that for any pair of nodes $u, v$ in graph $\mathcal{G}$ we have:

$$u \sim_G v \Rightarrow \rho(u) \sim_{G^*} \rho(v). \tag{42}$$

This implies that $\rho$ preserves triangles as well. In addition, since $n \cdot k > 3$ $\rho$ is locally injective. Thus, $\rho$ is a covering map, and $\mathcal{X}$ and $\mathcal{X}'$ are indistinguisable by HOMP. Finally, it is evident that $\text{diam}^2_{\mathcal{A}_{0,1}}(\mathcal{X}') = \infty$ since it consists of two disjoint connected components, each containing a non-empty set of nodes (0-cells) and triangles (2-cells). Conversely, $\text{diam}^2_{\mathcal{A}_{0,1}}(\mathcal{X}) < \infty$ because it consists of a single connected component. $\qquad\square$

**Pooling.** For the pooling example we focus on the Mapper algorithm Singh et al. (2007); Hajij et al. (2018); Dey et al. (2016), a topology preserving pooling algorithm which was previously used in combination with HOMP in Hajij et al. (2022b). We now define *mapper on graphs* (MOG), a pooling procedure that takes a general graph as input and produces a 2-dimensional combinatorial complex.

**Definition B.16** (Mapper on graphs). Let $\mathcal{G} = (\mathcal{V}, \mathcal{E})$ be a graph, $g : \mathcal{V} \to \mathbb{R}$ be a node function, and $\mathcal{U} = \{U_\alpha\}_{\alpha \in I}$ an open covering of $\mathbb{R}$. The MOG pooling of the graph, $\text{MOG}(\mathcal{G}) = (S, \mathcal{X}, \text{rk})$ is given by the following consturction.

1. Compute the pull-back cover $g^*(\mathcal{U}) = \{g^{-1}(U_\alpha)\}_{\alpha \in I}$.

2. Construct $\mathcal{V}_{\text{MOG}}$ to be the set connected components of the sub-graphs induced by $g^*(\mathcal{U})$.

3. Construct the pooled CC to be $(S, \mathcal{X}, \text{rk})$ with nodes $S = \mathcal{V}$, cells $\mathcal{X} = \mathcal{V} \cup \mathcal{E} \cup \mathcal{V}_{\text{MOG}}$, and rank

$$\text{rk}(x) = \begin{cases} 0 & x \in \mathcal{V} \\ 1 & x \in \mathcal{E} \\ 2 & x \in \mathcal{V}_{\text{MOG}}. \end{cases}$$

In the context of shape detection, a natural choice for $g$ is the average shortest-path distance $g(v) = \frac{1}{|\mathcal{V}|} \sum_{u \in V} d(u,v)$, as it only depends on the graph structure and is thus invariant to geometric transformations of the features. As for the covering $\mathcal{U}$, a natural choice is $\{(\eta \cdot i, \eta \cdot i + \epsilon)\}_{i \in \mathbb{N}}$ for hyper-parameters $\eta, \epsilon \in \mathbb{R}$. We now formally state Proposition 4.4 for MOG pooling.

**Proposition B.17.** *There exist pairs of graphs $\mathcal{G}$ and $\mathcal{G}'$ such that the combinatorial complexes $\mathcal{X} = \mathrm{MOG}(\mathcal{G})$ and $\mathcal{X}' = \mathrm{MOG}(\mathcal{G}')$ are indistinguishable by HOMP. This occurs despite the fact that the $(0,1,2)$ cross diameters of the two CCs are different.*

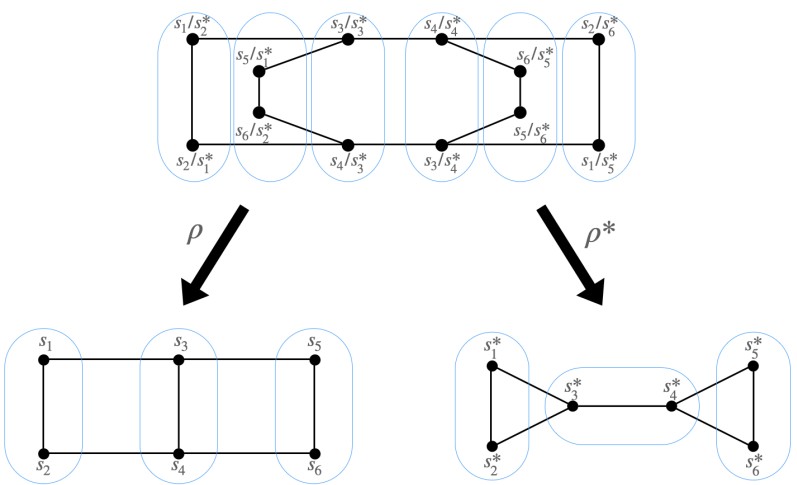

Figure 11: Two combinatorial complexes constructed by MOG pooling and their common cover. The two cells constructed by the MOG algorithm are marked in blue and the covering map is defined based on the node labels.

*Proof.* We begin with a base example. Let $\mathcal{G}$ and $\mathcal{G}'$ be the two graphs depicted at the bottom of Figure 11 before applying the MOG lifting procedure, and denote their node sets by $S = \{s_1, \ldots, s_6\}$ and $S' = \{s'_1, \ldots, s'_6\}$, with the order as shown in the Figure. Note that the sets $P_1 = \{s_1, s_2, s_5, s_6\}$ $P_2 = \{s_3, s_4\}$ form a partition of $S$, and nodes within the same partition set are automorphic (i.e., there exists a graph automorphism mapping one node to the other). The same holds for $P'_1 = \{s'_1, s'_2, s'_5, s'_6\}$ $P'_2 = \{s'_3, s'_4\}$. Thus, the function $g$, defined as the average shortest path distance (SPD) of each node, is constant on each of these sets. This implies that by choosing a sufficiently fine covering (i.e. selecting a small enough $\eta$), the 2-cells defined by the MOG algorithm for graphs $\mathcal{G}$ and $\mathcal{G}'$ will be $x_1 = \{s_1, s_2\}, x_2 = \{s_3, s_4\}, x_3 = \{s_5, s_6\}$ and $x'_1 = \{s'_1, s'_2\}, x'_2 = \{s'_3, s'_4\}, x'_3 = \{s'_5, s'_6\}$ respectively (We split sets $P_1$ and $P'_1$ to their connected components). Defining $\mathcal{X} = \mathrm{MOG}(\mathcal{G})$ and $\mathcal{X}' = \mathrm{MOG}(\mathcal{G}')$, we now aim to show that HOMP cannot distinguish between these two CCs. Figure 11 depicts $\mathcal{X}, \mathcal{X}'$ and an additional combinatorial complex $\tilde{\mathcal{X}}$ which covers both of these complexes. Since $\mathcal{X}$ and $\mathcal{X}'$ are connected and of the same size, Theorem B.1 shows that these complexes are indistinguishable by HOMP. Another quick calculation shows that $\mathrm{diam}^2_{\mathcal{A}_{0,1}}(\mathcal{X}) = 3$ while $\mathrm{diam}^2_{\mathcal{A}_{0,1}}(\mathcal{X}') = 2$ concluding the base example.

To demonstrate that this phenomenon occurs in larger graphs where the MOG procedure produces a small number of 2-cells, we expand upon the aforementioned example. For each integer $n \geq 3$, let $\mathrm{Cyc}(n)$ denote the cyclic graph of length $n$ with the node set $\mathcal{V}_n = \{v_1, \ldots, v_n\}$, where $v_i \sim v_j$ if and only if $i - j = \pm 1 \mod n$. Define $\mathcal{G}_n = \mathcal{G} \times \mathrm{Cyc}(n)$, $\mathcal{G}'_n = \mathcal{G}' \times \mathrm{Cyc}(n)$, , where $\times$ denotes the Cartesian graph product. We now demonstrate that the proof above remains valid for $\mathcal{G}_n$ and $\mathcal{G}'_n$ for any $n \geq 3$. First, defining $P_{1,n} = P_1 \times \mathcal{V}_n$, $P_{2,n} = P_2 \times \mathcal{V}_n$ where here $\times$ denotes set cartesian product, we notice that since all nodes in $P_i$ are $\mathcal{G}$ isomorphic, and all nodes in $\mathcal{V}_n$

are $\text{Cyc}(n)$ isomorphic, all nodes in $P_{i,n}$ are $\mathcal{G}_n$ isomorphic. The same holds for $P'_{1,n} = P'_1 \times \mathcal{V}_n$, $P'_{2,n} = P'_2 \times \mathcal{V}_n$. Thus, again, the function $g$, defined as the average shortest path distance of each node, is constant on each of these sets. This implies that by choosing a sufficiently fine covering, the 2-cells defined by the MOG algorithm for graphs $\mathcal{G}_n$ and $\mathcal{G}'_n$ will be $x_{i,n} = x_i \times \mathcal{V}_n$ and $x'_{i,n} = x'_i \times V_n$ respectively for all $i \in [3]$. Defining $\mathcal{X}_n = \text{MOG}(\mathcal{G}_n)$, $\mathcal{X}'_n = \text{MOG}(\mathcal{G}'_n)$ we now aim to show that HOMP cannot distinguish between these two CCs.

We define $\tilde{\mathcal{G}}$ to be the 1-skeleton of $\tilde{\mathcal{X}}$, $\tilde{\mathcal{G}}_n = \tilde{\mathcal{G}} \times \text{Cyc}(n)$ and $\tilde{\mathcal{X}}_n$ to be the CC whose 1-skeleton is $\tilde{\mathcal{G}}_n$ and whose 2-cells are: $\{x \times \mathcal{V}_n \mid x \in \tilde{\mathcal{X}}_2\}$. Let $\rho : \tilde{S} \to S$, $\rho' : \tilde{S} \to S'$ be the covering maps from $\tilde{\mathcal{X}}$ to $\mathcal{X}$ and $\mathcal{X}'$ respectively, as depicted in Figure 11. Define $\rho_n : \tilde{S} \times \mathcal{V}_n \to S \times \mathcal{V}_n$ by $\rho_n(s,v) = (\rho(s), v)$ and $\rho'_n : \tilde{S} \times \mathcal{V}_n \to S' \times \mathcal{V}_n$ by $\rho'_n(s,v) = (\rho'(s), v)$. By the definitions of $\mathcal{X}_n, \mathcal{X}'_n$ and $\tilde{\mathcal{X}}_n$, these two maps are covering maps from $\tilde{\mathcal{X}}_n$ to $\mathcal{X}_n, \mathcal{X}'_n$ respectively. Since $\mathcal{X}_n, \mathcal{X}'_n$ are connected and are of the same size Theorem B.1 implies they are indistinguishable by HOMP. Secondly Since $\mathcal{G}_n$ is a graph cartesian product, for every $(s,v), (s', v') \in S \times' g V_n$ we have:

$$d_{\mathcal{G}_n}((s,v),(s',v')) = d_{\mathcal{G}}(s,s') + d_{\text{Cyc}(n)}(v,v'). \tag{43}$$

Thus, it is easy to check that: $\text{diam}^2_{\mathcal{A}_{0,1}}(\mathcal{X}_n) = \text{diam}^2_{\mathcal{A}_{0,1}}(\mathcal{X}) = 3$. The same reasoning shows that $\text{diam}^2_{\mathcal{A}_{0,1}}(\mathcal{X}'_n) = \text{diam}^2_{\mathcal{A}_{0,1}}(\mathcal{X}') = 2$ concluding the proof. $\qquad\square$

## C   MULTI-CELLULAR NETWORKS

In this section we motivate and formally define MCN, expanding the discussion in Section 5. We rigorously define both the equivariant linear updates and the general tensor diagram forward pass.

**Multi-cellular cochain spaces**   As discussed in Section 5, given an $\ell$-dimensional CC $\mathcal{X}$ and an $(\ell + 1)$-tuple $\boldsymbol{k} \in \mathbb{N}^{\ell+1}$, the space of $\boldsymbol{k}$-multi-cellular cochains is defied by:

$$\mathcal{C}^{\boldsymbol{k}}(\mathcal{X}, \mathbb{R}^d) = \{\mathbf{h}_{\boldsymbol{k}} \mid \mathbf{h}_{\boldsymbol{k}} : \mathcal{X}_0^{k_0} \times \cdots \times \mathcal{X}_\ell^{k_\ell} \to \mathbb{R}^d\}. \tag{44}$$

Multi-cellular cochain spaces are a natural generalization of standard cochain spaces, providing a way to represent diverse types of CC data. For example, when $\boldsymbol{k} = \boldsymbol{e}_i$, we get that $\mathcal{C}^{\boldsymbol{e}_i}(\mathcal{X}, \mathbb{R}^d)$ is the space of function $\mathbf{h}_{\boldsymbol{e}_i} : \mathcal{X}_i \to \mathbb{R}^d$, i.e. $\mathcal{C}^{\boldsymbol{k}}(\mathcal{X}, \mathbb{R}^d)$ is the space of all possible $i$-rank cell features (i.e. $\mathcal{C}^{\boldsymbol{e}_i} \cong \mathcal{C}^i$ ). In addition, for any pair of integers $r_1, r_2$, the incidence neighborhood function $\mathcal{B}_{r_1, r_2}$ can be encoded as the map $\mathbf{h} : \mathcal{X}_{r_1} \times \mathcal{X}_{r_2} \to \mathbb{R}^d$ defined by:

$$\mathbf{h}(x,y) = \begin{cases} 1 & y \in \mathcal{B}_{r_1, r_2}(x) \\ 0 & \text{otherwise.} \end{cases} \tag{45}$$

Thus $\mathcal{B}_{r_1, r_2}$ can be regarded as an element of the space $\mathcal{C}^{\boldsymbol{e}_{r_1} + \boldsymbol{e}_{r_2}}$. Finally, neighborhood functions of the type $(\text{co})\mathcal{A}_{r_1, r_2}$ can be similarly encoded as the map $\mathbf{h} : \mathcal{X}_{r_1}^2 \to \mathbb{R}^d$ defined by:

$$\mathbf{h}(x,y) = \begin{cases} 1 & y \in (\text{co})\mathcal{A}_{r_1, r_2}(x) \\ 0 & \text{otherwise.} \end{cases} \tag{46}$$

Thus $(co)\mathcal{A}_{r_1, r_2}$ can be regarded as an element of the space $\mathcal{C}^{2 \cdot \boldsymbol{e}_{r_1}}$.

Multi-cellular cochain spaces recover many linear spaces studied in several previous works. For example, $\mathcal{C}^{k\boldsymbol{e}_r}$ matches the features space of a $k$-IGN layer operating on $r$-cells, and $\mathcal{C}^{\boldsymbol{e}_{r_1} + \boldsymbol{e}_{r_2}}$ corresponds to the input space of the exchangeable matrix layers introduced in Hartford et al. (2018).

**Equivariant linear maps between multi-cellular cochain spaces.**   Let $\mathcal{X}$ be a combinatorial complex. We define $n_r = |\mathcal{X}_r|$, representing the size of $\mathcal{X}_r$. For a tuple $\boldsymbol{k} = (k_0, \ldots, k_\ell)$, we define the product space $\mathcal{X}^{\boldsymbol{k}} = \mathcal{X}_0^{k_0} \times \cdots \times \mathcal{X}_\ell^{k_\ell}$. The group $S_{n_0} \times \cdots \times S_{n_\ell}$ is denoted by $G$. We aim to find a basis for the space of equivariant linear layers $L : \mathcal{C}^{\boldsymbol{k}}(\mathcal{X}, \mathbb{R}^d) \to \mathcal{C}^{\boldsymbol{k}'}(\mathcal{X}, \mathbb{R}^{d'})$ for each pair of tuples $\boldsymbol{k}, \boldsymbol{k}'$. Since the space $\mathcal{C}^{\boldsymbol{k}}$ can be identified with $\mathbb{R}^{n_0^{k_0} \times \cdots \times n_\ell^{k_\ell} \times d}$, the space of linear maps between the two can be considered as the space of matrices:

$$\mathcal{C}^{\boldsymbol{k}} \otimes \mathcal{C}^{\boldsymbol{k}'} = \mathbb{R}^{n_0^{k_0} \times \cdots \times n_\ell^{k_\ell} \times d \times n_0^{k'_0} \times \cdots \times n_\ell^{k'_\ell} \times d'}. \tag{47}$$

where we use $\otimes$ to denote the tensor product of vector spaces. The index set corresponding to this space of matrices is $\mathcal{X}^{\boldsymbol{k}} \times [d] \times \mathcal{X}^{\boldsymbol{k}'} \times [d']$.

Notice that the group $G$ acts naturally on $\mathcal{X}^{\boldsymbol{k}} \times \mathcal{X}^{\boldsymbol{k}'}$. For each $G$-orbit $\gamma$ and integers $j_1 \in [d_1]$, $j_2 \in [d_2]$, we define a matrix $\mathbf{B}^{\gamma, j_1, j_2} \in \mathbb{R}^{n_0^{k_0} \times \cdots \times n_\ell^{k_\ell} \times d \times n_0^{k'_0} \times \cdots \times n_\ell^{k'_\ell} \times d'}$:

$$\mathbf{B}^{\gamma, j_1, j_2}_{\boldsymbol{a}, i_1, \boldsymbol{b}, i_2} = \begin{cases} 1 & (\boldsymbol{a}, \boldsymbol{b}) \in \gamma, \quad i_1 = j_1, \quad i_2 = j_2 \\ 0 & \text{otherwise.} \end{cases} \tag{48}$$

Here $\boldsymbol{a} \in [n_0]^{k_0} \times \cdots \times [n_l]^{k_l}$, $\boldsymbol{b} \in [n_0]^{k'_0} \times \cdots \times [n_l]^{k'_l}$ $i_1 \in [d]$, $i_2 \in [d']$.

If $\mathbf{h} \in \mathcal{C}^{\boldsymbol{k}}$ and $\mathbf{h}' = \mathbf{B}^{\gamma, j_1, j_2} \mathbf{h}$ we have:

$$\mathbf{h}'(\boldsymbol{b})_j = \begin{cases} \sum_{(\boldsymbol{a}, \boldsymbol{b}) \in \gamma} \mathbf{h}(\boldsymbol{a})_{j_1} & j = j_2 \\ 0 & \text{otherwhise.} \end{cases} \tag{49}$$

where here by abuse of notation $\boldsymbol{a}, \boldsymbol{b}$ were used interchangeably to describe multi-indices and multi-cells. These maps were established as a basis for the space of all equivariant linear functions $L : \mathbb{R}^{n_0^{k_0} \times \cdots \times n_\ell^{k_\ell} \times d} \to \mathbb{R}^{n_0^{k'_0} \times \cdots \times n_\ell^{k'_\ell} \times d}$ in Maron et al. (2018) and thus can be used to characterize the space of equivariant linear layers $L : \mathcal{C}^{\boldsymbol{k}} \to \mathcal{C}^{\boldsymbol{k}'}$. This framework encompasses many layers from previously studied models. For example, by setting $\boldsymbol{k} = \boldsymbol{k}' = k \cdot \boldsymbol{e}_r$, the space of equivariant linear layers corresponds to the space of $k$-IGN layers, which take as input graphs defined on the $k$-rank cells of the input complexes (i.e. graphs whose node sets are $\mathcal{X}_r$). Similarly, with $\boldsymbol{k} = \boldsymbol{k}' = \boldsymbol{e}_r + \boldsymbol{e}_{r'}$ for $r \neq r' \in \mathbb{N}$, this space aligns with the space of linear maps used to construct the exchangeable matrix layer as described in Hartford et al. (2018).

We can now use this basis to construct learnable equivariant linear layers by taking a parametric linear combination of the basis functions:

$$F(\mathbf{h}) = \beta \left( \sum_{\gamma, j_1, j_2} w^{\gamma, j_1, j_2} B^{\gamma, j_1, j_2} \mathbf{h} \right). \tag{50}$$

To construct MCN, We augment HOMP architectures with these equivariant layers. This is embodied by incorporating equivariant linear layers into the tensor diagram scheme. Figure 5 depicts a MCN tensor diagram. We now describe the components of the MCN scheme.

**Diagram.** Similar to HOMP tensor diagrams, MCN tensor diagrams are layered directed graphs with labeled nodes and edges. Each node is labeled by a multi-cellular cochain space, extending the class of node labels used in HOMP tensor diagrams. Directed edges with source and target nodes labeled by $\mathcal{C}^{\boldsymbol{e}_r}$ can be labeled by any neighborhood function, while edges between nodes labeled by other types of multi-cellular cochain spaces are labeled with the new label "equiv".

**Input.** The input to the MCN model is determined by the 0-th layer of the tensor diagram, whose nodes can be labeled by the following types of multi-cellular cochain spaces: (1) nodes labeled by $\mathcal{C}^{\boldsymbol{e}_r}$ which take the $r$-rank cell features as input; (2) Nodes labeled by $\mathcal{C}^{\boldsymbol{e}_{r_1} + \boldsymbol{e}_{r_2}}$ which take the matrix form of the incidence neighborhood $\mathcal{B}_{r_1, r_2}$ as input; (3) Nodes labeled by $\mathcal{C}^{2\boldsymbol{e}_r}$ which take the matrix form of the (co)adjacency matrices $(\text{co})\mathcal{A}_{r, r'}$.

**Update.** At each layer of an MCN tensor diagram, if $v$ is a node labeled by $\mathcal{C}^{\boldsymbol{k}}$ we compute a multi-cellular cochain $\mathbf{h}^{(v)} \in \mathcal{C}^{\boldsymbol{k}}$ by

$$\mathbf{h}^{(v)}_{\boldsymbol{x}} = \bigotimes_{u \in \text{pred}(v)} \boldsymbol{m}_{u,v}(\boldsymbol{x}) \tag{51}$$

where $\boldsymbol{x} \in \mathcal{C}^{\boldsymbol{k}}$ and $\text{pred}(v)$ denotes the set of predecessor nodes in the diagram. Here messages $\boldsymbol{m}_{u,v} \in \mathcal{C}^{\boldsymbol{k}}$ are computed based on the label of the edge $(u, v)$. If the edge is labeled by a neighborhood function $\mathcal{N}$, (in which case $v$ and $u$ are labeled by standard cochain spaces), the message $\boldsymbol{m}_{u,v}$ is computed by

$$\bigoplus_{y \in \mathcal{N}(x)} \mathsf{MLP}_{u,v}(\mathbf{h}^{(u)}_x, \mathbf{h}^{(u)}_y). \tag{52}$$

Note that this is the exact message used in HOMP tensor diagrams. If the label is "equiv", the message is computed as described in Equation 50. By slight abuse of notation, we often denote the collection of multi-cellular cochains associated with the nodes in layer $t$ by $\mathbf{h}^{(t)}$. For all other labels, the message follows the standard tensor diagram update process. The last layer of the tensor diagram contains a single node representing a final readout layer (See Appendix G.1 for implementation details of both the update computation and the possible readout layers used in this paper).

## D  SCALABLE MULTI-CELLULAR NETWORKS

In this section, we provide an in-depth description of the SMCN model presented in Section 6. To construct SMCN, We augment MCN diagrams, reducing the number of possible node labels and adding a new "SCL" update. Figure 5 depicts a SMCN tensor diagram. We now describe the components of the MCN scheme. For the rest of this section, we borrow the notation scheme of Appendix C

**Diagram.**  Similar to HOMP and SMCN tensor diagrams, like MCN, are layered directed graphs with labeled nodes and edges. Each node is labeled by a multi-cellular cochain space $\mathcal{C}^{\boldsymbol{k}}$, but unlike MCN we restrict $\boldsymbol{k}$ such that $\sum k_r \leq 2$, Thus reducing memory complexity. Like before, directed edges with source and target nodes labeled by $\mathcal{C}^{e_r}$ can be labeled by any neighborhood function, while edges between nodes labeled by other types of multi-cellular cochain spaces can be labeled with the new label "equiv". Additionally, directed edges with source and target nodes labeled by $\mathcal{C}^{e_r + e_{r'}}$ can be labeled by 'SCL'.

**Input.**  The input to the SMCN model is identical to the input to the MCN model. As we will now show, the matrix form of neighborhood functions $\mathcal{B}_{r_1, r_2}$ play a similar role to that of node marking policies in subgraph networks Bevilacqua et al. (2021); Frasca et al. (2022); Zhang et al. (2023b). Subgraph networks process feature maps of the form $\mathbf{h} : \mathcal{V} \times \mathcal{V} \to \mathbb{R}^d$ where $\mathcal{V}$ is a set of nodes. Node marking policies employ an initial feature map of:

$$\mathbf{h}^{(0)}(v, u) = \begin{cases} 1 & u = v \\ 0 & \text{otherwise.} \end{cases} \tag{53}$$

Similarly, SCL updates process cochains of the form $\mathbf{h} : \mathcal{X}_{r_1} \times \mathcal{X}_{r_2} \to \mathbb{R}^d$, and the matrix form of $\mathcal{B}_{r_1, r_2}$ satisfies:

$$\boldsymbol{B}(x, y) = \begin{cases} 1 & x \subseteq y \\ 0 & \text{otherwise.} \end{cases} \tag{54}$$

**Update.**  The SMCN update is computed in the same way as the MCN update is computed (see Appendix C), where the message of a directed edge $(u, v)$ labeled by "SCL" is computed by

$$\boldsymbol{m}_{u,v}(x, y) = \bigotimes_{r=0, r'=0}^{\ell} \mathsf{MLP}^{r,r'}\left( \mathbf{h}^{(t)}_{x,y}, \mathbf{h}^{(t)}_{(\text{co})\mathcal{A}_{r_1,r}(x), y}, \mathbf{h}^{(t)}_{x, (\text{co})\mathcal{A}_{r_2, r'}(y)}, \mathbf{h}_{x, \mathcal{B}_{r_1, r_2}(x)}, \mathbf{h}_{\mathcal{B}^{\top}_{r_2, r_1}(y), y} \right), \tag{55}$$

where if $\mathcal{Q}_1 \subseteq \mathcal{X}_{r_1}$ and $\mathcal{Q}_2 \subseteq \mathcal{X}_{r_2}$ are sets of cells, $\mathbf{h}_{\mathcal{Q}_1, y} := \sum_{x' \in \mathcal{Q}_1} \mathbf{h}_{x', y}$ and $\mathbf{h}_{x, \mathcal{Q}_2} := \sum_{y' \in \mathcal{Q}_2} \mathbf{h}_{x, y'}$. The SCL update can be considered an aggregation of subgraph updates on the augmented Hasse graphs induced by the input CC. For each choice of $r, r'$ the compotation inside the aggregation function in Equation 55 is identical to a CS-GNN (Bar-Shalom et al., 2024) update on the augmented Hasse graph (see Definition 6.1) $\mathcal{H}_{(\text{co})\mathcal{A}_{r_1, r}}$ where the set of "super-nodes"[9] is $\mathcal{X}_{r_2}$ and the connectivity of the super nodes is given according to the Hasse graph $\mathcal{H}_{(\text{co})\mathcal{A}_{r_2, r'}}$. More specifially, for $r_1 = r_2 = 0, r, r' = 1$ we recreate the GNN-SSWL+ (Zhang et al., 2023b) update.

**Computational complexity.**  The computational complexity of SMCN depends on the choice of multi-cellular cochains spaces in the tensor diagram. For $\mathcal{C}^{e_{r_1} + e_{r_2}} \to \mathcal{C}^{e_{r_1} + e_{r_2}}$ updates the worst-case computational complexity for the most general model is $\mathcal{O}\left( \ell^2 \cdot d \cdot n_{r_1} \cdot n_{r_2} \right)$, where $d$ is the maximal degree w.r.t any neighborhood function. In our experiments, we use tensor diagrams containing a single type of multi-cellular cochains spaces resulting in models with a runtime complexity

---

[9]CS-GNN is a subgraph model which uses subsets of nodes to construct the bag of subgraphs. These sets of nodes are termed "super-nodes".

of $\mathcal{O}(d \cdot n_0 \cdot n_1)$ and $\mathcal{O}(d \cdot n_0 \cdot n_2)$. As we demonstrate in Proposition 6.6, SMCN architectures with runtime $\mathcal{O}(d \cdot n_0 \cdot n_2)$ are still strictly more expressive than HOMP. This is useful, since in most natural cases $n_2 \ll n_0$. This allows for flexibility in trading off computational complexity and expressive power.

## E  MCN EXPRESSIVE POWER

In this section, we analyze the expressive power of MCN defined in Section 6. We begin by formally defining CC isomorphism, as described in Hajij et al. (2022b).

**Definition E.1** (CC isomorphism). A pair of CCs $(S, \mathcal{X}, \text{rk})$, $(S', \mathcal{X}', \text{rk}')$ are isomorphic if there exists a bijective map $\rho : \mathcal{X} \to \mathcal{X}'$ such that:

1. $\text{rk}(x) = \text{rk}'(\rho(x)) \quad \forall x \in \mathcal{X}$,

2. $x \subseteq y \Rightarrow \rho(x) \subseteq \rho(y) \quad \forall x, y \in \mathcal{X}$ .

Is there exists an isomorphism $\rho : \mathcal{X} \to \mathcal{X}'$ we say that $\mathcal{X}$ and $\mathcal{X}'$ are *isomorphic*; if such an isomorphism does not exist, we say that $\mathcal{X}$ and $\mathcal{X}'$ are *non-isomorphic*.

**Proposition E.2** (MCN expressive power). *If $\mathcal{X}$ and $\mathcal{X}'$ are non-isomorphic there exists an MCN model* M *such that*

$$\mathsf{M}(\mathcal{X}) \neq \mathsf{M}(\mathcal{X}'). \tag{56}$$

*Proof.* First, let $\mathcal{H} = (\mathcal{V}, \mathcal{E})$ and $\mathcal{H}' = (\mathcal{V}', \mathcal{E}')$ be the Hasse graphs of $\mathcal{X}$ and $\mathcal{X}^*$ respectively, defined by

$$\mathcal{V} = \mathcal{X}, \tag{57}$$
$$\mathcal{V}' = \mathcal{X}', \tag{58}$$
$$\mathcal{E} = \{(x, y) \in \mathcal{X} \times \mathcal{X} \mid x \subseteq y, \text{rk}(x) = \text{rk}(y) - 1\}, \tag{59}$$
$$\mathcal{E}' = \{(x', y') \in \mathcal{X}' \times \mathcal{X}' \mid x' \subseteq y', \text{rk}'(x') = \text{rk}'(y') - 1\}. \tag{60}$$

It was shown in Hajij et al. (2022b) that a pair of CCs is isomorphic if and only if their corresponding Hasse graphs are isomorphic. Therefore, in our case, $\mathcal{H}$ and $\mathcal{H}'$ are non-isomorphic graphs. Since any pair of non-isomorphic graphs of size $n$ are $n$-WL distinguishable, and $k$-IGN networks can distinguish between any pair of $k$-WL indistinguishable graphs (see Maron et al. (2019)), it is enough to prove that there exists a MCN model which is able to simulate any $k$-IGN network on the Hasse graphs. Let $\boldsymbol{A}$ be the adjacency matrix of $\mathcal{H}$ and define $n = |\mathcal{X}|$, $n_r = |\mathcal{X}_r|$ for all $r \in \{0, \dots, \ell\}$. $\boldsymbol{A}$ can be decomposed into block matrices $\boldsymbol{A}^{r_1, r_2}$ for $r_1, r_2 \in \{0, \dots, \ell\}$ defined by:

$$\boldsymbol{A}^{r_1, r_2} = \begin{cases} \boldsymbol{0}_{n_i \times n_j} & r_1 \neq r_2 + 1 \\ \boldsymbol{B}_{r_1, r_2} & r_1 = r_2 + 1, \end{cases} \tag{61}$$

where $\boldsymbol{B}_{r_1, r_2}$ is the matrix form of neighborhood function $\mathcal{B}_{r_1, r_2}$. matrices $\boldsymbol{A}^{r_1, r_2}$ can be view as a multi-cellular cochains $\mathbf{h}^{r_1, r_2} \in \mathcal{C}^{\boldsymbol{e}_{r_1} + \boldsymbol{e}_{r_2}}(\mathcal{X}, \mathbb{R})$ so $\boldsymbol{A}$ can be realized as an element of $\mathcal{Q} := \bigtimes_{r_1=0, r_2=0}^{\ell} \mathcal{C}^{\boldsymbol{e}_{r_1} + \boldsymbol{e}_{r_2}}(\mathcal{X}, \mathbb{R})$. Recall that all neighborhood matrices $\boldsymbol{B}_{r_1, r_2}$ are given as input to the MCN model and so we can recover $\boldsymbol{A}$. To show that MCN can simulate any $k$-IGN update on $\boldsymbol{A}$, we need to show that it can compute $L(\boldsymbol{A})$ for any $S_n$-equivariant linear function $L : \mathcal{Q}^{\otimes k} \to \mathcal{Q}^{\otimes k'}$, where $\mathcal{Q}^{\otimes k}$ represents taking the tensor product of $\mathcal{Q}$ with itself $k$ times. Let $G < S_n$ be the subgroup of permutations preserving the subsets $\{1, \dots, n_0\}$, $\{n_0 + 1, \dots, n_0 + n_1\}$, $\dots$, $\{n_0 + \cdots + n_{\ell-1} + 1, \dots, n_0 + \cdots + n_\ell\}$; $G \cong S_{n_0} \times \cdots \times S_{n_\ell} \subseteq [n]$. Since $G$ is a subgroup of $S_n$, all $S_n$ equivariant linear maps are also $G$-equivariant. Thus it is enough to show that we are can compute $L(\mathbf{h})$ for all $G$-equivariant linear maps $L : \mathcal{Q}^{\otimes k} \to \mathcal{Q}^{\otimes k'}$.

The space $\mathcal{Q} = \bigtimes_{r_1=0, r_2=0}^{\ell} \mathcal{C}^{\boldsymbol{e}_{r_1} + \boldsymbol{e}_{r_2}}(\mathcal{X}, \mathbb{R})$ can be embedded into the multi-cellular cochain space $\mathcal{C}^{\mathbf{1}_{\ell+1}}(\mathcal{X}, \mathbb{R}^{(\ell+1)^2})$ via the following map:

$$T(\mathbf{h})(x_0, \dots x_\ell) = \mathop{\Big\|}_{r_1=0, r_2=0}^{\ell} \mathbf{h}^{r_1, r_2}(x_{r_1}, x_{r_2}), \tag{62}$$

where $\|$ stands for concatenation, $\mathbf{1}_{\ell+1} = (1, \dots, 1) \in \mathbb{R}^{\ell+1}$ is the all ones vector, $x_r \in \mathcal{X}_r$ is a cell of rank $r$ and $\mathbf{h} \in \mathcal{Q}$ composed of the multi-cellular cochains $\mathbf{h}^{r_1, r_2} \in \mathcal{C}^{\boldsymbol{e}_{r_1} + \boldsymbol{e}_{r_2}}(\mathcal{X}, \mathbb{R})$. MCN can

use any linear function $L : \mathcal{C}^{k \cdot \mathbf{1}_{\ell+1}}(\mathcal{X}, \mathbb{R}^{(\ell+1)^2}) \to \mathcal{C}^{k' \cdot \mathbf{1}_{\ell+1}}(\mathcal{X}, \mathbb{R}^{(\ell+1)^2})$ which is $G$-equivariant, and so it can compute $L(\mathbf{h})$ for all linear maps as defined above, concluding the proof. $\qquad\square$

# F  SMCN EXPRESSIVE POWER

## F.1  TOPOLOGICAL AND METRIC PROPERTIES

In this section, we formally demonstrate the SMCN's ability to mitigate many of the expressive limitations demonstrated in Appendix B. We begin by providing a useful lemma that allows us to leverage several expressivity results from the subgraph GNN literature in our setting. We then provide an in-depth discussion on the ability of SMCN to express each one of the four aforementioned metric/topological properties: diameter, orientability, planarity, and homology.

**Lemma F.1.** *For any CS-GNN (Bar-Shalom et al., 2024) model* $\mathsf{M}$ *operating on the Hasse graph* $\mathcal{H}_{(co)\mathcal{A}_{r_1,r_2}}$ *using cells of rank $r \geq r_1$ as super-nodes, there exits an SMCN model* $\mathsf{M}'$, *such that for any CC $\mathcal{X}$ of dimension $\geq r_1, r_2, r$,* $\mathsf{M}(\mathcal{H}_{(co)\mathcal{A}_{r_1,r_2}}) = \mathsf{M}'(\mathcal{X})$.

*Proof.* First, note that the incidence matrix $\mathcal{B}_{r_1,r} \in \mathcal{C}^{e_{r_1}+e_r}$ is equivalent to the "simple node marking" defined in Bar-Shalom et al. (2024), so SMCN can recover the input to the CS-GNN architecture. Second, by taking

$$\mathsf{MLP}^{r,r'}(x,y) = \begin{cases} \mathsf{MLP}(x,y) & \text{if } r = r_2 \text{ and } r' = r_1, \\ 0 & \text{otherwise} \end{cases} \tag{63}$$

for some fixed MLP, Equation 55 becomes identical to the CS-GNN update. $\qquad\square$

*Remark* F.2. For the case where $r = r_1$ (i.e. super-nodes are regular Hasse graph nodes) the CS-GNN architecture becomes equivalent to GNN-SSWL+ (Zhang et al., 2023b).

**Diameter.** We first show SMCN is capable of fully leveraging the information provided by the (cross) diameters of an input CC. see Appendix B for a definition.

**Proposition F.3** (SMCN can compute diameters)**.** *If $\mathcal{X}, \mathcal{X}'$ are CCs such that*

$$\mathrm{diam}^r_{\mathcal{A}_{r_1,r_2}}(\mathcal{X}) \neq \mathrm{diam}^r_{\mathcal{A}_{r_1,r_2}}(\mathcal{X}'), \tag{64}$$

*for $r_1, r_2, r \in \mathbb{N}$ with $r_1 \leq r$, then there exists an SMCN model $\mathsf{M}$ such that $\mathsf{M}(\mathcal{X}) \neq \mathsf{M}(\mathcal{X}')$*

*Proof.* In Zhang et al. (2023b), it was shown that GNN-SSWL+, with standard node marking applied to a graph $\mathcal{G} = (\mathcal{V}, \mathcal{E})$, can compute a final feature representation:

$$\mathbf{h}^{(T)}_{u,v} = d_{\mathcal{G}}(u,v) \quad \text{for } u,v \in \mathcal{V}. \tag{65}$$

By taking the maximum over $\mathbf{h}^{(T)}_{u,v}$, GNN-SSWL+ can distinguish between graphs with different diameters. Similarly, It was shown in Bar-Shalom et al. (2024) that CS-GNN with standard node marking applied to a graph $\mathcal{G} = (\mathcal{V}, \mathcal{E})$ and super-node set $\mathcal{V}^*$ can compute a final feature representation

$$\mathbf{h}^{(T)}_{S,v} = d_{\mathcal{G}}(S,v) \quad \text{for } v \in \mathcal{V} \text{ and } S \in \mathcal{V}^*. \tag{66}$$

By taking the maximum over $\mathbf{h}^{(T)}_{S,v}$, CS-GNN with standard node marking can distinguish between graphs with different cross diameters. Thus, applying Lemma F.1 and Remark F.2 on the Hasse graph $\mathcal{H}_{\mathcal{A}_{r_1,r_2}}$ with $\mathcal{X}_r$ as "super-nodes" we get that SMCN can distinguish between CCs with different (cross) diameters. $\qquad\square$

**Orientability and planarity.** We now show SMCN is able to separate the cylinder and the Möbius strip. This implies that SMCN is strictly better than HOMP at detecting planarity and orientability. Understanding SMCN's ability to fully detect orientability or planarity is still and open question and is left for future work.

**Proposition F.4** (SMCN can separate a cylinder and a Möbius strip)**.** *For any two integers $h, p \in \mathbb{N}$ such that $h, p \geq 3$, there exists an SMCN model $\mathsf{M}$, such that:*

$$\mathsf{M}(Cyl_{h,p}) \neq \mathsf{M}(M\ddot{o}b_{h,p}). \tag{67}$$

*Proof.* First, using the terms "edge" and "1-cell" interchangeably, we define two types of edges on $\text{Cyl}_{h,p}$ and $\text{Möb}_{h,p}$. An edge $x \in \mathcal{X}_1$ is called an *interior edge* if $|\mathcal{B}_{1,2}(x)| > 1$, otherwise it's called a *boundary edge*. We denote the boundary edge graph (node set are the nodes contained in the boundary edges and edge set is the boundary edges themselves) of a CC $\mathcal{X}$ by $\partial \mathcal{X}$. We construct the model M by first using a $\mathcal{B}_{1,2}$ aggregation to get the cochain $\mathbf{h}^{(1)} \in \mathcal{C}^{e_1}(\mathcal{X}, \mathbb{R})$

$$\mathbf{h}^{(1)}(x) = \deg_{\mathcal{B}_{1,2}}(x). \tag{68}$$

Next, we use an equivariant linear update to construct the multi-cellular cochain $\mathbf{h}^{(2)} \in \mathcal{C}^{2e_1}(\mathcal{X}, \mathbb{R}^2)$ defined by:

$$\mathbf{h}^{(2)}_{x_1,x_2} = \deg_{\mathcal{B}_{1,2}}(x_1) \,\|\, \deg_{\mathcal{B}_{1,2}}(x_2), \tag{69}$$

where, $\|$ denotes concatenation. Recall that the matrix form of $\text{co}\mathcal{A}_{1,0}$ defines a cochain $\mathbf{h}_{\text{co}\mathcal{A}_{1,0}} \in \mathcal{C}^{2e_1}$ which can be used as input to SMCN. Using $\mathbf{h}_{\text{co}\mathcal{A}_{1,0}}$ can now construct

$$\mathbf{h}^{(3)}_{x_1,x_2} = (\mathbf{h}_{\text{co}\mathcal{A}_{1,0}})_{x_1,x_2} \,\|\, \deg_{\mathcal{B}_{1,2}}(x_1) \,\|\, \deg_{\mathcal{B}_{1,2}}(x_2). \tag{70}$$

Finally, using a stack of equivariant linear layers, we can construct a fourth cochain $\mathbf{h}^{(4)}_{x_1,x_2} = \text{MLP}(\mathbf{h}^{(3)}_{x_1,x_2})$. We use the Memorization Theorem (Yun et al., 2019), and choose MLP that satisfies

$$\text{MLP}(a, b, c) = \begin{cases} 1 & a = b = c = 1 \\ 0 & \text{otherwise.} \end{cases} \tag{71}$$

$\mathbf{h}^{(4)}$ represents the adjacency matrix of $\partial \mathcal{X}$. $\partial \text{Cyl}_{h,p}$ is composed of two disconnected cycles of length $p$; $\partial \text{Möb}_{h,p}$ is composed of a single cycle of length $2p$. These two graphs are distinguishable by subgraph architectures like GNN-SSWL+. Thus, using Lemma F.1 and Remark F.2 we can continue the construction of M so that it will be able to differentiate between $\text{Cyl}_{h,p}$ and $\text{Möb}_{h,p}$. $\square$

**Homology.** We first show that SMCN is able to count the number of connected components i.e. the 0-th homology.

**Proposition F.5** (SMCN can count connected components)**.** *Let $\mathcal{X}, \mathcal{X}'$ be CCs. If the number of connected components of the augmented Hasse graphs $\mathcal{H}_{\mathcal{A}_{r_1,r_2}}$ and $\mathcal{H}'_{\mathcal{A}_{r_1,r_2}}$ is different for some $r_1, r_2 \in \mathbb{N}$ then there exists an SMCN model M such that $\mathsf{M}(\mathcal{X}) \neq \mathsf{M}(\mathcal{X}')$.*

*Proof.* For a graph $\mathcal{G}$, $C(\mathcal{G})$ represents the set of connected components of $\mathcal{G}$, and $\mathcal{G}_v$ denotes the connected component of a node $v \in \mathcal{V}$. Using Lemma F.1 and Remark F.2, it suffices to show that GNN-SSWL+ can distinguish graphs with different numbers of connected components. It was shown in Zhang et al. (2023b) that adding an additional aggregation of the form $\mathbf{h}_{u,v} \mapsto \sum_{v' \in V} \mathbf{h}_{u,v'}$ to GNN-SSWL+ does not affect its capacity to separate graphs. Therefore, for the remainder of this proof, we include this aggregation in GNN-SSWL+. As previously demonstrated, GNN-SSWL+ can compute a feature vector of the form:

$$\mathbf{h}^{(t)}_{u,v} = d_{\mathcal{G}}(u,v) \quad \text{for } u, v \in \mathcal{V}. \tag{72}$$

If $u$ and $v$ are in different connected components, their distance is encoded as $-1$. Let $g_1 : [-1, |\mathcal{V}|] \to \mathbb{R}$ be a continuous function such that:

$$g_1(x) = \begin{cases} 0 & \text{if } x = -1, \\ 1 & \text{if } x \geq -\frac{1}{2}. \end{cases} \tag{73}$$

We can approximate $g_1$ using an MLP and apply it to $\mathbf{h}^{(t)}_{u,v}$ to obtain:

$$\mathbf{h}^{(t+1)}_{u,v} = \begin{cases} 0 & \text{if } v \notin \mathcal{G}_u, \\ 1 & \text{if } v \in \mathcal{G}_u. \end{cases} \tag{74}$$

We now take $\mathbf{h}^{(t+2)}_{u,v} = \sum_{v' \in V} \mathbf{h}^{(t+1)}_{u,v'}$, to get

$$\mathbf{h}^{(t+2)}_{u,v} = |\mathcal{G}_u|. \tag{75}$$

Define $g_2[1, |\mathcal{V}|] \to \mathbb{R}$ to be

$$g_2(x) = \frac{1}{x}. \tag{76}$$

We can approximate $g_2$ using an MLP and apply it to $\mathbf{h}_{u,v}^{(t+2)}$ to obtain

$$\mathbf{h}_{u,v}^{(t+3)} = \frac{1}{|\mathcal{G}_u|}. \tag{77}$$

It was shown in Zhang et al. (2023b) that the final output of a GNN-SSWL+ model can be computed based on the final feature vector $h_{u,v}^{(T)}$ by

$$h_{\text{out}} = \sum_{u,v \in \mathcal{V}} \mathbf{h}_{u,v}^{(T)}. \tag{78}$$

Applying this to $\mathbf{h}_{u,v}^{(t+3)}$, we get

$$\mathbf{h}_{\text{out}} = \sum_{u,v \in \mathcal{V}} \frac{1}{|\mathcal{G}_u|} = \sum_{\mathcal{G}^* \in C(\mathcal{G})} \sum_{u \in \mathcal{G}^*} \frac{|\mathcal{V}|}{|\mathcal{G}^*|} = \sum_{\mathcal{G}^* \in C(\mathcal{G})} |\mathcal{V}| = |\mathcal{V}||C(\mathcal{G})|. \tag{79}$$

Now let $\mathcal{G}, \mathcal{G}'$ be a pair of graphs with a different number of connected components. If these two graphs have a different number of nodes, they can be easily distinguished by GNN-SSWL+. On the other hand, if they have the same number of nodes they can be distinguished by GNN-SSWL+ based on Equation 79. Thus, we have shown that an augmented GNN-SSWL+ model can distinguish between $\mathcal{H}_{\mathcal{A}_{r_1,r_2}}$ and $\mathcal{H}'_{\mathcal{A}_{r_1,r_2}}$, and therefore, there exists an SMCN model M that can separate $\mathcal{X}$ and $\mathcal{X}'$. $\qquad \square$

Since the 0-th homology satisfies $H_0(\mathcal{X}) = \mathbb{Z}^{|C(\mathcal{X})|}$ we additionally get the following corollary.

**Corollary F.6** (SMCN can compute the 0-th homology). *If $\mathcal{X}, \mathcal{X}'$ are CCs such that the 0-th homology group of their induced topological spaces are different, then there exists an SMCN model M such that $\mathsf{M}(\mathcal{X}) \neq \mathsf{M}(\mathcal{X}^*)$.*

Exploring SMCN's capacity to differentiate between CCs based on their higher-order homology groups is left for future work. As a first step, we show that SMCN can successfully separate a natural family of CCs — two-dimensional surfaces embeddable in $\mathbb{R}^3$ — based on any homology group/Betti number.

**Proposition F.7** (SMCN can compute homology groups of surfaces). *Let $\mathcal{X}, \mathcal{X}'$ be two cellular complexes that are realizations of 2-dimensional manifolds (with or without boundary) $\mathcal{M}, \mathcal{M}'$ which are embeddable in $\mathbb{R}^3$. If $\exists r \in \mathbb{N}$ such that $H_r(\mathcal{M}) \neq H_r(\mathcal{M}')$ then there is an SMCN model M such that $\mathsf{M}(\mathcal{X}) \neq \mathsf{M}(\mathcal{X}')$.*

*Proof.* First, since $\mathcal{M}$ is 2-dimensional, the only non-trivial homology groups it may have are of order $0 \leq r \leq 2$. The 0-th homology group of $\mathcal{M}$, is of the form $H_0(\mathcal{M}) = \mathbb{Z}^{k_0}$ where $k_0$ is the number of $\mathcal{M}$'s connected components. Furthermore, as each connected component of $\mathcal{M}$ is a connected 2-dimensional manifold with a boundary that can be embedded in $\mathbb{R}^3$, it must either be orientable or have a non-empty boundary. If such a component is orientable, then by the Poincaré duality, its second homology group is $\mathbb{Z}$. On the other hand, if it has a boundary, it is homotopic to a 1-dimensional cellular complex, and thus its second homology group is trivial. Therefor, $H_2(\mathcal{M}) = \mathbb{Z}^{k_2}$, where $k_2$ is the number of connected components of $\mathcal{M}$ with no boundary. Finally, since $\mathcal{M}$ is embeddable in $\mathbb{R}^3$, its 1-st homology groups is $H_1(\mathcal{M}) = \mathbb{Z}^{k_1}$ for some integer $k_1$. The Euler characteristic of the manifold $\mathcal{M}$ defined by $k_0 - k_1 + k_2$ can be computed using the number of cells of $\mathcal{X}$ using the following formula:

$$\chi(\mathcal{M}) = k_2 - k_1 + k_0 = |\mathcal{X}_2| - |\mathcal{X}_1| + |\mathcal{X}_0|. \tag{80}$$

Thus in order to separate $\mathcal{X}$ from $\mathcal{X}'$ we need to be able to construct a SMCN model that is able to separate CCs that are different in either one of the following three quantities:

1. The Euler characteristic.

2. The number of connected components.

3. The number of connected components with no boundary.

Computing the Euler characteristic is be computed by standard HOMP updates, as it is a function of the sizes of $\mathcal{X}_0, \mathcal{X}_1$, and $\mathcal{X}_2$. For the second quantity, we have seen SMCN models can separate CCs with a different number of connected components in Proposition F.5. As for the third quantity, a connected component of $\mathcal{X}$ has a boundary if and only if it contains 1-cells whose degree with respect to the neighborhood function $\mathcal{B}_{1,2}$ is exactly 1. We can use a stack of standard HOMP layers to compute the 1-cells features

$$\mathbf{h}_x = \begin{cases} 1 & x \text{ is in the same connected component as a boundary edge} \\ 0 & \text{otherwise.} \end{cases} \tag{81}$$

Using $\mathbf{h}$, we can adjust the proof of Proposition F.5 by summing in Equation 79 only over 1-cells for which $\mathbf{h}_x = 0$, resulting in the number of connected components of $\mathcal{X}$ with no boundary. This shows that SMCN can distinguish CCs based on *either one* of the aforementioned three properties, concluding the proof. □

### F.2 LIFTING AND POOLING

In this section, we rigorously state and prove Proposition 6.6 which appears in Section 6.1 for graph triangular lifting and for the MOG (graph Mapper) pooling algorithm. Corresponding results for the HOMP case can be found in Appendix B.3. We start with triangular lifting (Definition B.14).

**Proposition F.8.** *There exist pairs of graphs $\mathcal{G}$ and $\mathcal{G}'$ such that the combinatorial complexes $\mathcal{X} = 3\text{-}CL(\mathcal{G})$ and $\mathcal{X}' = 3\text{-}CL(\mathcal{G}')$ are indistinguishable by HOMP, but can be distinguished by an SMCN model with asymptotic runtime $\mathcal{O}(m_{\deg} \cdot n_0 \cdot n_2 \cdot T)$ where $n_0$ is the number of nodes in the original graph, $n_2$ is the number of 2-rank cells constructed by triangular lifting, $m_{\deg}$ is the maximal degree and $T$ is the number of layers.*

*Proof.* In Proposition B.15, we find a family of graph pairs whose triangular lifts are indistinguishable by HOMP despite having different diameters of type $\mathrm{diam}^2_{\mathcal{A}_{0,1}}$. In Proposition F.3 we saw that using only SCL updates, SMCN can compute this type of diameter and is thus able to distinguish between the aforementioned pairs of CCs. Recall the SCL update rule:

$$\boldsymbol{m}_{u,v}(x,y) = \bigotimes_{r=0,r'=0}^{\ell} \mathsf{MLP}^{r,r'}\left(\mathbf{h}_{x,y}^{(t)}, \mathbf{h}_{(\mathrm{co})\mathcal{A}_{r_1,r}(x),y}^{(t)}, \mathbf{h}_{x,(\mathrm{co})\mathcal{A}_{r_2,r'}(y)}^{(t)}, \mathbf{h}_{x,\mathcal{B}_{r_1,r_2}(x)}, \mathbf{h}_{\mathcal{B}^\top_{r_2,r_1}(y),y}\right),$$

where if $\mathcal{Q}_1 \subseteq \mathcal{X}_{r_1}$ and $\mathcal{Q}_2 \subseteq \mathcal{X}_{r_2}$ are sets of cells, $\mathbf{h}_{\mathcal{Q}_1,y} := \sum_{x' \in \mathcal{Q}_1} \mathbf{h}_{x',y}$ and $\mathbf{h}_{x,\mathcal{Q}_2} := \sum_{y' \in \mathcal{Q}_2} \mathbf{h}_{x,y'}$. Observing the proof of Proposition F.3, we note that in order to be able to compute $\mathrm{diam}^2_{\mathcal{A}_{0,1}}$. It is enough to use only values corresponding to $r = r' = 1, r_1 = 0, r_2 = 2$. the asymptotic runtime of this type of SCL layer is $O(m_{\deg} \cdot n_0 \cdot n_2)$. Thus the overall runtime of our SMCN model is $\mathcal{O}(m_{\deg} \cdot n_0 \cdot n_2 \cdot T)$, completing the proof. □

We now move on to MOG pooling.

**Proposition F.9.** *There exist pairs of graphs $\mathcal{G}$ and $\mathcal{G}'$ such that the combinatorial complexes $\mathcal{X} = MOG(\mathcal{G})$ and $\mathcal{X}' = MOG(\mathcal{G}')$ are indistinguishable by HOMP, but can be distinguished by an SMCN model with asymptotic runtime $\mathcal{O}(d \cdot n_0 \cdot n_2 \cdot T)$ where $n_0$ is the number of nodes in the original graph, $n_2$ is the number 2-rank cells constructed by mapper, $d$ is the maximal degree and $T$ is the number of layers.*

*Proof.* In Proposition B.17, we saw a family of graph pairs for which the CCs obtained by MOG pooling are indistinguishable by HOMP despite having different $(0, 1, 2)$ cross diameters. In Proposition F.3 we saw that using only SCL updates, SMCN can compute this type of diameter and is thus able to distinguish between the aforementioned pairs of CCs. As seen in the proof above, by choosing an appropriate aggregation function $\bigotimes$ the runtime of each SCL layer becomes $O(d \cdot n_0 \cdot n_2)$ (note that by stacking layers that use this aggregation we are still able to compute $\mathrm{diam}^2_{\mathcal{A}_{0,1}}$). Thus the overall runtime of our SMCN model is $O(d \cdot n_0 \cdot n_2 \cdot T)$, completing the proof. □

### F.3 COMPARING THE EXPRESSIVE POWER OF SMCN WITH BASELINES

Section 7 compares the empirical performance of SMCN with several relevant baselines. In this section, we briefly discuss the expressive power of SMCN in comparison to each one of these baselines.

**GNNs.** In Section 7, we evaluate SMCN against various MPNNs (e.g., GIN, GCN), subgraph networks (e.g., DS-GNN (Bevilacqua et al., 2021), SUN (Frasca et al., 2022), GNN-SSWL+ (Zhang et al., 2023b)), and other expressive GNNs (e.g., PPGN (Maron et al., 2019), PPGN+ (Puny et al., 2023)). It is important to note that since GNNs are designed to process graphs, a meaningful comparison of expressivity with SMCN is only valid when SMCN is applied to lifted graphs (see discussion in Section 4.3. Among all MPNNs and subgraph networks, GNN-SSWL+ is the most expressive. Since SMCN can implement GNN-SSWL+ using the $\mathcal{A}_{0,1}$ neighborhood function (which corresponds to the original graph), it follows that SMCN is at least as expressive as all the MPNNs and subgraph networks, regardless of the chosen lifting procedure. Furthermore, SMCN can implement edge deletion subgraph policies, which, as demonstrated in Bevilacqua et al. (2021), are capable of distinguishing certain instances of graphs that are indistinguishable by the 3-WL test. Since the expressivity of GNN-SSWL+ has been shown in Zhang et al. (2023b) to be bounded by the 3-WL test, it follows that SMCN is *strictly* more expressive than all the aforementioned MPNNs and subgraph networks. Finally, the expressive power of PPGN and PPGN+ has also been shown to be bounded by the 3-WL test (Maron et al., 2019; Puny et al., 2023), indicating that there are instances of graphs distinguishable by SMCN but not by PPGN or PPGN+. A comprehensive comparison of the expressive power of PPGN and SMCN is deferred to future work.

**Topological neural networks.** We compare SMCN with four topological neural networks: CIN (Bodnar et al., 2021a), CIN++ (Giusti et al., 2023), CIN + CycleNet (Yan et al., 2024) and Cellular Transformer (Ballester et al., 2024). SMCN can directly implement CIN and CIN++ through the use of suitable tensor diagrams, establishing it as at least as expressive as these architectures. Furthermore, CIN and CIN++ are HOMP-based architectures. As demonstrate in Section 6.1, SMCN can distinguish between CCs that HOMP architectures cannot, making it strictly more expressive.

Additionally, CycleNet processes graphs by applying BasisNet + spectral embedding (Lim et al., 2022) to the 1-Hodge Laplacian of the input graph. The resulting output is then used as edge features, which are subsequently processed by a final CIN applied to the graph lifted to a cellular complex by cyclic lift. In cases where the input graphs are simple and undirected, the 1-Hodge Laplacian is exactly equal to the coadjacency matrix $\mathrm{co}\mathcal{A}_{0,1}$ (plus a diagonal term of $2I$ which can be ignored). SMCN can apply subgraph GNNs to the $\mathrm{co}\mathcal{A}_{0,1}$ Hasse graph, and use the resulting features as inputs for a CIN model. This reduces our proof to demonstrating that subgraph GNNs are more expressive than BasisNet. As recently shown in Zhang et al. (2024), BasisNet + spectral embedding is strictly less expressive than PSWL, a subclass of subgraph GNNs that is itself less expressive than the base subgraph GNN used in SMCN. This shows that SMCN is strictly more expressive that CIN + CycleNet.

Finally, the Cellular Transformer breaks equivariance with respect to $G = S_{n_1} \times \cdots \times S_{n_\ell}$ using Laplacian positional encoding, allowing it to gain expressivity. Architectures that break equivariance are often fully expressive, but they can also incorrectly differentiate between instances of isomorphic CCs. Consequently, the Cellular Transformer is not a fair comparison to SMCN in terms of expressivity.

## G EXPERIMENTAL DETAILS

**Setup.** All models are implemented in PyTorch (Paszke et al., 2019) using the PyTorch Geometric framework (Fey & Lenssen, 2019). We used TopoNetX (Hajij et al., 2024) and NetworkX (Hagberg et al., 2008) to perform lifting operations. Hyperparameter tuning is carried out using Weights and Biases (Biewald, 2020). All experiments were conducted on a single NVIDIA A100-SXM4-40GB GPU. For each experiment, we report the mean and standard deviation over 5 runs with random seeds from 1 to 5. Reported test scores are computed at the epoch achieving the best score.

### G.1 MODEL IMPLEMENTATION

The SMCN framework is highly flexible, allowing for a wide range of model construction approaches. To narrow down the search space, we focus on two types of tensor diagrams, sequential and parallel, each composed of smaller blocks and updates formally defined below. As in the main body, we sometimes omit the subscript in $\mathbf{h}_r$ when the rank is clear from context. We also

use the notation $\mathbf{h}_{\mathcal{Q}} = \sum_{x \in \mathcal{Q}} \mathbf{h}_x$. Additionally, we use the augmented concatenation operator $\mathsf{Cat}(\mathbf{x} \,\|\, \mathbf{y}) := \mathsf{MLP}_1(\mathsf{MLP}_2(\mathbf{x}) \,\|\, \mathsf{MLP}_3(\mathbf{y}))$. All MLPs have a single hidden layer.

**Initialization.** In the case $\mathcal{X}$ is lifted from a graph with node and edge features, these features are used as the 0 and 1 cochains respectively. Otherwise, the 0, and 1 cochains are initialized with zeros. In case the initial features are categorical we apply an embedding layer. Similarly to Bodnar et al. (2021a), we initialize the 2-cell cochain by

$$(\mathbf{h}_2^{(0)})_x = (\mathbf{h}_0^{(0)})_{\mathcal{B}_{2,0}^\top(x)} = \sum_{x' \in \mathcal{B}_{2,0}^\top(x)} (\mathbf{h}_0^{(0)})_{x'}. \tag{82}$$

Figure 12: CIN Block.

**CIN block.** The first HOMP block we consider is the CIN block from Bodnar et al. (2021a), whose tensor diagram is illustrated in Figure 12. A CIN block updates the cochains $\mathbf{h}_0^{(t)}$, $\mathbf{h}_1^{(t)}$ and $\mathbf{h}_2^{(t)}$ via the following update rules. For $x \in \mathcal{X}_0$,

$$\mathbf{h}_x^{(t+1)} = \mathsf{MLP}_{0,1}^{(t)} \left( (1 + \epsilon_0)\mathbf{h}_x^{(t)} + \sum_{x' \in \mathcal{A}_{0,1}(x)} \mathsf{MLP}_{0,2}^{(t)} \left( \mathbf{h}_{x'}^{(t)} \,\|\, \mathbf{h}_{\mathcal{E}(x,x')}^{(t)} \right) \right), \tag{83}$$

for $x \in \mathcal{X}_1$,

$$\mathbf{h}_x^{(t+1)} = \mathsf{Cat} \left( (1 + \epsilon_1^1)\mathbf{h}_x^{(t)} + \mathbf{h}_{\mathcal{B}_{1,0}^\top(x)}^{(t)} \,\middle\|\, (1 + \epsilon_1^2)\mathbf{h}_x^{(t)} + \sum_{x' \in \mathcal{A}_{1,2}(x)} \mathsf{MLP}_{0,2}^{(t)} \left( \mathbf{h}_{x'}^{(t)} \,\|\, \mathbf{h}_{\mathcal{E}(x,x')}^{(t)} \right) \right) \tag{84}$$

and for $x \in \mathcal{X}_2$,

$$\mathbf{h}_x^{(t+1)} = \mathsf{MLP}_2^{(t)} \left( (1 + \epsilon_2)\mathbf{h}_x^{(t)} + \mathbf{h}_{\mathcal{B}_{2,1}^\top(x)}^{(t)1} \right), \tag{85}$$

where if $x, x' \in \mathcal{X}_r$, $\mathcal{E}(x,x') = \{ y \in \mathcal{X}_{r+1} \mid x, x' \subseteq y \}$ and $\epsilon_0, \epsilon_1^1, \epsilon_1^2, \epsilon_2$ are non-learnable hyperparameters.

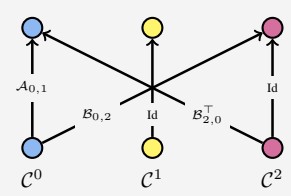

Figure 13: Custom HOMP.

**Custom HOMP block.** In addition to CIN blocks, we also use "custom HOMP" blocks that are designed to minimize the influence of 1-cells in the block update, thereby reducing dependence on the edge structure of graphs. Empirical results, as shown in Table 2, demonstrate that this approach improves performance on certain topological property prediction tasks. The exact update for this block is defined via the following rule. For $x \in \mathcal{X}_0$,

$$\mathbf{h}_x^{(t+1)} = \mathsf{Cat} \left( (1 + \epsilon_0^1)\mathbf{h}_x^{(t)} + \mathbf{h}_{\mathcal{B}_{0,2}(x)}^{(t)} \,\middle\|\, (1 + \epsilon_0^2)\mathbf{h}_x^{(t)} + \mathbf{h}_{\mathcal{A}_{0,1}(x)}^{(t)} \right) \tag{86}$$

for $x \in \mathcal{X}_1$,

$$\mathbf{h}_x^{(t+1)} = \mathbf{h}_x^{(t)}, \tag{87}$$

and for $x \in \mathcal{X}_2$

$$\mathbf{h}_x^{(t+1)} = \mathsf{MLP}^{(t)} \left( (1 + \epsilon_2)\mathbf{h}_x^{(t)} + \mathbf{h}_{\mathcal{B}_{2,0}^\top(x)}^{(t)} \right). \tag{88}$$
.

**Multi-cellular cochain initialization.** SCL layers take as input multi-cellular cochains of the form $\mathbf{h}_{0,1}^{(t)} \in \mathcal{C}_{0,1}$ and $\mathbf{h}_{0,2}^{(t)} \in \mathcal{C}_{0,2}$. These multi-cellular cochains are initialized with

$$(\mathbf{h}_{0,r}^{(t)})_{x_1,x_2} = \mathsf{MLP}_{r,1}^{(t)}((\mathbf{h}_0^{(t)})_{x_1}) + \mathsf{MLP}_{r,2}^{(t)}((\mathbf{h}_r^{(t)})_{x_2}) + \mathsf{MLP}_{r,3}^{(t)}(\mathrm{mark}(x_1,x_2)), \tag{89}$$

where $\mathrm{mark}(x_1, x_2)$ is a marking strategy. Similarly to Zhang et al. (2023b) and Bar-Shalom et al. (2024) we consider two marking strategies: (1) binary marking:

$$\mathrm{mark}_{\mathrm{B}}(x_1, x_2) = \begin{cases} 1 & x_1 \in \mathcal{B}_{0,r}(x_2) \\ 0 & \text{otherwise.} \end{cases} \tag{90}$$

(2) distance-based marking:

$$\mathrm{mark}_{\mathrm{D}}(x_1, x_2) = \min_{x \in \mathcal{B}_{r,0}^\top(x_2)} d_{\mathcal{A}_{0,1}}(x_1, x), \tag{91}$$

where $d_{\mathcal{A}_{0,1}}$ is the shortest path distance on $\mathcal{H}_{\mathcal{A}_{0,1}}$.

**SCL updates.** We use two types of SCL updates, 1-SCL and 2-SCL, defined by

$$
\mathbf{h}_{x_1,x_2}^{(t+1)} = \mathsf{Cat}_1\left((1+\epsilon_1^1)\mathbf{h}_{x_1,x_2}^{(t+1)} + \sum_{x' \in \mathcal{A}_{0,1}(x_1)} \mathsf{MLP}_1^{(t)}\left(\mathbf{h}_{x',x_2}^{(t)} \,\|\, \mathbf{h}_{\mathcal{E}(x_1,x')}^{(t)}\right) \,\Big\|\, (1+\epsilon_1^2)\mathbf{h}_{x_1,x_2}^{(t)} + \mathbf{h}_{x_1,\mathcal{B}_{0,1}(x_1)}^{(t)}\right)
$$
(92)

for $x_1 \in \mathcal{X}_0$, $x_2 \in \mathcal{X}_1$, and

$$
\mathbf{h}_{x_1,x_2}^{(t+1)} = \mathsf{Cat}_2\left((1+\epsilon_2^1)\mathbf{h}_{x_1,x_2}^{(t+1)} + \sum_{x' \in \mathcal{A}_{0,1}(x_1)} \mathsf{MLP}_2^{(t)}\left(\mathbf{h}_{x',x_2}^{(t)} \,\|\, \mathbf{h}_{\mathcal{E}(x_1,x')}^{(t)}\right) \,\Big\|\, (1+\epsilon_2^2)\mathbf{h}_{x_1,x_2}^{(t)} + \mathbf{h}_{x_1,\mathcal{B}_{0,2}(x_1)}^{(t)}\right)
$$
(93)

for $x_1 \in \mathcal{X}_0$, $x_2 \in \mathcal{X}_2$, respectivley.

**SCL pooling.** To use a HOMP block after an SCL block we need to pool the information from multi-cellular cochains to standard cochains. This is done via an SCL pooling block, defied by

$$
(\mathbf{h}_0)_x^{(t)} = \mathsf{MLP}_0^{(t)}\left((\mathbf{h}_{0,r}^{(t)})_{x,\mathcal{X}_r}\right) = \mathsf{MLP}_0^{(t)}\left(\sum_{x' \in \mathcal{X}_r}(\mathbf{h}_{0,r}^{(t)})_{x,x'}\right)
$$

$$
(\mathbf{h}_r)_x^{(t)} = \mathsf{MLP}_r^{(t)}\left((\mathbf{h}_{0,r}^{(t)})_{\mathcal{X}_0,x}\right) = \mathsf{MLP}_r^{(t)}\left(\sum_{x' \in \mathcal{X}_0}(\mathbf{h}_{0,r}^{(t)})_{x',x}\right)
$$
(94)

**Readout.** All tasks considered in this paper require predicting a single value per input CC. Therefore, we employ a final readout layer of the form:

$$
\begin{aligned}
h_{\text{out}} = \mathsf{MLP}\Big( &\mathsf{agg}_0\left(\{\!\{\mathbf{h}_x^{(T)} \mid x \in \mathcal{X}_0\}\!\}\right) + \mathsf{agg}_1\left(\{\!\{\mathbf{h}_x^{(T)} \mid x \in \mathcal{X}_1\}\!\}\right) \\
&+ \mathsf{agg}_2\left(\{\!\{\mathbf{h}_x^{(T)} \mid x \in \mathcal{X}_2\}\!\}\right) + \mathsf{agg}_3\left(\{\!\{\mathbf{h}_{x_1,x_2}^{(T)} \mid x_1 \in \mathcal{X}_0, x_2 \in \mathcal{X}_1\}\!\}\right) \\
&+ \mathsf{agg}_4\left(\{\!\{\mathbf{h}_{x_1,x_2}^{(T)} \mid x_1 \in \mathcal{X}_0, x_2 \in \mathcal{X}_2\}\!\}\right) \Big),
\end{aligned}
$$
(95)

where $T$ is the final layer and $\mathsf{agg}_i$ are either mean aggregation, sum aggregation, or the zero function.

**Tensor diagrams.** We use two types of tensor diagrams: sequential and parallel, both illustrated in Figure 14. In sequential tensor diagram, a stack of HOMP blocks is followed by a stack of SCL updates and then another stack of HOMP blocks. The parallel tensor diagram uses concurrent stacks of HOMP and SCL layers. Both the types of blocks/layers and the number of blocks/layers within each stack are treated as hyperparameters in the model.

## G.2 SYNTHETIC BENCHMARKS

**Torus dataset.** To construct the torus dataset we first select three parameters: $m$ which specifies the number of nodes in the smallest CC in the dataset, $M$ which specifies the number of nodes in the largest CC, and $n$, which specifies the maximum number of connected components in any CC within the dataset. The dataset is then constructed by iterating over all possible choices for the number of nodes and connected components, generating all possible disjoint unions of 2-dimensional tori with the specified parameters. We then select all the pairs that have the same size (number of nodes). As mentioned in the main text, each such pair is indistinguishable by HOMP despite differing in basic metric/topological properties: they either have distinct homology, or they differ in the diameters of some of the components. In our experiments, we use $m = 18$ (the smallest size that admits indistinguishable pairs), $M = 40$, and $n = 3$, resulting in 223 pairs.

To evaluate the ability of both HOMP and SMCN to distinguish between each pair, we follow the training and evaluation protocol presented in Wang & Zhang (2024). For each pair, we generate 64 copies where the order of cells is randomly permuted. The model is then trained to minimize the cosine similarity between the outputs corresponding to the two CCs in each pair. We measure the number of pairs where the output difference is statistically significant. Our results show that, while HOMP fails to distinguish any of the pairs, SMCN successfully differentiates all of them.

We use an SMCN model implementing a sequential tensor diagram composed of two CIN blocks, followed by four 1-SCL updates, and concluding with two additional CIN blocks. In comparison, the HOMP model consists of a stack of four consecutive CIN blocks, designed to have a comparable number of learnable parameters. The readout layer of both models uses a zero function as the aggregation for all multi-cellular cochains and a sum aggregation for all standard cochains. All models are trained for 20 epochs on each individual pair using a constant learning rate of $0.001$. The embdding dimension used by all models is 128.

**Lifted ZINC cross-diameter.** We construct a CC dataset by adding cycles of length $\leq 18$ as 2-cells to graphs taken from the ZINC-12K dataset (Sterling & Irwin, 2015). We remove edge and node features, and predict the $(0, 1, 2)$ cross diameter, computed by:

$$\max_{\substack{x \in \mathcal{X}_0, \, x' \in y \\ y \in \mathcal{X}_2}} \min d_{\mathcal{A}_{0,1}}(x, x') \tag{96}$$

where $d_{\mathcal{A}_{0,1}}(x, x')$ is the shortest path distance w.r.t the original graph. Training targets are normalized to have mean $0$ and standard deviation $1$. The model is trained using an MSE loss. At test time, we evaluate both the MSE of the normalized target as well as the accuracy in predicting the cross-diameter value, which has 18 possible outcomes. We compare three architectures: the first two are HOMP models which employ a stack of four CIN blocks, and a stack of four custom HOMP blocks respectively. The third is an SMCN model, which implements a sequential tensor diagram constructed by a single stack of six 2-SCL blocks, followed by a non-learnable pooling procedure as described in Equation 94. All models are constrained to a budget of 500K learnable parameters. The readout layer of all three models is defined according to Equation 95 where $\text{agg}_1, \text{agg}_3, \text{agg}_4$ are taken to be the zero function and $\text{agg}_0, \text{agg}_2$ are taken to be the mean function. Models are trained for 200 epochs using a constant learning rate of $0.0001$. The hidden dimension used by all models is 64.

**Lifted ZINC second Betti number.** For the second topological property prediction task we tested our model's ability to learn to predict the second order Betti numbers- the ranks of the second homology group. To this end we constructed our benchmark dataset the following way: We started with the ZINC-FULL datasets (containing $250k$ molecular graphs), lifting all graphs to CCs as in the cross-diameter task. We then computed the second Betti number for each of the lifted graphs and randomly selected 850 samples from each of the 6 most common values (which were $0, 1, 2, 3, 4$ and $6$), resulting in a balanced dataset of size $5,100$. We used a $60\%$, $20\%$, $20\%$ random split for training, validation, and test sets. As before, we remove all node and edge features, and normalized training targets to have mean $0$ and standard deviation $1$. The models are then trained using an MSE loss. At test time we evaluated both the MSE of the normalized target as well as the accuracy of predicting the seconnd Betti number. We use the same 3 models reported in the last experiment with the same exact hyperparameters.

The results of both lifted ZINC experiments are presented in Tables 2. Following the experimental setting of (Rieck, 2023) in which TDL models are tasked with learning metric properties of graphs taken from the MOLHIV dataset (Hu et al., 2020), we report the model's accuracy in predicting the correct target value. We additionally provide the normalized MSE score of the model.

SMCN significantly outperforms both HOMP methods in learning both the cross-diameter and the second Betti numbers, achieving higher accuracy as well as significantly lower standard deviation indicating a more stable learning process. This is particularly evident in the custom HOMP model, which, although it surpasses the CIN model in performance, suffers from a considerably larger standard deviation. Additionally, the SMCN model achieves strong results after a significantly lower number of epochs compared to both HOMP models, as illustrated in Figure 15.

Moreover, the results of the three synthetic experiments further demonstrate SMCN's superior capability in capturing the topological properties of CCs compared to existing HOMP architectures. While our theoretical analysis established that SMCN can express topological information beyond the capabilities of any HOMP architecture, the synthetic experiments validate that SMCN models can effectively learn and leverage these properties in practice.

## G.3 REAL-WORLD GRAPH BENCHMARKS

**ZINC Dataset (Sterling & Irwin, 2015; Dwivedi et al., 2023).** The ZINC-12K dataset comprises 12,000 molecular graphs, extracted from the ZINC database, which is a collection of commercially

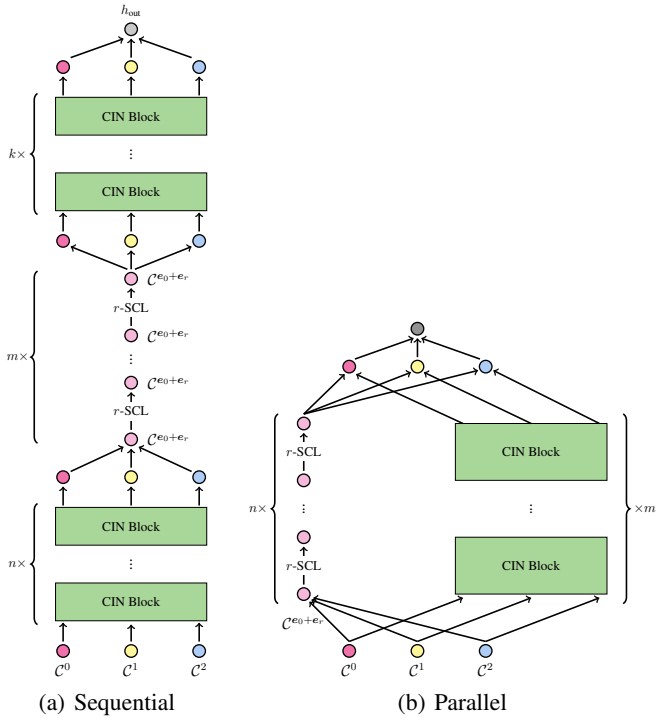

Figure 14: Experiments are using two types of SMCN tensor diagrams. Sequential diagrams 14(a) stacks CIN blocks and SCL updates; Parallel diagrams 14(b) performs SCL updates and CIN blocks in parallel.

available chemical compounds. These molecular graphs vary in size, ranging from 9 to 37 nodes each. In these graphs, nodes correspond to heavy atoms, encompassing 28 distinct atom types. Edges in the graphs represent chemical bonds, with three possible bond types. We perform regression on the constrained solubility (logP) of the molecules. The dataset is pre-partitioned into training, validation, and test sets, containing 10,000, 1,000, and 1,000 molecular graphs, respectively.

For this experiment, we use an SMCN model implementing a sequential tensor diagram. The architecture consists of a single CIN block, followed by a stack of five 1-SCL layers, and concludes with an additional CIN block. Each CIN block has an embedding dimension of 85, while each SCL layer uses an embedding dimension of 70, resulting in a model with fewer than 500k parameters, as outlined in Dwivedi et al. (2023). The readout layer is defined according to Equation 95, where $\text{agg}_2$, $\text{agg}_3$, and $\text{agg}_4$ are zero functions, and $\text{agg}_0$ and $\text{agg}_1$ are a sum aggregation. Since we observed that this model converges slowly, we trained it for 2000 epochs, following the approach of Ma et al. (2023). The learning rate is initialized at 0.001 and decays by a factor of 0.5 every 300 epochs.

**OGB Datasets (Hu et al., 2020).** MOLHIV and MOLESOL are molecular property prediction datasets, adapted by the Open Graph Benchmark (OGB) from MoleculeNet. These datasets employ a unified featurization for nodes (atoms) and edges (bonds), encapsulating various chemophysical properties. The task in MOLHIV is to predict the capacity of compounds to inhibit HIV replication. The task in MOLESOL is regression on water solubility (log solubility in mols per liter) for common organic small molecules.

For both datasets, we use an SMCN model that utilizes a parallel tensor diagram. For MOLHIV the architecture consists of two CIN blocks, and tow parallel 2-SCL blocks. CIN blocks have an embedding dimension of 64 and dropout in between layers with probability of 0.2 while SCL layers have an embedding dimension of 24 and dropout in between layers with probability of 0.5. The final SCL layer is followed by a non learnable pooling operation as per Equation 94. The readout layer is defined according to Equation 95, where $\text{agg}_3$, and $\text{agg}_4$ are zero functions, and $\text{agg}_0$, $\text{agg}_1$ and $\text{agg}_2$ are a mean aggregation. The mdoel is trained for 100 epochs with a constant learning rate of 0.0001.

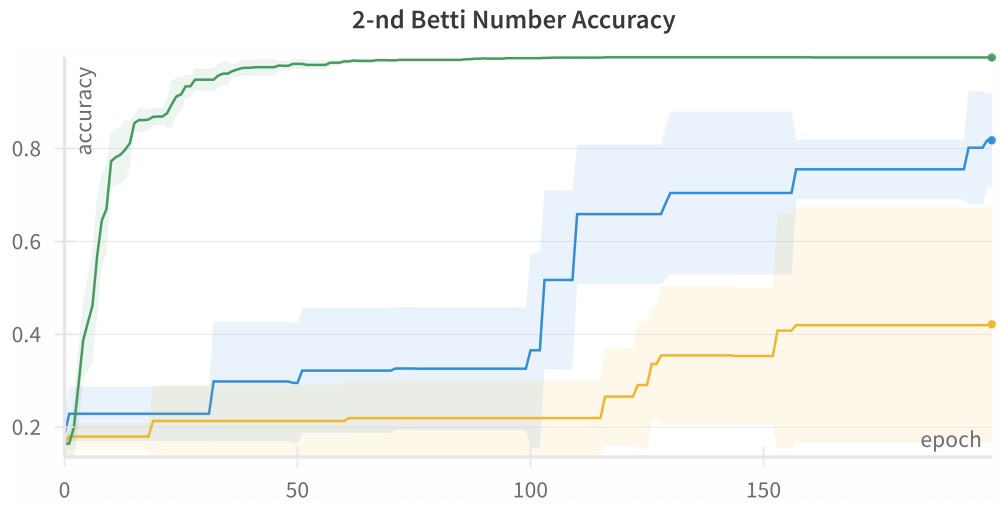

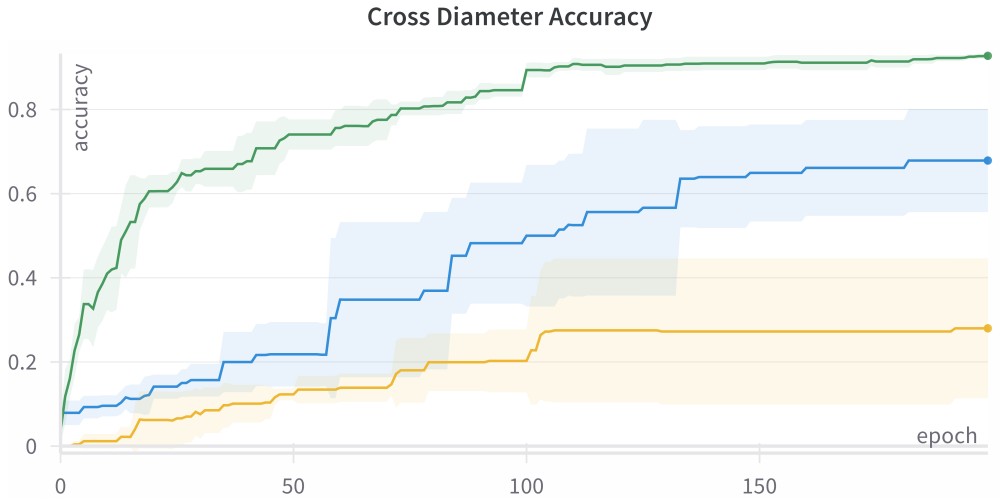

Figure 15: The accuracy per epoch of both synthetic lift experiments. SMCN is depicted in green, Custom HOMP is depicted in blue and CIN is depicted in yellow.

Table 3: Accuracy on the MUTAG dataset.

| Model | Accuracy ($\uparrow$) |
|---|---|
| GIN (Xu et al., 2018) | $89.4 \pm 5.6\%$ |
| CIN (Bodnar et al., 2021a) | $92.7 \pm 6.1\%$ |
| PPGN (Maron et al., 2019) | $90.6 \pm 8.7\%$ |
| DS-GNN (Bevilacqua et al., 2021) | $91.0 \pm 4.8\%$ |
| DSS-GNN (Bevilacqua et al., 2021) | $91.1 \pm 7.0\%$ |
| SUN (Frasca et al., 2022) | $92.7 \pm 5.8\%$ |
| SMCN (ours) | $92.5 \pm 6.2\%$ |

Table 4: **Train time (seconds per epoch).** Time measurements from training each of the models. Measures are averaged across 10 epochs, starting from epoch number 20.

| | SMCN | GNN-SSWL+ | CIN |
|---|---|---|---|
| ZINC | $7.39 \pm 0.17$ | $9.65 \pm 0.19$ | $5.35 \pm 0.33$ |
| MOLHIV | $17.70 \pm 0.42$ | $51.02 \pm 0.25$ | $14.34 \pm 0.27$ |

For MOLESOL the architecture consists of two custom HOMP blocks, and two parallel 1-SCL layers. HOMP blocks have an embedding dimension of 16 while SCL layers have an embedding dimension of 228 where neither block uses dropout. The final SCL block is followed by a non learnable pooling operation as per Equation 94. The readout layer is defined according to Equation 95, where $\text{agg}_2$, $\text{agg}_3$, and $\text{agg}_4$ are zero functions, and $\text{agg}_0$ and $\text{agg}_1$ are a mean aggregation. The model is trained for 200 epochs with a constant learning rate of 0.0001.

**MUTAG.** The MUTAG dataset, part of the TUDataset benchmarks (Morris et al., 2020), comprises 188 molecular graphs. The task is to identify mutagenic molecular compounds, which are relevant for the development of potentially marketable drugs (Kazius et al., 2005; Riesen & Bunke, 2008). Our training setup and evaluation procedure adhere to those outlined in Xu et al. (2018). For this experiment, the SMCN model utilizes a sequential tensor diagram constructed with a single stack of six 2-SCL blocks, followed by a non-learnable pooling procedure, as detailed in Equation 94. Results are presented in Table 3.

### G.4 RUNTIME EVALUATIONS

To empirically measure the runtime differences between SMCN, subgraph GNNs and HOMP, we ran several wall clock measurements for data set construction, single epoch training and test set evaluation. We evaluate the SMCN variants used for the ZINC and MOLHIV experiments, GNN-SSWL+ which is the backbone subgraph GNN for SMCN, and CIN, a standard HOMP model. Runtime was evaluated across 10 runs, all experiments ran on a single NVIDIA A100 48GB GPU. Dataset construction times (in seconds) are:

- ZINC: $322.21 \pm 8.764$,

- MOLHIV: $678.01 \pm 11.38$.

We used the same lifting procedures and dataset construction for both SMCN and CIN so construction times are identical. Results for the train/test times appear in Tables 4 and 5 respectively. SMCN incurs a computational overhead of approximately $23\%$ on the MOLHIV benchmark and $38\%$ on ZINC benchmark compared to CIN (trade-off for its improved predictive performance). Additionally SMCN consistently surpasses subgraph networks in runtime, achieving a 2.9x speedup on MOLHIV

Table 5: **Test time (seconds per full test set inference).** Time measurements for inference over the entire test set. Results are averaged over 10 runs.

| | SMCN | GNN-SSWL+ | CIN |
|---|---|---|---|
| ZINC | $0.93 \pm 0.08$ | $1.04 \pm 0.03$ | $0.71 \pm 0.05$ |
| MOLHIV | $2.09 \pm 0.15$ | $3.07 \pm 0.03$ | $2.02 \pm 0.12$ |

and a $1.3$x speedup on ZINC. This improvement over subgraph networks stems from the fact SMCN uses fewer subgraphs updates and leverages higher order topological information instead.

