# OpenReview forum: "Topological Blindspots: Understanding and Extending Topological Deep Learning Through the Lens of Expressivity"
_ICLR.cc/2025/Conference — ICLR 2025 Oral_

### Official Review · Reviewer_8Wou · 2024-11-02

**Soundness:** 3
**Presentation:** 2
**Contribution:** 3
**Rating:** 8
**Confidence:** 3

**Summary:**

The paper presents an in-depth exploration of Topological Deep Learning (TDL) architectures through the lens of Higher-Order Message Passing (HOMP). The authors investigate HOMP's expressivity limitations in capturing topological invariants (e.g., orientability, planarity, diameter, and homology) in combinatorial complexes and propose two novel architectures: Multi-Cellular Networks (MCN) and Scalable Multi-Cellular Networks (SMCN). These architectures aim to enhance HOMP’s expressivity in distinguishing between topological structures. Empirical evaluations using newly designed benchmarks and real-world graph datasets demonstrate that SMCN outperforms existing models, highlighting the potential of using expressive topological information in TDL.

**Strengths:**

1.	The authors provide a rigorous examination of HOMP’s expressivity limitations concerning fundamental topological and metric invariants, establishing the groundwork for understanding the weaknesses in current TDL approaches.

2.	The introduction of MCN and SMCN provides new pathways for achieving higher expressivity. The authors demonstrate that MCN can theoretically achieve full expressivity, while SMCN offers a computationally feasible alternative, balancing expressivity with scalability.

3.	SMCN demonstrates substantial improvements over traditional HOMP and GNNs, validating the model's efficacy in capturing and leveraging topological features in learning tasks.

**Weaknesses:**

1.	While SMCN offers a scalable alternative to MCN, it still encounters significant challenges in managing large and complex combinatorial complexes due to its super-linear scaling with respect to the number of cells. It would be beneficial for the authors to provide a comparative analysis of SMCN’s runtime performance against existing HOMP methods and other GNNs, such as CIN (Bodnar et al., 2021) and the backbone subgraph GNN used in SMCN.

2.	The paper would benefit from a more detailed rationale for why topological invariants—such as diameter, orientability, planarity, and homology—are critical for machine learning models to differentiate. Specifically, it would be helpful to address the importance of these invariants in machine learning tasks, either through empirical evidence, theoretical reasoning, or relevant literature.

3.	There appears to be a discrepancy between SMCN’s theoretical expressivity and its empirical accuracy. For instance, it is unclear why SMCN does not achieve full accuracy in predicting cross-diameter and the second Betti number, which warrants further investigation.

**Questions:**

1.	To enhance readability and comprehension, the main paper should be more self-contained by clearly explaining topological invariants—such as diameter, orientability, planarity, and homology. For clarity, consider presenting formulas or examples within the main text (e.g., rather than relying solely on descriptions on line 233).

2.	In the introduction of the concept of CC covering, an illustrative case could help readers grasp this concept more intuitively.

3.	Could the authors discuss the proposed method in relation to relevant research? For instance, [1] proposed a cycle-invariant positional encoding where cell/cycle features are initially encoded using an invariant network and subsequently incorporated into a standard GNN as additional edge features. A comparison would contextualize SMCN’s contributions within the broader landscape of topological GNN methods.

[1] Yan Z, Ma T, Gao L, et al. Cycle invariant positional encoding for graph representation learning. LoG 2024.

---

> ### Author Response · Authors · 2024-11-17
>
> We thank the reviewer for the positive review and constructive feedback. We address the concerns and questions raised by the reviewer below:
> - **Role of topology in ML (W2):** The reviewer raises an important question that is central to the wider ongoing research in TDL: "What is the significance of topological information to machine learning tasks?" This is indeed an open problem in TDL, highlighted as Open Problem 9 in a recent position paper [5]. In response to this question, numerous TDL approaches attempt to incorporate topological information, either by lifting objects such as graphs into topological spaces (e.g., HOMP models) or to inject specific topological features directly into the learning pipeline [2, 6, 10, 11, 12]. The papers that develop these approaches often show they have desirable theoretical properties, or empirically demonstrate improved performance, suggesting that incorporating topology can meaningfully enhance model capabilities. Section 2 of the paper reviews several methods, which use topological and metric invariants closely related to our discussion. For example, persistent homology [2,6] relies on homology, while architectures such as [7] use metric features closely related to the graph diameter. Additionally, properties like planarity are of interest, as many real-world graphs, such as those representing molecular structures and electrical circuits, are planar. Collectively, these approaches demonstrate both theoretical and practical advantages, often enhancing the expressive power and performance of base models like MPNNs, which do not inherently incorporate topological information.
>
>   In this context, our paper makes two key contributions to the TDL literature. First, we show that despite being the de facto standard TDL architecture, HOMP is unable to capture fundamental topological properties—often the same properties that other methods explicitly integrate into models (e.g. homology in [2,6] and diameter in [7]). This limitation highlights the need to design more expressive TDL architectures  to fully investigate topology's contribution to learning processes. To that end, we introduce the SMCN framework, which enables models to access a broader range of topological properties. We demonstrate that this framework improves performance on standard graph benchmarks. These findings advance our understanding of the role of topology in machine learning, suggesting a correlation between a model's ability to capture topological information and its empirical performance over real world benchmarks.
> - **Runtime concerns (W1):** In the revised version (to be uploaded within the discussion period), we will provide a wall-clock training/inference comparison of all of SMCN, CIN and subgraph GNNs as well as a more emphasized runtime complexity analysis. See the next comment for timing analysis. Additionally, we note that:
>   - Generally, as is common in GNNs, methods that enhance expressivity often come with increased runtime complexity [1,3,4]. Our approach provides another datapoint on this spectrum.
>   - SMCN is a general framework and runtimes can change for different specifications, e.g. for ZINC we use a version of SMCN with runtime complexity of $O(d \cdot n_0 \cdot n_1)$ and for MOLHIV+MOLESOL+synthetic experiments we use a version with runtime complexity of $O(d \cdot n_0 \cdot n_2)$.
>   - As mentioned in proposition 6.6, SMCN architectures with runtime complexity $O(d \cdot n_0 \cdot n_2)$ are still more expressive than HOMP approaches like CIN, but have a better runtime complexity then subgraph methods, since in most relevant, real-world settings $n_2 \ll n_0$ (e.g. number of cycles is smaller then number of nodes).
> - **Comparison to Yan et. al (Q3):** We thank the reviewer for introducing us to this relevant paper. We will make sure to add a comparison of our approach to this paper in the next version. In terms of experimental results, CycleNet got a score of 0.068 on ZINC while our model got a score of 0.060. In terms of expressive power we provide a proof sketch showing that SMCN is more expressive than CycleNet:
>   - First, recall that CycleNet leverages BasisNet + spectral embedding [8] applied to the 1-Hodge Laplacian of the input graph. The resulting output is then used as edge features, which are subsequently processed by a final MPNN applied to the graph. In cases where the input graphs are simple and undirected, the 1-Hodge laplacian is exactly equal to the 1-0 coadjacency matrix (plus a diagonal term of $2\mathbf{I}$ which can be ignored). SMCN can apply subgraph GNNs to the $co\mathcal{A}_{1,0}$ Hasse graph, and use the resulting features as inputs for an MPNN. This reduces our proof to demonstrating that subgraph GNNs are more expressive than BasisNet. As recently shown in [9], BasisNet + spectral embedding is strictly less expressive than PSWL, a subclass of subgraph GNNs that is itself less expressive than the base subgraph GNN used in SMCN. This completes the proof.

---

> > ### Author Response · Authors · 2024-11-17
> > **Additional Comments and Timing Analysis**
> >
> > **Additional comments:**
> >
> > - **Learning topological properties in the synthetic experiments (W3):** There is a difference between a model's expressive power and its ability to generalize to unseen test datasets. For instance, MLPs are fully expressive, yet they are not able to achieve perfect accuracy on any task as they are not always able to generalize. The same applies here. That being said, SMCN’s enhanced topological expressivity directly improves its ability to learn topological properties, as demonstrated in our experiments.
> >
> > - **Readability and comprehension (Q1):** Most of the topological properties mentioned in the paper are easy to intuitively explain but require a long technical discussion to be rigorously defined. Due to length constraints, we only provided intuitive definitions and referred the reader to the Appendix/external sources for a thorough overview. This practice is common in other TDL studies as well (e.g. [2,6]). If the reviewer thinks this would be helpful, we can add full definitions of the topological properties in the Appendix.
> >
> > - **CC covering example (Q2):** We have given an intuitive example of a CC covering in the proof sketch of Theorem 4.3, and depicted it in Figure 3. Concrete constructions of covering maps are given in several instances in appendix A.2, and A.3 (Lemma A.5, Proposition A.12, Proposition A.15 and Proposition A.17) and several more figures depicting coverings are given (Figure 9, Figure 10). Due to the page limit we may not be able to add any of these to the main text, but we will make clearer references to the Appendix for better readability.
> >
> > **Timing analysis:**
> > - Dataset construction times (seconds):
> >   - ZINC: 322.21 ± 8.764
> >   - MOLHIV: 678.01 ± 11.38
> >
> > - Train time (seconds) / performance:
> > | Dataset | SMCN | GNN-SSWL+ | CIN |
> > |--------------|------------------|------------------|------------------|
> > | ZINC  | 7.39 ± 0.17 / 0.060 ± 0.004 | 9.65 ± 0.19 / 0.070 ± 0.005 | 5.35 ± 0.33 / 0.079 ± 0.006 |
> > | MOLHIV | 17.70 ± 0.42 / 81.16 ± 0.90 | 51.02 ± 0.25 / 79.58 ± 0.35 | 14.34 ± 0.27 / 80.94 ± 0.57 |
> >
> > - Test time (seconds) / performance:
> > | Dataset | SMCN | GNN-SSWL+ | CIN |
> > |--------------|------------------|------------------|------------------|
> > | ZINC  | 0.93 ± 0.08 / 0.060 ± 0.004 | 1.04 ± 0.03 / 0.070 ± 0.005 | 0.71 ± 0.05 / 0.079 ± 0.006 |
> > | MOLHIV | 2.09 ± 0.15 / 81.16 ± 0.90 | 3.07 ± 0.03 / 79.58 ± 0.35 | 2.02 ± 0.12 / 80.94 ± 0.57 |
> >
> > All experiments were done on an NVIDIA A100 48 GB GPU. We used the same lifting procedures and dataset construction for both SMCN and CIN so construction time is identical. SMCN incurs computational overhead of approximately 23% on the MOLHIV benchmark and 38% on ZINC compared to CIN (trade-off for its improved predictive performance). Additionally SMCN consistently outperforms subgraph networks in runtime, achieving a ~2.9x speedup on MOLHIV and a ~1.3x speedup on ZINC. The improvement over subgraph networks stems from the fact SMCN uses significantly fewer subgraphs updates and leverages higher order topological information instead.

---

> > > ### Author Response · Authors · 2024-11-17
> > > **References**
> > >
> > > [1] Beatrice Bevilacqua, Fabrizio Frasca, Derek Lim, Balasubramaniam Srinivasan, Chen Cai, Gopinath Balamurugan, Michael M Bronstein, and Haggai Maron. Equivariant subgraph aggregation networks. arXiv preprint arXiv:2110.02910, 2021.
> > >
> > > [2] Max Horn, Edward De Brouwer, Michael Moor, Yves Moreau, Bastian Rieck, and Karsten Borg- wardt. Topological graph neural networks. arXiv preprint arXiv:2102.07835, 2021
> > >
> > > [3] Haggai Maron, Heli Ben-Hamu, Hadar Serviansky, and Yaron Lipman. Provably powerful graph networks. Advances in neural information processing systems, 32, 2019.
> > >
> > > [4] Christopher Morris, Martin Ritzert, Matthias Fey, William L Hamilton, Jan Eric Lenssen, Gaurav Rattan, and Martin Grohe. Weisfeiler and leman go neural: Higher-order graph neural networks. In Proceedings of the AAAI conference on artificial intelligence, volume 33, pages 4602–4609, 2019.
> > >
> > > [5] Theodore Papamarkou, Tolga Birdal, Michael M Bronstein, Gunnar E Carlsson, Justin Curry, Yue Gao, Mustafa Hajij, Roland Kwitt, Pietro Lio, Paolo Di Lorenzo, et al. Position: Topological deep learning is the
> > > new frontier for relational learning. In the Forty-first International Conference on Machine Learning, 2024.
> > >
> > > [6] Bastian Rieck. On the expressivity of persistent homology in graph learning. arXiv preprint arXiv:2302.09826, 2023.
> > >
> > > [7] Bohang Zhang, Shengjie Luo, Liwei Wang, and Di He. Rethinking the expressive power of gnns via graph biconnectivity. arXiv preprint arXiv:2301.09505, 2023.
> > >
> > > [8] Lim, Derek, et al. "Sign and basis invariant networks for spectral graph representation learning." arXiv preprint arXiv:2202.13013 (2022).
> > >
> > > [9] Bohang Zhang, Lingxiao Zhao, and Haggai Maron. On the expressive power of spectral invariant graph neural networks. arXiv preprint arXiv:2406.04336, 2024.
> > >
> > > [10] Bodnar, Cristian, et al. "Weisfeiler and lehman go cellular: Cw networks." Advances in neural information processing systems 34 (2021): 2625-2640.
> > >
> > > [11] Bodnar, Cristian, et al. "Weisfeiler and lehman go topological: Message passing simplicial networks." International Conference on Machine Learning. PMLR, 2021.
> > >
> > > [12] Hajij, Mustafa, et al. "Topological deep learning: Going beyond graph data." arXiv preprint arXiv:2206.00606 (2022).

---

> > ### Comment · Reviewer_8Wou · 2024-11-17
> >
> > I thank the authors for their rebuttal. After reading all the reviews and responses, I think the authors have largely addressed my concerns, and will raise my score accordingly.

---

### Official Review · Reviewer_sF9V · 2024-11-03

**Soundness:** 3
**Presentation:** 2
**Contribution:** 4
**Rating:** 8
**Confidence:** 2

**Summary:**

The paper presents a new way to define expressivity but from a topological perspective. The paper shows that existing Topological Deep Learning (TDL) models have trouble estimating certain important topological properties such as diameter, orientability, planarity, and homology. The authors then propose a new model called MCN and its scalable version SMCN that can capture the topological properties better. The paper also presents new benchmarks focusing on learning topological properties of complexes.

**Strengths:**

The paper presents a novel essential perspective that differentiates TDL and deep learning on graphs, which used to be compared under the same umbrella previously (in terms of graph expressivity and benchmarks). This work partly bridges the gap between TDL and traditional computational topology methods, which largely focus on homology. The paper also presents theoretical insights showing that higher-order message passing is incapable of capturing certain topological metrics. The experiments also support these insights. The experiments also constrain on parameter budgets to show the effectiveness of the method with respect to the baseline. The paper acknowledges the weakness of MCN if we consider higher-order spaces, so the authors present a scalable version that leverages subgraph GNNs to encode higher-order features. The novel benchmarks are a great contribution which can facilitate a better comparison standard for TDL methods.

**Weaknesses:**

The paper is hard to follow and the presentation is not good. For example, the authors can be more explicit on lines 309 and 310 and discuss (1), (2), and (3) more instead of stating them. The notations involve many upper scripts and lower scripts while not explaining their purposes clearly make the formula confusing (line 319 and 320). The paper isn’t self-contained; for example, the paper can discuss more about IGN as the model itself leverages an architecture similar to IGN. The same thing applies to Subgraph GNNs. Please also refer to the question section for further comments on clarity. Also, even when the paper focuses on expressivity from a topological perspective, I think it would be helpful to include a brief section discussing the proposed method with respect to graph expressivity so that there is a smoother transition between deep learning on graphs and TDL. Lastly, please refer to question 3 for experiments on runtime and lifting time.

**Questions:**

1. For Figure 5, can you elaborate more on colored nodes of SMCN and MCN? I think this part can be improved to make it clearer for the audience.
2. For Figure 7, the superscript for \mathcal{X} and \mathcal{H} isn’t discussed in the main text, so it is confusing.
3. Can the authors comment on the lifting and runtime complexities with respect to CIN? I think the paper only mentions the runtime complexity and neglects the discussion. It is also helpful to include an experiment on wall-clock training/inference time to see the model scalability in practice when comparing with other TDL models.

---

> ### Author Response · Authors · 2024-11-17
> **Response to Reviewer sF9V**
>
> We greatly appreciate the reviewer’s positive feedback and constructive criticism. We address the concerns and questions raised by the reviewer below:
>
> - **Clarity:** We thank the reviewer for their insightful suggestions! We’ll integrate these notes in the revised version of the paper (to be uploaded within the discussion period). Specifically, we’ll make an effort to add further discussion and illustrations to elaborate on neighborhood functions (equations (1), (2) and (3)), multicellular cochains (lines 309 and 310) and group action indices (lines 319 and 320). We note that due to length restrictions, some of these might be added to the Appendix.
>
> - **IGN and subgraph GNNs overview:** We’ll additionally add a deeper overview of IGNs and subgraph GNNs in the Appendix.
>
> - **Graph expressivity:** We note that the SMCN and MCN frameworks introduced in the paper are both direct generalizations of the HOMP framework, including CIN [2]. As demonstrated in [2], CIN is strictly more expressive than MPNNs, and this property extends to both SMCN and MCN. Furthermore, the SMCN framework subsumes subgraph-based architectures like ESAN and GNN-SSWL+ [1,6], making it at least as expressive as these methods. The expressive power of these architectures has been thoroughly studied [3,6], providing additional context for understanding the expressivity of SMCN relative to other graph-based approaches. This discussion will be added to the Appendix.
>
> - **Figure 5 tensor diagram node clarification (Q1):** These nodes represent types of multicellular cochain spaces, we will elaborate on them in the updated version.
>
> - **Figure 7 notation clarification:** Thank you for this comment! The superscript indicates cell/node marking in the complex/graph (e.g., a subgraph layer can process a bag of graphs with marked nodes) will add a clarification in the updated version.
>
> - **Runtime analysis:** In the revised version, we will include a wall-clock training and inference time comparison between SMCN, CIN, and subgraph GNNs. For your convenience, we also provide this comparison below:
>
>   - Dataset construction times (seconds):
>     - ZINC: 322.21 ± 8.764
>     - MOLHIV: 678.01 ± 11.38
>   - Train time (seconds) / performance:
>     | Dataset | SMCN | GNN-SSWL+ | CIN |
>     |--------------|------------------|------------------|------------------|
>     | ZINC  | 7.39 ± 0.17 / 0.060 ± 0.004 | 9.65 ± 0.19 / 0.070 ± 0.005 | 5.35 ± 0.33 / 0.079 ± 0.006 |
>     | MOLHIV | 17.70 ± 0.42 / 81.16 ± 0.90 | 51.02 ± 0.25 / 79.58 ± 0.35 | 14.34 ± 0.27 / 80.94 ± 0.57 |
>
>   - Test time (seconds) / performance:
>     | Dataset | SMCN | GNN-SSWL+ | CIN |
>     |--------------|------------------|------------------|------------------|
>     | ZINC  | 0.93 ± 0.08 / 0.060 ± 0.004 | 1.04 ± 0.03 / 0.070 ± 0.005 | 0.71 ± 0.05 / 0.079 ± 0.006 |
>     | MOLHIV | 2.09 ± 0.15 / 81.16 ± 0.90 | 3.07 ± 0.03 / 79.58 ± 0.35 | 2.02 ± 0.12 / 80.94 ± 0.57 |
>
>   All experiments were done on an NVIDIA A100 48 GB GPU. We used the same lifting procedures and dataset construction for both SMCN and CIN so construction time is the identical.  SMCN incurs computational overhead of approximately 23% on the MOLHIV benchmark and 38% on ZINC compared to CIN (trade-off for its improved predictive performance). Additionally SMCN consistently outperforms subgraph networks in runtime, achieving a ~2.9x speedup on MOLHIV and a ~1.3x speedup on ZINC.  The improvement over subgraph networks stems from the fact SMCN uses fewer subgraph updates and leverages higher order topological information instead.
>
> **References:**
>
> [1] Beatrice Bevilacqua, Fabrizio Frasca, Derek Lim, Balasubramaniam Srinivasan, Chen Cai, Gopinath Balamurugan, Michael M Bronstein, and Haggai Maron. Equivariant subgraph aggregation networks. arXiv preprint arXiv:2110.02910, 2021.
>
> [2] Cristian Bodnar, Fabrizio Frasca, Nina Otter, Yuguang Wang, Pietro Lio, Guido F Montufar, and Michael Bronstein. Weisfeiler and lehman go cellular: Cw networks. Advances in neural information processing systems, 34:2625–2640, 2021.
>
> [3] Fabrizio Frasca, Beatrice Bevilacqua, Michael Bronstein, and HaggainMaron. Understanding and extending subgraph gnns by rethinking their symmetries. Advances in Neural Information Processing Systems, 35:31376–31390, 2022.
>
> [4] Haggai Maron, Heli Ben-Hamu, Hadar Serviansky, and Yaron Lipman. Provably powerful graph networks. Advances in neural information processing systems, 32, 2019.
>
> [5] Christopher Morris, Martin Ritzert, Matthias Fey, William L Hamilton, Jan Eric Lenssen, Gaurav Rattan, and Martin Grohe. Weisfeiler and leman go neural: Higher-order graph neural networks. In Proceedings of the AAAI conference on artificial intelligence, volume 33, pages 4602–4609, 2019.
>
> [6] Bohang Zhang, Guhao Feng, Yiheng Du, Di He, and Liwei Wang. A complete expressiveness hierarchy for subgraph gnns via subgraph weisfeiler-lehman tests. International Conference on Machine Learning, pages 41019–41077. PMLR, 2023.

---

> > ### Comment · Reviewer_sF9V · 2024-11-25
> >
> > Thank you very much for addressing my questions!

---

### Official Review · Reviewer_TNgH · 2024-11-03

**Soundness:** 3
**Presentation:** 2
**Contribution:** 3
**Rating:** 8
**Confidence:** 4

**Summary:**

This work studies the expressivity of Topological Deep Learning (TDL) architectures, particularly focusing on the limitations of Higher-Order Message Passing (HOMP) for distinguishing combinatorial complexes. The first half of the paper extends Bamberger’s (2022) work on the expressivity limitations of message-passing Graph Neural Networks (GNNs), which characterized, using covering maps, graphs that GNNs cannot distinguish. In a similar vein, this paper reveals "topological blindspots" in HOMP frameworks: (1) complexes that share a cover are indistinguishable by HOMP, and (2) HOMP cannot distinguish complexes that differ in important topological and metric properties such as diameter, orientability, planarity, and homology. In the second half, the authors address these limitations by adapting techniques from expressive graph architectures that process features over tuples of nodes. Similarly, the work extends HOMP with multi-cellular feature spaces and equivariant linear updates. This extension, Multi-Cellular Networks (MCN), achieves full expressive power, allowing it to (1) distinguish non-isomorphic complexes and (2) differentiate complexes based on properties like diameter, 0-th homology group, and also distinguish between a Moebius strip and a cylinder (which disagree on planarity). The work also introduces a more computationally scalable version of MCN, aptly called SMCN.  Lastly, the authors empirically validate MCN and SMCN on benchmarks designed to capture topological expressivity, demonstrating the superiority of these architectures over standard HOMP and expressive GNN models.

**Strengths:**

(++++) **Novelty and Relevance**: This work is new and addresses questions of significant importance and urgency for the Topological Deep Learning (TDL) community.

(++++) **Theoretical Contribution**: The theoretical contribution is strong, rigorous, and sound. The answers provided to the question considered by the work are satisfying.

(++) **Empirical Validation**: The authors validate their models with real-world and synthetic benchmarks designed to capture topological expressivity, demonstrating clear improvements over standard HOMP and expressive GNN models.

(++) **New Benchmarks**: The work introduces benchmarks that test models on topological invariants to assess TDL expressivity, which will serve as a valuable tool for the TDL community.

**Weaknesses:**

(---) **Presentation**: The architecture descriptions may be opaque for readers unfamiliar with TDL. Section 5 gives examples of CC data that the new multicellular nodes can encode, but it is unclear which nodes and connections should be included and when. For example, the choice of multicellular nodes and connections such as in the example tensor diagrams in Figure 5 would benefit from additional explanation.

(---) **Limited Empirical Evaluation**: The proposed architectures are benchmarked on only three-world datasets.

(--) **Related Work**: Although Section 4 extends Bamberger’s (2020) result, this work is only mentioned once in Section 4.1. An earlier mention in Section 2 (Previous Work) would help readers place this work within a broader research context.

**Questions:**

1. The authors demonstrate that HOMP can be extended to achieve full or greater expressivity with components that may make the proposed models computationally impractical for large combinatorial complexes. Is this an inherent limitation that comes with achieving full/greater expressivity, or did the authors only intend to demonstrate that such levels of expressivity are achievable and the proposed extensions sufficed for that purpose?
2. Could the authors give recommendations or guidelines for which additional multicellular nodes and connections to include in the MCN and SMCN models? For example, how should a practitioner choose and connect multicellular nodes in the MCN and SMCN layers as in the example in Figure 5?
3. Could the authors expand their real-world benchmarks with some more tasks/datasets? For example, the TUDatasets or trajectory classification tasks from Bodnar et al. (2021).
4. Are the group actions in Section 5 required to be compatible with the underlying complex? For example, are permutations that exchange nodes while fixing higher-dimensional (rank) simplices allowed?
5. Could the authors make their code publicly available?
6. Could the authors provide a brief account of Bamberger (2020) in the main text to help contextualize how earlier methods relate to and may have motivated the approaches developed in the first half of this work?

---

> ### Author Response · Authors · 2024-11-17
> **Response to Reviewer TNgH**
>
> We thank the reviewer for highlighting the strengths of our paper, particularly its novelty and relevance to the TDL community. We also appreciate the reviewer’s constructive feedback and address their comments below:
>
> - **Limited Empirical Evaluation (W2 + Q3):** While our work has a strong theoretical focus, we also place significant emphasis on experimental validation. Not only have we introduced several novel benchmarks that address a gap in the field (the torus dataset and topological property prediction tasks), but our paper also includes more extensive experimental evaluation compared to typical theory-focused papers. For instance, [4] (Outstanding Paper Award at ICLR 2023) evaluates only on a single real-world dataset (ZINC) alongside synthetic experiments, [5] (Oral Presentation at ICML2023) explores three real-world benchmarks (matching ours) with one synthetic experiment, and [6] (200+ citations) presents no real-world benchmarks at all.
>
>   Nevertheless, in response to this feedback, we will make an effort to expand our real-world evaluation in the revised version by adding experiments on several datasets from the TUDatasets repository. Finally, we note that we cannot include the trajectory classification tasks from Bodnar et al. (2021) as these rely on cells with orientation, which is out of the scope of our architecture.
>
> - **Presentation (W1 + Q2):** Due to the paper's page limit, we only provided an overview of the architectures in the main text, providing a more in-depth description in the Appendix. Specifically, in Appendix B and C, we provide a more thorough description of the general MCN and SMCN frameworks respectively. In addition, as the SMCN framework offers a versatile space of layers and updates, in Appendix F.1 we focus on two potential types of SMCN implementations and thoroughly describe their exact forward pass. Despite this, we agree the main text explanations could be clearer. In the revised version (to be uploaded within the discussion period), we will:
>   - Add details to section 5 to explain what nodes to use and when for each of the examples.
>   - Expand the description of Figure 5 to better illustrate the relationship between node types and update rules
>   - Add specific guidelines for practitioners on selecting appropriate multicellular structures.
>
>   Overall, we observe that SCL layers updating edge features, combined with "sequential tensor diagrams", tend to increase the risk of overfitting but perform well on tasks like ZINC, where the training and test distributions are closely aligned. In contrast, SCL layers that update 2-cells, paired with "parallel tensor diagrams", are more effective for tasks sensitive to overfitting. All relevant definitions, including those of "sequential" and "parallel" tensor diagram are provided in Appendix F1.
>
> - **Related Work (W3 + Q6)**: We agree that the connection to Bamberger (2020) should be better contextualized. We will:
>   - Add a dedicated mention in Section 2 explaining Bamberger's work on covering maps.
>   - Clarify how our topological criterion extends their results.
>
> - **Computational practicality (Q1):** The trade-off between expressivity and computational complexity is well-documented in machine learning, particularly in graph learning (e.g. [1,2,3]). Our work provides another data point in this trade-off. Moreover, our framework is flexible – specific instantiations can make different complexity trade-offs. For example, we can design models whose complexity primarily depends on the number of 2-cells, which are typically much fewer than nodes or edges. This enables practitioners and other researchers to find the optimal expressivity/computation trade-off for their application. We will clarify these points in the revised manuscript. Additionally, as discussed in the general comment, the runtime of our architecture is comparable to that of CIN (a standard HOMP architecture) and is better than that of subgraph GNNs, see next comment for timing analysis.
>
> - **Group actions compatibility (Q4):** Thank you for this important question! No, the group actions do not need to be compatible with the underlying complex structure. The group acts on the enumeration of the cells, not on the complex itself. For example, a cell permutation can reorder the cells c₁ = {s₁, s₂}, c₂ = {s₃, s₄} to c₁ = {s₃, s₄}, c₂ = {s₁, s₂}, effectively shuffling the cell labels while preserving the internal structure of each cell. We will clarify this point in the revised text.
>
> - **Code availability (Q5):** Yes, we will make our code publicly available as soon as possible. We are currently in the process of cleaning and organizing it to ensure it is user-friendly for public release. We aim to publish the code within the discussion period, and commit to making it available before submission of the camera-ready version.

---

> ### Author Response · Authors · 2024-11-17
> **Timing Analysis and References**
>
> **Timing analysis:**
> - Dataset construction times (seconds):
>   - ZINC: 322.21 ± 8.764
>   - MOLHIV: 678.01 ± 11.38
>
> - Train time (seconds) / performance:
> | Dataset | SMCN | GNN-SSWL+ | CIN |
> |--------------|------------------|------------------|------------------|
> | ZINC  | 7.39 ± 0.17 / 0.060 ± 0.004 | 9.65 ± 0.19 / 0.070 ± 0.005 | 5.35 ± 0.33 / 0.079 ± 0.006 |
> | MOLHIV | 17.70 ± 0.42 / 81.16 ± 0.90 | 51.02 ± 0.25 / 79.58 ± 0.35 | 14.34 ± 0.27 / 80.94 ± 0.57 |
>
> - Test time (seconds) / performance:
> | Dataset | SMCN | GNN-SSWL+ | CIN |
> |--------------|------------------|------------------|------------------|
> | ZINC  | 0.93 ± 0.08 / 0.060 ± 0.004 | 1.04 ± 0.03 / 0.070 ± 0.005 | 0.71 ± 0.05 / 0.079 ± 0.006 |
> | MOLHIV | 2.09 ± 0.15 / 81.16 ± 0.90 | 3.07 ± 0.03 / 79.58 ± 0.35 | 2.02 ± 0.12 / 80.94 ± 0.57 |
>
> All experiments were done on an NVIDIA A100 48 GB GPU. We used the same lifting procedures and dataset construction for both SMCN and CIN so construction time is identical. SMCN incurs computational overhead of approximately 23% on the MOLHIV benchmark and 38% on ZINC compared to CIN (trade-off for its improved predictive performance). Additionally SMCN consistently outperforms subgraph networks in runtime, achieving a ~2.9x speedup on MOLHIV and a ~1.3x speedup on ZINC. The improvement over subgraph networks stems from the fact SMCN uses significantly fewer subgraphs updates and leverages higher order topological information instead.
>
> **References:**
>
> [1] Beatrice Bevilacqua, Fabrizio Frasca, Derek Lim, Balasubramaniam Srinivasan, Chen Cai, Gopinath Balamurugan, Michael M Bronstein, and Haggai Maron. Equivariant subgraph aggregation networks. arXiv preprint arXiv:2110.02910, 2021.
>
> [2] Haggai Maron, Heli Ben-Hamu, Hadar Serviansky, and Yaron Lipman. Provably powerful graph networks. Advances in neural information processing systems, 32, 2019.
>
> [3] Christopher Morris, Martin Ritzert, Matthias Fey, William L Hamilton, Jan Eric Lenssen, Gaurav Rattan, and Martin Grohe. Weisfeiler and leman go neural: Higher-order graph neural networks. In Proceedings of the AAAI conference on artificial intelligence, volume 33, pages 4602–4609, 2019.
>
> [4] Zhang, Bohang, et al. "Rethinking the Expressive Power of GNNs via Graph Biconnectivity." The Eleventh International Conference on Learning Representations.
>
> [5]  Omri Puny, Derek Lim, Bobak Kiani, Haggai Maron, and Yaron Lipman. Equivariant polynomials for graph neural networks. In International Conference on Machine Learning, pages 28191–28222. PMLR, 2023.
>
> [6] Ralph Abboud, Ismail Ilkan Ceylan, Martin Grohe, and Thomas Lukasiewicz. The surprising power of graph neural networks with random node initialization. arXiv preprint arXiv:2010.01179, 2020.

---

> > ### Comment · Reviewer_TNgH · 2024-11-18
> >
> > I thank the authors for their thorough rebuttal. All my questions and concerns have been addressed clearly and thoughtfully, and I am raising my score accordingly.

---

### Author Response · Authors · 2024-11-17
**General Response**

We thank all the reviewers for their positive evaluations and constructive feedback. We underscore four key contributions of our paper mentioned by the reviewers:

1. Our paper addresses a fundamental and unexplored question in TDL (**TNgH**, **sF9V**).
2. Our theory section answers this question, exposing a fundamental limitation of a large class of TDL architectures (**TNgH**, **sF9V**, **8Wou**).
3. We construct provably expressive architectures and empirically demonstrate their improved performance (**TNgH**, **8Wou**).
4. We construct valuable datasets and benchmarks (**TNgH**, **sF9V**).

Additionally, in response to reviewers' inquiries regarding computational practicality, we conducted a runtime comparison between SMCN, CIN and the backbone subgraph GNNs. SMCN incurs computational overhead of approximately 23% on the MOLHIV benchmark and 38% on ZINC compared to CIN (trade-off for its improved predictive performance). Additionally SMCN consistently outperforms subgraph networks in runtime, achieving a ~2.9x speedup on MOLHIV and a ~1.3x speedup on ZINC.

---

### Author Response · Authors · 2024-11-24
**Updated Manuscript and Code**

Dear reviewers, we express our sincere appreciation for the positive evaluations. We have updated the manuscript incorporating your suggestions, with all changes highlighted in light blue for easy reference. We additionally provide code for three of our experiments in the supplementary material, the entire repo would be added to the camera ready version.

---

### Meta-Review · Area_Chair_fY7u · 2024-12-12

**Metareview:**

The paper provides a thorough examination of the expressivity limitations of Higher-Order Message Passing (HOMP) architectures in Topological Deep Learning. It identifies specific "topological blindspots" in HOMP, such as its inability to distinguish between complexes differing in topological and metric properties like diameter, orientability, planarity, and homology. To address these limitations, the authors propose two new architectures: Multi-Cellular Networks (MCN) and Scalable Multi-Cellular Networks (SMCN). MCN achieves full expressive power, allowing it to distinguish non-isomorphic complexes and differentiate key topological features. SMCN offers a computationally scalable variant of MCN, maintaining strong expressivity while reducing complexity. The authors also introduce benchmarks designed to evaluate topological expressivity, demonstrating empirical improvements of the proposed methods over baseline HOMP and GNN models on these benchmarks and real-world graphs. Overall, this work conducts a strong generalization of the existing extensive studies on GNN expressive power to TDL.

**Strengths**  The research is strong across many aspects including theoretical analysis, algorithmic design, and extensive evaluation. In particular, the systematic extension of the notion of GNN expressive power to TDL is impressive and useful.

**Weaknesses** The main concern is about presentation and clarity. The paper is difficult to follow, particularly for readers unfamiliar with TDL. Complex notations and insufficient explanation of architectural components (e.g., Figure 5 and the superscripts in Figure 7) reduce accessibility. Moreover, in practice, the complexity of the proposed methods, compared with other more efficient models with even better empirical performance should be better justified.

Overall, I think this work is a solid theoretical contribution to the field of graph and geometric deep learning, though empirical value deserves further justification and more intuitive exposition is also recommended.

**Additional Comments On Reviewer Discussion:**

The authors address the reviewers' concerns in the discussion, which wins a unanimous acceptance of the work.

---

### Decision · Program_Chairs · 2025-01-22

Accept (Oral)